# PolypSense3D: A Multi-Source Benchmark Dataset for Depth-Aware Polyp Size Measurement in Endoscopy

**Ruyu Liu**[a,b]    **Lin Wang**[a]    **Mingming Zhou**[a]    **Jianhua Zhang**[c,d,*]    **Haoyu Zhang**[a,*]
**Xiufeng Liu**[b]    **Xu Cheng**[d]    **Sixian Chan**[e]    **Yanbing Shen**[f]    **Sheng Dai**[f]
**Yuping Yan**[g]    **Yaochu Jin**[g]    **Lingjuan Lv**[h]

[a] Hangzhou Normal University    [b] Technical University of Denmark
[c] Hohai University    [d] Tianjin University of Technology
[e] Zhejiang University of Technology    [f] Sir Run Run Shaw Hospital Zhejiang
[g] Westlake University    [h] Sony AI
[*] Corresponding authors

*Correspondence to*: Jianhua Zhang <*zjh@ieee.org*>, Haoyu Zhang <*haoyu.zhang@hznu.edu.cn*>
*{ruyu.liu, xiufeng, xu.cheng}@ieee.org, {2023111011034, 2024112011018}@stu.hznu.edu.cn*
*{shenyb2006, colon}@zju.edu.cn, {yanyuping, jinyaochu}@westlake.edu.cn, Lingjuan.Lv@sony.com*

## Abstract

Accurate polyp sizing during endoscopy is crucial for cancer risk assessment but is hindered by subjective methods and inadequate datasets lacking integrated 2D appearance, 3D structure, and real-world size information. We introduce **PolypSense3D**, the first multi-source **benchmark dataset** specifically targeting depth-aware polyp size measurement. It uniquely integrates over 43,000 frames from virtual simulations, physical phantoms, and clinical sequences, providing synchronized RGB, dense/sparse depth, segmentation masks, camera parameters, and millimeter-scale size labels derived via a novel forceps-assisted in-vivo annotation technique. To establish its value, we benchmark state-of-the-art segmentation and depth estimation models. Results quantify significant domain gaps between simulated/phantom and clinical data and reveal substantial error propagation from perception stages to final size estimation, with the best fully automated pipelines achieving an average Mean Absolute Error (MAE) of 0.95 mm on the clinical data subset. Publicly released under CC BY-SA 4.0 with code and evaluation protocols, PolypSense3D offers a standardized platform to accelerate research in robust, clinically relevant quantitative endoscopic vision. The benchmark dataset and code are available at: `https://github.com/HNUicda/PolypSense3D` and `https://doi.org/10.7910/DVN/K13H89`.

## 1 Introduction

Endoscopy is the cornerstone of early gastrointestinal cancer screening, enabling the detection and removal of high-risk polyps and saving millions of lives annually [52, 36]. However, a critical limitation persists: while endoscopists visualize polyp morphology, accurately measuring polyp size in real-time remains a significant challenge. This information gap directly impacts clinical decision-making, as polyp diameter strongly correlates with malignancy risk – polyps over 20 mm carry up to 15% risk, compared to <2% for those under 10 mm [16]. Precise sizing is therefore essential for risk assessment and tailoring surveillance strategies [16]. Current methods, including subjective visual estimation or error-prone ex-vivo measurements [10], are inadequate, often leading to misjudgments and potentially inappropriate patient management. Consequently, an automated,

39th Conference on Neural Information Processing Systems (NeurIPS 2025) Track on Datasets and Benchmarks.

objective, and accurate method for in-vivo polyp size measurement is an urgent clinical imperative and presents a significant challenge for quantitative computer vision.

Recent computer vision advances have sped up progress in endoscopic image analysis [37], particularly in 2D segmentation and 3D reconstruction. Datasets like Kvasir-SEG [19] and CVC-ColonDB [51] have hatched sophisticated 2D polyp segmentation algorithms [44, 14, 23]. However, focusing solely on pixel-level masks, these datasets fundamentally lack the 3D spatial context (depth) and real-world scale information indispensable for physical size measurement. A 200-pixel polyp mask yields no true size about whether the lesion is 3 mm or 15 mm, a distinction critical for clinical action. Conversely, while 3D perception datasets like SimCol3D [41] or EndoMapper [1], derived from simulations [40, 6] or SLAM on phantoms [39], provide valuable geometric information for navigation or scene understanding, they typically lack fine-grained annotations of specific polyp instances and, crucially, omit the corresponding ground-truth physical size measurements needed for metrology. This creates a significant bottleneck for machine learning research targeting clinical size estimation: **segmentation datasets lack the necessary geometric and scale information**, while **3D reconstruction datasets lack the specific, instance-level size labels**. Therefore, no existing public resource provides the integrated multi-modal data (appearance, structure, scale) required to develop, train, and rigorously benchmark machine learning models for the specific, clinically vital task of accurate, depth-aware polyp size estimation directly from endoscopic video.

To fill this critical gap in data resources for quantitative endoscopic vision research, we introduce **PolypSense3D**, the first comprehensive, multi-source benchmark dataset specifically engineered for developing and evaluating depth-aware polyp size measurement techniques. PolypSense3D bridges virtual simulation, physical modeling, and real clinical scenarios, providing a unique resource comprising over 43,000 frames. It distinctively features synchronized RGB images, corresponding dense (virtual/physical-estimated) or sparse (clinical) depth maps, precise 2D segmentation masks, calibrated camera intrinsic parameters, and, most importantly, **verifiable millimeter-scale ground-truth polyp size annotations**. Data spans realistic virtual polyps derived from CT data (sizes 1.79-20.52mm), precisely fabricated polyps in 3D-printed phantoms offering single textures (sizes 4.0-14.89mm), and challenging in-vivo clinical cases (sizes 1.39-9.98mm) annotated using our novel, reproducible biopsy-forceps-assisted technique (detailed in Section 3.3 and validated in Appendix D). By providing all essential components – appearance (RGB), structure (depth), camera geometry (intrinsics), and physical scale (size labels) – within a unified framework, PolypSense3D enables the ML community to tackle the full pipeline from perception (segmentation, depth) to measurement (size quantification). It serves as a much-needed standardized platform to benchmark algorithms, quantify domain shifts, analyze error propagation, and ultimately accelerate the development of robust, clinically relevant quantitative endoscopic vision systems.

The primary contributions of this work are:

- **The PolypSense3D Benchmark Dataset:** We construct and release the first large-scale, multi-source (virtual, physical, clinical) benchmark dataset specifically designed for developing and evaluating depth-aware polyp size measurement algorithms. It features over 16k/43k+ (with polys/total) frames with synchronized RGB, depth, segmentation masks, camera parameters, and ground-truth millimeter-scale size annotations. The dataset's dense virtual and sparse clinical depth enable a pre-training/fine-tuning workflow. This is designed to resolve the scale ambiguity in self-supervised methods and adapt large models for clinically relevant, end-to-end measurement tasks.
- **Novel Verifiable In-Vivo Annotation Protocol:** We develop, detail, and provide initial validation (see Appendix D) for a reproducible biopsy-forceps-assisted annotation strategy, enabling quantitative size and sparse depth estimation directly from standard clinical endoscopic video sequences. This sparse annotation process is both challenging and highly valuable, adding real-world complexity to the benchmark dataset.
- **Comprehensive Benchmarking for ML Models:** We establish strong baseline results on PolypSense3D by evaluating representative state-of-the-art segmentation and depth estimation models. Our analysis quantifies performance across the challenging virtual-physical-clinical domain shifts and investigates error propagation dynamics, demonstrating the dataset's utility for rigorous ML model evaluation and highlighting key research challenges. We also investigated the adaptability of foundational models in real-world scenarios, emphasizing that large-scale foundational models can be initially trained using dense data and subsequently fine-tuned with sparse clinical annotations to better align with the requirements of real-world applications.

- **Open Resources for Transparency and Reproducibility:** We publicly release the PolypSense3D dataset, associated annotation tools (including code for the forceps-based annotation method), baseline implementation code, and evaluation protocols under permissive open-source licenses (CC BY-SA 4.0, MIT License), ensuring transparency and promoting reproducible research within the NeurIPS community and beyond.

## 2 Related Work

Our work builds upon advancements in endoscopic computer vision, specifically in 2D polyp segmentation and 3D perception. We review existing work in these areas to precisely situate the contribution of PolypSense3D.

Table 1: Summary of public endoscopic datasets for polyp analysis. Abbreviations: Polyp Segmentation (PS), Depth Estimation (DE), 3D Reconstruction (3DR), Dense Depth Map (DDM), Sparse Point Cloud (SPC).

| Dataset | Type | Organ | Task | Image Count | Resolution(s) | Depth Type | Depth Source | Mask |
|---------|------|-------|------|-------------|---------------|------------|--------------|------|
| ASU-Mayo [3] | Clinical | Colon | PS | 18902 | – | – | – | ✓ |
| ETIS-Larib [48] | Clinical | Colon | PS | 196 | $1225 \times 966$ | – | – | ✓ |
| CVC-ClinicDB [4] | Clinical | Colon | PS | 612 | $576 \times 768$ | – | – | ✓ |
| CVC-ColonDB [51] | Clinical | Colon | PS | 380 | $574 \times 500$ | – | – | ✓ |
| Kvasir-SEG [19] | Clinical | Colon | PS | 1000 | Mixed | – | – | ✓ |
| UCL Depth [40] | Virtual | Colon | DE | 16016 | $256 \times 256$ | DDM | CT | – |
| Zhang et al. [68] | Virtual | Colon | DE / 3DR | Seqs. | – | DDM | CT | – |
| VR-Caps [18] | Virtual | Colon | DE / 3DR | Seqs. | – | DDM | CT | – |
| Yang et al.[62] | Virtual | Colon | DE | 4500 | – | DDM | CT | – |
| EndoSLAM [39] | Physical | Colon+ | Disp. Est. | 42700+ | Mixed | SPC | 3D Scanner | – |
| C3VD [6] | Physical | Colon | DE / 3DR | 10015 | – | DDM | 3D Model | – |
| EndoMapper [1] | Clinical | Colon | 3DR | 59+ Seqs. | $320 \times 240$ | SPC | SLAM | – |
| LDPolypVideo [34] | Clinical | Colon | PS | 33k / 40k | $560 \times 480$ | – | – | ✓ |
| SUN-SEG [21] | Clinical | Colon | PS | 49k / 158k | $1158 \times 1008$ | – | – | ✓ |
| **PolypSense3D** | **Mixed** | **Colon** | **PS/DE/3DR** | **16k+ / 43k+** | **Mixed** | **Mixed** | **Mixed** | ✓ |

### 2.1 2D Polyp Segmentation Datasets and Methods

Significant progress in 2D endoscopic polyp segmentation has been enabled by high-quality public datasets such as Kvasir-SEG [19], CVC-ClinicDB [4], CVC-ColonDB [51], ETIS-Larib [48], and the ASU-Mayo Clinic dataset [3]. These resources have encouraged the development of sophisticated algorithms, initially dominated by Convolutional Neural Network (CNN) based encoder-decoder architectures like U-Net and its variants [44, 7, 72], which excel at capturing fine-grained spatial details. Innovations include attention mechanisms and specialized modules focusing on boundary refinement or feature enhancement [14, 2, 64, 55, 56, 46]. More recently, Transformer-based architectures [13, 50, 59, 66, 67] and large-scale pre-trained models like the Segment Anything Model (SAM) [23] and its adaptations [11, 26] have demonstrated strong capabilities, particularly in handling global context and challenging cases like small or occluded polyps [22].

Despite impressive gains in segmentation accuracy, existing 2D datasets suffer from a critical limitation for clinical metrology: they **lack associated depth or scale information**. Segmentation masks are purely planar representations (pixel coordinates) without any link to the polyp's real-world physical dimensions. This fundamental **absence of calibrated spatial context and ground-truth size annotations** prevents the direct use of these datasets for developing or validating algorithms aimed at quantitative polyp measurement, necessitating datasets that integrate these missing components.

### 2.2 Endoscopic 3D Perception Datasets and Methods

Endoscopic 3D perception techniques, including depth estimation and 3D reconstruction, aim to recover the spatial structure of the colon [41, 63]. Resources supporting this research include: **Virtual synthetic datasets** [40, 68, 18], often generated from CT models, provide paired RGB-depth data but may lack photorealism, exhibit non-physiological scaling, or omit realistic lesion modeling. **Physical phantom datasets** [39, 6] use 3D-printed models to offer greater realism but are often limited to

sparse depth annotations due to sensor capabilities. **Clinical datasets** [1] provide authenticity but typically lack ground-truth depth maps due to intraoperative constraints.

Methodologically, traditional geometric approaches like Structure-from-Motion (SfM) and SLAM [35, 49, 15, 24] struggle with the texture-poor and deformable nature of endoscopic scenes, often yielding sparse, scale-ambiguous reconstructions. Learning-based methods have shown promise, leveraging supervised training on synthetic data [33, 42, 38, 29, 62] or self-supervised signals like photometric consistency from video [31, 69, 12, 47, 43, 30, 17]. However, supervised methods face the synthetic-to-real domain gap, while self-supervised methods can be sensitive to illumination changes and lack metric scale.

Crucially, both existing 3D perception paradigms and datasets primarily focus on global scene reconstruction or dense depth estimation across the entire view, rather than providing the specific tools needed for accurate, lesion-centric metrology. They generally lack datasets containing numerous, well-defined polyp instances that are simultaneously annotated with **both precise 3D location/structure and verified ground-truth physical size**. This disconnect prevents the direct evaluation of algorithms on the clinically vital task of polyp measurement. PolypSense3D is specifically designed to fill this gap by providing multi-source data with jointly annotated segmentation, depth, camera parameters, and critically, ground-truth millimeter-scale polyp sizes.

# 3 PolypSense3D Dataset Construction

To create a robust benchmark for depth-aware polyp size measurement, PolypSense3D integrates data from three distinct yet complementary sources. Each source utilizes a tailored pipeline, detailed below, designed to maximize fidelity while addressing specific research needs (e.g., controlled ground truth vs. real-world complexity). Exhaustive protocols, parameter details, and specific validation studies (e.g., for the clinical annotation protocol in Appendix D) for each source are provided in Appendix A (Virtual), Appendix B (Physical), and Appendix C (Clinical).

## 3.1 Virtual Simulation Dataset: Controlled Environment with Dense GT

By operating on the virtual camera of simulator, we built a high-fidelity virtual colon environment with Polyps in Unity based on Vr-Caps [18]. This allows for precise ground truth and systematic variation. Using anonymized patient CT scans from The Cancer Imaging Archive (TCIA) [1], we reconstructed anatomical models via InVesalius [2], refined them in Blender for anatomical accuracy, and accurately scaled them to realistic human dimensions (validation in Appendix A.2.1). We then designed and procedurally textured 30 distinct polyp models, varying shape, size (1.79-20.52mm), and morphology (details in Appendix A.2.2), embedding them in clinically relevant locations (ascending, transverse, sigmoid colon, near folds, bends) with varied orientations. These virtual polyp models provide a solid and diverse foundation for validation. It is important to note that this dataset is not restrictive, and our modeling pipeline will be open-sourced, allowing researchers to customize polyp variants according to their specific needs.

A custom physics-based controller simulated realistic endoscope dynamics (forward/backward translation, pitch/yaw rotation), mimicking capsule endoscopy movement patterns (controller details in Appendix A.3). Adjustable lighting (intensity, cone angle, position) and camera parameters (FOV set to 78.4°, typical focal length 200mm equivalent) were tuned to approximate clinical conditions while avoiding rendering artifacts (Appendix A.4). Using Unity Recorder, we captured over 32,000 synchronized frames including: (1) High-resolution RGB images with polyps; (2) Dense, per-pixel metric depth maps (16-bit PNG, derived from geometry buffer); (3) Calibrated camera intrinsics per frame; (4) 6DoF camera pose trajectory logs; (5) Automatically generated pixel-perfect polyp segmentation masks (derived from model geometry), subsequently reviewed and confirmed by experienced clinical endoscopists. This yields a rich dataset ideal for supervised training and controlled evaluation of segmentation, depth, and size estimation.

---

[1] https://www.cancerimagingarchive.net
[2] https://invesalius.github.io/

## 3.2 Physical Phantom Dataset: Bridging the Reality Gap

To introduce real-world textures, lighting, and sensor characteristics, we created a physical benchmark. A 50cm section of the transverse colon was 3D printed using White Resin material on a UnionTech SLA lite600 3D printer (further design details in Appendix B.1), based on CT data [39]. This phantom features an interlocking modular design for assembly and a semi-open structure for external visibility during experiments. 13 solid polyps (diameters 4.00-14.89 mm) were physically embedded at known locations. We acquired video data using a commercially available small-diameter veterinary endoscope connected via USB (eCap software), manually navigating the lumen.

Crucially, we performed meticulous camera calibration *prior* to data acquisition using Zhang's method [70] implemented in MATLAB. A $4 \times 5 \times 3$ mm checkerboard was determined optimal after evaluating multiple sizes and distances (details in Appendix B.2). Calibration was performed at a working distance of 22mm, capturing views across all image quadrants. The resulting intrinsic parameters (see Appendix B.2 for final parameters used) were used for all subsequent image undistortion and metric calculations. This physical dataset (13 videos) provides realistic appearance data paired with known polyp sizes and locations, ideal for evaluating sim-to-real transfer and validating scale estimation from camera intrinsics.

## 3.3 Real Clinical Dataset: In-Vivo Complexity and Sparse Cues

To provide data from the target environment, we collected videos during routine colonoscopies using standard clinical equipment, specifically an Olympus EVIS EXERA III endoscopic system with an Olympus CF-H290I colonoscope, adhering to approved institutional protocols (IRB approval details in Section 8). The clinical endoscope was calibrated using the same rigorous procedure as the phantom endoscope (details in Appendix B.2). We extracted stable video segments ("freeze-frames") showing detected polyps during inspection.

Obtaining dense, metric ground truth in-vivo is infeasible; therefore, we developed and rigorously applied a novel **biopsy-forceps-assisted annotation protocol**, whose validation regarding reproducibility and inter-operator variability is detailed in Appendix D:

1. **Reference Frame Selection & Segmentation:** To minimize the effects of camera motion and intestinal peristalsis, stable and clear frames are first selected by trained annotators under the supervision of experienced endoscopists. Segmentation masks are then manually refined with the assistance of prompt-based models such as SAM [23], ensuring accurate and consistent delineation of polyp boundaries.

2. **Size Annotation (Proximal Comparison):** In designated frames captured by the physician, fully opened biopsy forceps (calibrated 5mm physical tip-to-tip distance) are positioned adjacent to the polyp, aiming for the same apparent depth plane. Annotators precisely mark the forceps tips (illustrated in Appendix C.3). The pixel distance $d_{pixel}$ yields a local scale factor: $scale = 5mm/d_{pixels}$ (mm/pixel). This scale factor is applied to the refined segmentation mask's dimensions (e.g., major axis $L_{pixels}^{major}$) to compute the polyp's physical size (e.g., $L_{mm}^{major} = L_{pixels}^{major} \times scale$). This method grounds the measurement in a known physical reference visible within the frame.

3. **Sparse Depth Annotation (Extension Contact):** In separate stable frames, forceps with clear 5mm gradation markings are used. The physician extends the forceps so the tip gently contacts the polyp apex or adjacent flat mucosa. The annotator measures the visible pixel length of the extended marked portion of the forceps. Using the known physical length per gradation (5mm), calibrated camera intrinsics, and a carefully validated perspective rectification and regression model (detailed in Appendix C.3), the metric depth $Z$ of that contact point is calculated. This provides sparse but metrically accurate depth points on or near the lesion.

Quality control involved cross-checking annotations between multiple trained annotators and physician review of selected cases (details in Appendix C.3 and Appendix D). While this protocol introduces potential operator variability and provides only sparse depth, it offers the first feasible method, to our knowledge, for obtaining quantitative size and partial depth labels directly from standard in-vivo colonoscopy videos, creating a uniquely challenging and realistic dataset reflecting true clinical conditions, particularly for smaller polyps prevalent in screening.

Table 2: Summary Statistics of the PolypSense3D Dataset. See Appendix E for detailed breakdown.

| Metric | Virtual Simulation | Physical Phantom | Clinical Data | Total / Overall |
|---|---|---|---|---|
| Total Frames (approx.) | ~32k | ~13 videos | ~11k | ~43k+ |
| Frames with Annotated Polyps (approx.) | ~16k (50%) | ~74 (cut from 13 videos) | ~438 | ~16k |
| Unique Polyp Instances | 30 (Models) | 13 (Embedded) | 53 (from 127 Patients) | 114 |
| Size Range (mm, approx. diameter) | 1.79–20.52 | 4.00–14.89 | 1.39–9.98 | 1.39–32.8 |
| Size Distribution (<10 / 10–20 / >20 mm) | 50.0% / 43.3% / 6.7% | 69.2% / 30.8% / 0.0% | 100% / 0.0% / 0.0% | 80.2% / 17.71% / 2.1% |
| Depth Annotation Type | Dense (GT) | - | Sparse (Forceps GT) | Mixed |
| Resolution(s) | 320×320(Scalable) | 640×480 | 1157×1006 | Multiple |
| Camera Intrinsics Provided | Yes (Full) | Yes (Calibrated) | Yes (Calibrated) | Yes |

# 4  Dataset Analysis and Statistics

PolypSense3D offers a substantial and diverse resource, comprising over 43,000 frames across three progressively challenging subsets, synthetic, physical, and clinical subsets. These subsets are designed to reflect increasing levels of real-world complexity. The synthetic subset provides dense, precise annotations under idealized conditions but lacks realistic textures and lighting variability. The physical dataset, built from 3D-printed models, introduces real and accurate geometry while remaining limited in dynamic content and texture richness. The clinical subset poses the greatest challenge, featuring uncontrolled endoscope motion, variable illumination, specular highlights, complex mucosal textures, tool occlusions, and motion blur from real procedures.

A distinctive feature of the clinical data is its multimodal richness: a single polyp may correspond to multiple frames annotated with RGB images, depth maps, and segmentation masks. In total, nearly 16,000 polyp-containing frames are annotated, enabling fine-grained evaluation of detection, measurement, and spatial reasoning algorithms. Table 2 provides a high-level overview, and detailed statistics on polyp characteristics, imaging resolutions, and camera parameters are included in Appendix E.

# 5  Benchmark Experiments and Evaluation

We conducted experiments using PolypSense3D to establish baselines for depth-aware polyp sizing. Our evaluation aims to: (1) assess the performance of state-of-the-art models on each dataset component, (2) evaluate model robustness across the three progressively challenging subsets, synthetic (idealized with dense annotations), physical (real geometry with static but limited texture), and clinical (highly realistic with motion, lighting, and occlusion challenges), and (3) analyze how upstream errors in segmentation and depth estimation propagate to final size measurements.

## 5.1  Experimental Setup

**Tasks & Metrics:** (1) *2D Polyp Segmentation* (mDice, mIoU, Recall, Precision, F1 Score); (2) *Monocular Depth Estimation* (RMSE($10 \times e - 3$), AbsRel, Log10, $\delta < 1.25^x$); (3) *End-to-End Polyp Size Estimation* (MAE mm), using outputs from (1) and (2). For key mean metrics, standard deviations are reported in Appendix G to indicate result variability. **Baseline Models:** Segmentation: MSNet[28], PraNet [14], SAM [23], MobileSAM [65], VM-UNet [45]. Depth: DAM-V1 [60], DAM-V2 [61], ZoeDepth [5], DAM V2-mini [61]. Rationale included SOTA status and diverse architectures. **Implementation:** Models trained/fine-tuned on our training split were evaluated across all test splits (7:3 ratio). These models were all trained on our Virtual Simulation Dataset. Details regarding training procedures, hyperparameters, and computational resources are provided in Appendix F. The size estimation pipeline is detailed in Appendix G.1.

## 5.2  Benchmark Results

We measure polyp size in metric scale by combining contour segmentation and absolute depth estimation (see Appendix G.1 for details). Comprehensive evaluations are conducted on all three proposed datasets for: (1) polyp segmentation, (2) depth estimation, and (3) integrated size measurement. Full per-polyp results and extended error analyses are available in Appendix G.

---

[3]`http://github.com/RubyQianru/Depth-Anything-V2-Mini`

**Virtual Simulation Evaluation:** On the virtual test set, segmentation models exhibited strong performance (Table 3). Among them, Sam2-unet achieved the highest mDice of 0.9399, indicating excellent adaptability to synthetic endoscopic imagery. MSNet and PraNet followed closely. MobileSAM and VM-UNet yielded lower scores, likely due to MobileSAM's lightweight design optimized for speed, and VM-UNet's shallow architecture with limited representational capacity. Depth estimation models, trained on densely labelled data, have achieved remarkably strong performance (Table 10), with DAM V2-mini showing the lowest RMSE (0.185e-03). This level of accuracy in depth estimation holds significant promise for substantially enhancing polyp size measurement. Combining predicted segmentation (MSNet) and ground truth depth yielded an average size MAE of 0.43mm (Table 6), while using ground truth segmentation and predicted depth resulted in a higher MAE of 4.95mm, quantifying error propagation even in this controlled setting. Detailed per-polyp results are in Table 16 (Appendix G).

**Physical Phantom Evaluation:** Applying models to the physical phantom revealed a clear sim-to-real gap. Compared to the synthetic dataset, the lack of realistic texture in the 3D-printed environment led to a general drop in performance across tasks. As shown in Table 3, segmentation accuracy decreased for all models, though polyp-specific methods such as MSNet and Sam2-unet still maintained satisfactory results. These findings align with the qualitative visualizations provided in Appendix G.2. Depth estimation was particularly affected by the inability to acquire dense, full-frame ground-truth depth on this platform, limiting both training and evaluation to qualitative visualizations (also in Appendix G.2). Compared to the synthetic domain, estimated depth maps on physical data exhibited degraded

Table 3: Segmentation Performance on Unity, Physical, and Clinical Datasets

| Dataset | Method | mDice | mIoU | Recall | Precision | F1 Score |
|---|---|---|---|---|---|---|
| Unity | MSNet[28] | 0.9301 | 0.9000 | **0.9816** | 0.9169 | 0.9412 |
| | PraNet [14] | 0.9062 | 0.8525 | 0.9733 | 0.8676 | 0.9062 |
| | SAM [23] | 0.8803 | 0.8180 | 0.8552 | **0.9418** | 0.8803 |
| | MobileSAM [65] | 0.8341 | 0.7054 | 0.8776 | 0.8334 | 0.8341 |
| | VM-Unet [45] | 0.8432 | 0.7288 | 0.9172 | 0.8044 | 0.8432 |
| | ASPS [25] | 0.8951 | 0.8321 | 0.9409 | 0.8717 | 0.8951 |
| | Sam2-unet [57] | **0.9399** | **0.9010** | 0.9723 | 0.9243 | **0.9428** |
| Physical | MSNet[28] | 0.8812 | 0.8432 | 0.9425 | 0.8457 | 0.8847 |
| | PraNet [14] | 0.8523 | 0.8028 | 0.9106 | 0.8138 | 0.8523 |
| | SAM [23] | 0.8693 | 0.8218 | 0.9603 | 0.8351 | 0.8693 |
| | MobileSAM [65] | 0.6874 | 0.5091 | 0.6304 | **0.9887** | 0.6874 |
| | VM-Unet [45] | 0.2546 | 0.1459 | 0.3631 | 0.1960 | 0.2546 |
| | ASPS [25] | 0.7526 | 0.6834 | 0.7599 | 0.7649 | 0.7540 |
| | Sam2-unet [57] | **0.8983** | **0.8501** | **0.9981** | 0.8513 | **0.9064** |
| Clinical | MSNet[28] | 0.6989 | 0.6250 | 0.7441 | 0.7201 | 0.7068 |
| | PraNet [14] | 0.7183 | 0.6316 | 0.7082 | 0.7999 | 0.7183 |
| | SAM [23] | **0.7992** | **0.7032** | 0.8569 | 0.7978 | **0.7992** |
| | MobileSAM [65] | 0.2728 | 0.1437 | 0.2131 | **0.9586** | 0.2728 |
| | VM-Unet [45] | 0.0634 | 0.0328 | 0.0780 | 0.0526 | 0.0634 |
| | ASPS [25] | 0.5179 | 0.4239 | 0.5574 | 0.5586 | 0.5183 |
| | Sam2-unet [57] | 0.7346 | 0.6665 | **0.8720** | 0.7208 | 0.7524 |

detail and smoothness. These limitations in segmentation and depth estimation propagated to polyp size measurement. As shown in Table 7, size estimation errors on the phantom were notably higher than in the virtual setting, with all methods yielding an Abs.Error above 1mm. This underscores the challenge of domain transfer from ideal simulation to physical systems with realistic geometry but limited texture and sensing variation. A per-polyp breakdown is provided in Table 17 (Appendix G).

**Clinical Data Evaluation:** The clinical test set represents the most challenging scenario due to in vivo factors such as variable lighting, fluid interference, tissue motion, and camera shake. These conditions significantly degrade performance, as evidenced by further drops in segmentation accuracy (e.g., MSNet mDice 0.6989, Sam2-unet mDice 0.7346; Table 3). Dense ground-truth depth is unavailable; instead, we

Table 4: Benchmark on Unity Dataset: Depth Estimation Performance

| Method | Error Metrics ↓ | | | Accuracy Metrics↑ | |
|---|---|---|---|---|---|
| | RMSE | abs. REL | Log10 | $\delta_1$ | $\delta_2$ |
| DAM V1 [60] | 0.242 | 0.020 | 0.009 | **0.998** | **0.999** |
| DAM V2 [61] | 0.216 | 0.015 | 0.006 | 0.996 | **0.999** |
| ZoeDepth [5] | 0.213 | 0.015 | 0.006 | **0.998** | 0.999 |
| DAM V2-mini [61] | **0.185** | **0.011** | **0.005** | 0.997 | 0.999 |

employed biopsy forceps contact points for sparse but reliable supervision (Appendix C.3.2). These annotations confirmed a notable decline in depth accuracy compared to the synthetic dataset, which features dense depth supervision and near-ideal imaging conditions (Table 5). Despite increased segmentation and depth errors, automated size estimation remained competitive. The average MAE reached 0.95mm, outperforming physician visual estimates (1.84mm Table 8). This highlights the potential of algorithmic measurements to assist clinical polyp assessment, even under realistic constraints. Per-case results are provided in Table 18 (Appendix G).

**Performance Across Multi-source Datasets**

Performance consistently declined from the Unity to the Physical and Clinical datasets, reflecting increasing real-world complexity. The Unity subset, built with 3D-modeled polyps in a clean, artifact-free environment, enables precise ground-truth generation. The Physical set

Table 5: Benchmark on Clinical Dataset: Depth Estimation Performance

| Method | Error Metrics ↓ | | | | Accuracy Metrics↑ | |
|---|---|---|---|---|---|---|
| | RMSE | abs.REL | Log10 | MAE | $\delta_1$ | $\delta_2$ |
| DAM V1 [60] | 10.983 | 0.428 | 0.249 | 8.348 | 0.098 | 0.390 |
| DAM V2 [61] | **8.089** | 0.383 | **0.152** | **5.757** | **0.463** | **0.732** |
| ZoeDepth [5] | 12.481 | 0.491 | 0.319 | 9.784 | 0.122 | 0.268 |
| DAM V2-mini [61] | 8.801 | **0.359** | 0.162 | 6.069 | 0.415 | 0.659 |

adds real imaging and geometry via 3D-printed
models but lacks texture diversity. The Clinical
set poses the greatest challenge, with motion blur, lighting variation, tool occlusions, and complex tissue textures. These results highlight the need for segmentation models with stronger cross-domain robustness.

Table 6: Benchmark on Unity Dataset: Per-Polyp Size Estimation (mm). Full table in Appendix G (Table 16).

| Polyp ID | Label (Forceps) | Seg(GT)+Depth(Pred) | | Seg(Pred)+Depth(Pred) | | Doctor Visual Est.(Avg) | |
|---|---|---|---|---|---|---|---|
| | | Predicted | Abs. Error | Predicted | Abs. Error | Predicted | Abs. Error |
| P1 | 11.55 | 13.46 | 1.92 | 19.06 | 7.51 | 9.26 | 2.29 |
| ... | ... | ... | ... | ... | ... | | |
| P30 | 10.04 | 9.48 | 0.56 | 16.48 | 6.44 | 7.55 | 2.49 |
| **Average** | **9.57** | **9.27** | **4.34** | **14.42** | **7.52** | **7.26** | **4.88** |

Table 7: Benchmark on 3D-Printed Dataset: Per-Polyp Size Estimation (mm). Full table in Appendix G (Table 17).

| Polyp ID | Label | Seg(GT)+Depth(Pred) | | Seg(Pred)+Depth(Pred) | |
|---|---|---|---|---|---|
| | | Predicted | Abs. Error | Predicted | Abs. Error |
| P1 | 6.20 | 6.22 | 0.02 | 8.46 | 2.26 |
| P13 | 4.94 | 6.74 | 1.08 | 8.89 | 3.85 |
| **Average** | **8.03** | **8.38** | **1.48** | **9.48** | **1.99** |

Table 8: Benchmark on Clinical Dataset: Per-Case Size Estimation (mm). Full table in Appendix G (Table 18).

| Polyp ID | Label (Forceps) | Seg(GT)+Depth(Pred) | | Seg(Pred)+Depth(Pred) | | Doctor Visual Est.(Avg) | |
|---|---|---|---|---|---|---|---|
| | | Predicted | Abs. Error | Predicted | Abs. Error | Predicted | Abs. Error |
| P1 | 3.28 | 2.17 | 1.11 | 1.73 | 1.55 | 4.20 | 0.92 |
| ... | ... | ... | ... | ... | ... | ... | ... |
| P53 | 5.95 | 5.99 | 0.04 | 4.22 | 1.73 | 8.67 | 2.72 |
| **Average** | **3.28** | **2.80** | **1.19** | **2.66** | **0.95** | **4.73** | **1.84** |

# 6 Discussion

PolypSense3D provides a comprehensive benchmark for depth-aware polyp size estimation from two key perspectives. First, it enables systematic evaluation of segmentation and depth estimation performance across three increasingly challenging datasets: synthetic (ideal and richly annotated), physical (real imaging with limited texture), and clinical (dynamic, noisy, and highly realistic). This reveals clear performance degradation, especially when models trained on virtual data are applied to clinical scenarios. Second, the dataset allows analysis of how upstream perception errors propagate to downstream size estimation. Despite these errors, automated measurements remain competitive with physician estimates, underscoring the clinical value of robust, end-to-end methods under real-world conditions.

**Limitations and Biases:** While PolypSense3D offers unique advantages for benchmarking, certain limitations must be acknowledged. Virtual simulations, despite efforts towards realism, cannot fully replicate complex tissue biomechanics or intricate light-tissue interactions. Physical phantoms utilize simplified materials and lack the dynamic nature of living tissue. Our novel clinical annotation method, while providing invaluable in-vivo quantitative labels currently unavailable elsewhere, relies on physician interaction during the procedure and yields only sparse depth information. This introduces potential operator variability and inherent precision limits tied to operator skill and strict adherence to the protocol; a detailed validation of this protocol, including inter-operator agreement and accuracy assessments, is provided in Appendix D. We have implemented measures like standardized training and cross-checks to mitigate this variability, but it remains a factor reflecting real-world measurement challenges. In annotating and predicting polyp size, our estimation method is limited by the biopsy forceps' angle of approach, position, and the perspective of the polyp images, which may lead to certain inaccuracies. Additionally, the clinical data also originates from a specific, single-center institutional context, reflecting its patient demographics and endoscopic equipment, which may introduce sampling bias and limit immediate generalizability without adaptation (ethical considerations in Section 8). Users must carefully consider these factors when designing experiments and interpreting results obtained using PolypSense3D.

**Future Research Directions Enabled by PolypSense3D:** This benchmark dataset is designed to catalyze progress in several key areas of machine learning relevant to the NeurIPS community:

- **Domain Adaptation and Generalization:** The pronounced domain gap between PolypSense3D's virtual, physical, and clinical subsets provides a challenging, realistic testbed for evaluating and developing novel unsupervised and semi-supervised domain adaptation techniques, with ongoing efforts to expand multi-center, cross-device data benchmark collection. This is crucial for transferring models trained on readily available synthetic or controlled data to real-world clinical settings where labeled data is scarce.
- **Learning with Sparse and Imperfect Labels:** The clinical subset, with its sparse depth annotations derived via the forceps protocol (validation in Appendix D) and associated measurement variability, directly facilitates research into methods robust to sparse, potentially noisy, or weakly supervised labels. This includes areas like few-shot depth completion, uncertainty-aware learning, incorporating geometric priors, or learning robust representations that are less sensitive to label noise. At the same time, these sparse depth measurements can effectively address the inherent scale ambiguity in self-supervised methods, serving as constraints to assist the self-supervised sparse-to-dense depth completion framework with ground truth (GT) size and depth annotations[32]. Additionally, sparse measurement points can act as strong cues to guide or fine-tune large foundational models[27], enabling them to adapt to the clinical domain and recover absolute scale.
- **End-to-End Quantitative Perception:** By providing synchronized multi-modal data including ground-truth size, PolypSense3D enables the direct training and evaluation of end-to-end networks that predict polyp real-scale size from endoscopic images or video snippets[58]. Such approaches could potentially bypass cascaded segmentation and depth estimation steps, mitigating cumulative error propagation.
- **Leveraging Temporal Information:** Although presented frame-wise, the underlying video sequences (especially for clinical and phantom data) open avenues for exploring temporal consistency models. Utilizing information across frames could lead to more robust and temporally stable depth and size estimations, drawing on techniques from video understanding and time-series analysis[8].
- **Uncertainty Quantification for Clinical Trust:** The multi-source nature, inclusion of challenging clinical data, and availability of ground-truth allow for benchmarking methods that quantify prediction uncertainty[71]. Reliable uncertainty estimates for automated size measurements are essential for assessing model trustworthiness and facilitating safer clinical integration.
- **Multi-Task Learning and Synergies:** The dataset naturally supports investigating multi-task learning frameworks that jointly optimize for segmentation, depth estimation, and size prediction, potentially uncovering synergistic benefits between these related tasks.

PolypSense3D provides the necessary data diversity and annotations to rigorously pursue these and other related ML research questions in quantitative medical vision.

## 7 Conclusion

Accurate polyp size measurement is a critical unmet need in clinical endoscopy. We introduced PolypSense3D, the first multi-source benchmark dataset specifically designed to facilitate research and evaluation in depth-aware polyp size estimation. By combining virtual, physical, and clinically annotated data with synchronized multi-modal information (RGB, depth, segmentation, size, camera parameters), PolypSense3D provides an essential resource currently missing in the field. Our comprehensive benchmark experiments establish strong baselines, quantify the performance challenges faced by current SOTA models, particularly regarding domain generalization and error propagation, and confirm the dataset's value for evaluating the complete perception-to-measurement pipeline. We release PolypSense3D publicly with code and protocols, aiming to standardize evaluation and accelerate the development of reliable automated tools for quantitative endoscopic analysis, ultimately benefiting patient care.

## 8 Ethical Considerations

The intestinal models utilized for both our virtual synthetic dataset and physical phantom in this study were derived from publicly available medical data. The collection of clinical data was conducted by experienced medical professionals at Sir Run Run Shaw Hospital, Zhejiang University School of

Medicine, and received approval from the hospital's ethics committee. Throughout the clinical data acquisition process, we strictly adhered to relevant regulations, ensuring the anonymization of all data to comprehensively protect patient privacy and personal information. It is crucial to emphasize that this dataset is intended to advance research in the field of in-vivo spatial intelligence perception based on endoscopic images. Its purpose is not for direct clinical diagnosis or medical decision-making, and any clinical application based on this dataset should undergo thorough evaluation and supervision by qualified medical professionals.

## 9 Acknowledgements

We would like to express our gratitude to Chenyu Yan, Yihao Ying, Tianyu Zhao, Yuanyuan Zhang, Gaoqi Huang, Yibo Wang, and Peng Lu from Tianjin University of Technology, Hangzhou Normal University for their contributions and efforts in the experimental annotation work. The work is supported by the Marie Skłodowska-Curie Postdoctoral Individual Fellowship under Grant No. 101154277, National Natural Science Foundation of China under Grant 62202137, 62306097, Zhejiang Provincial Natural Science Foundation of China under Grant LMS25F020009.

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

# Appendix

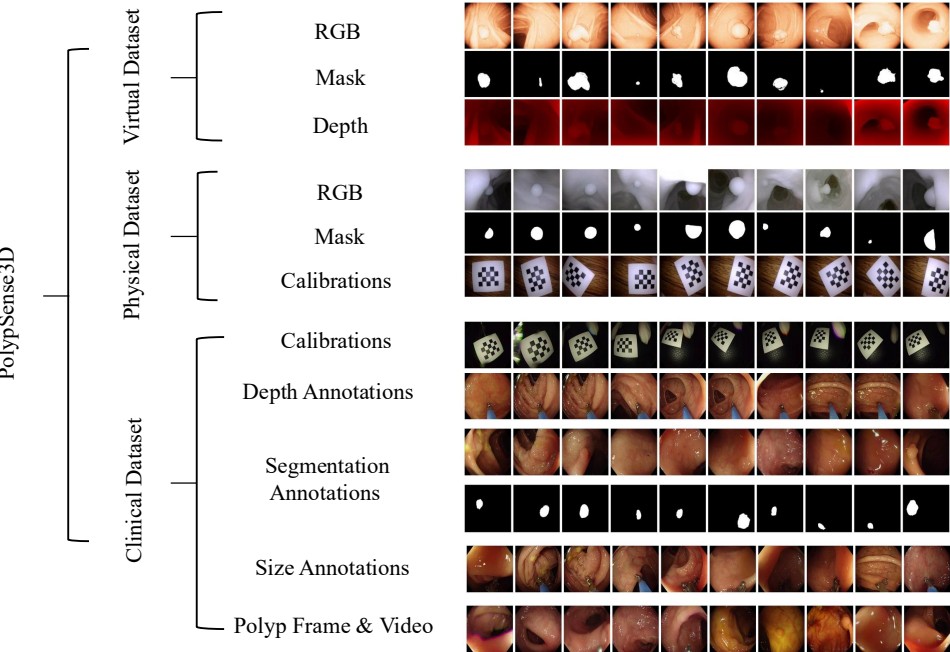

Figure 1: The PolypSense3D dataset architecture, depicting its multi-source composition (Virtual, Physical, and Clinical datasets) and the associated data components: RGB images, segmentation masks, depth information, calibration parameters, and size annotations, organized within a polyp frame and video structure.

## A  Virtual Simulation Dataset Collection and Annotation

### A.1  Background and Motivation

This appendix provides a detailed extension of Section 3.1 in the main text, which introduces the construction of the virtual synthetic dataset in PolypSense3D. While the main text outlines the overall framework, this supplementary material elaborates on the technical implementation, including anatomical model scaling, virtual endoscope motion control, multimodal data acquisition, and the generation of accurate polyp segmentation masks. Together, these details ensure the virtual dataset meets the specific spatial and annotation requirements for developing and validating depth-aware polyp measurement algorithms.

### A.2  Colon and Polyp Modeling with Scale Standardization

#### A.2.1  Colon Modeling and Dimension Normalization

We began by analyzing anatomical references [9] to define realistic colon segment lengths: ascending colon (12-20cm), transverse colon (40-50cm), descending colon (25cm), and sigmoid colon (40cm), as illustrated in Figure 2. The average diameter of the adult colon ranges from 6 to 9cm, with a total length of approximately 150cm.

Although existing platforms like VR-Caps [18] offer Unity-based environments, their original models often lack proper physical scale. To correct this, we imported DICOM data from The Cancer Imaging Archive (TCIA) [4] into InVesalius [5] to reconstruct a colon model (original dimensions shown in Figure 4-Left). The extracted unrealistic size (e.g., 24,860m × 38,841m × 19,542m) was normalized

---

[4] https://www.cancerimagingarchive.net
[5] https://invesalius.github.io/

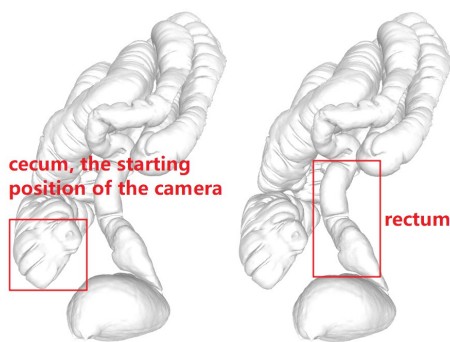

Figure 2: 3D colon model highlighting the anatomical positions of the cecum and rectum.

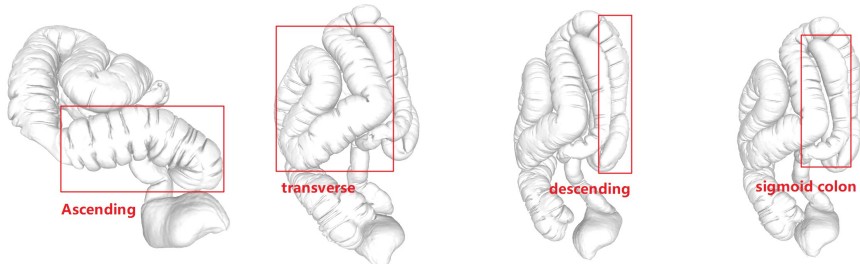

Figure 3: Segmented anatomical regions of the colon model: ascending, transverse, descending, and sigmoid colon.

by downscaling (e.g., $100\times$) and converting to Unity units (1:1000), yielding realistic dimensions (e.g., $0.249m \times 0.388m \times 0.195m$). We similarly rescaled reference models to ensure consistency with real-world anatomical standards (Figure 4-Right).

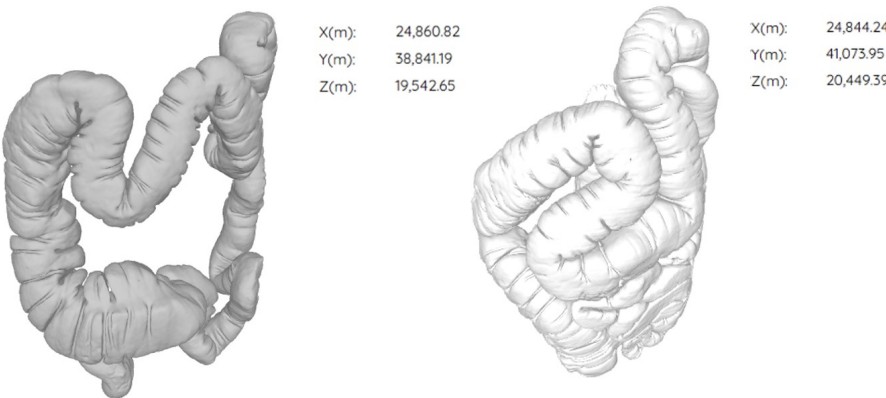

Figure 4: **Left:** Raw model dimensions of TCIA colon data in InVesalius. **Right:** Original reference model dimensions before scale normalization.

To rigorously validate the accuracy of the rescaled model, we applied three independent scale verification methods:

**(1) Diameter validation:** We extracted the internal cross-sectional diameter at multiple representative sections using InVesalius's measuring tools (Figure 5-Left). The average diameter measured approximately 58mm (5.8cm), falling within the accepted clinical range (6-9cm), confirming realistic transverse profiles.

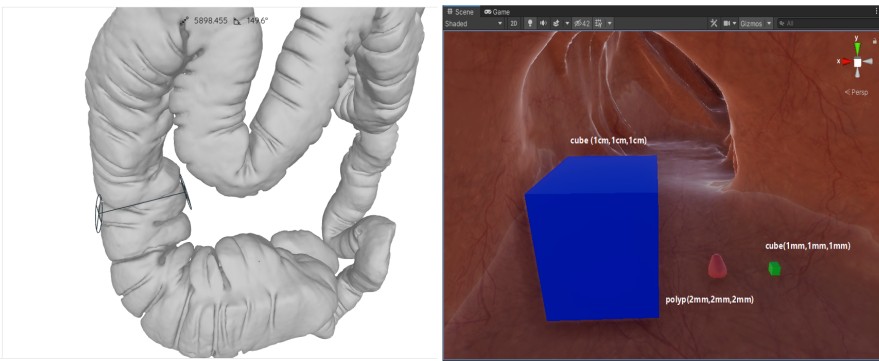

Figure 5: **Left:** Pre-normalization colon lumen diameter measured (e.g., 58mm after scaling) in InVesalius. **Right:** Scale validation using embedded reference objects (1cm$^3$ cube, 2mm polyp, 1mm$^3$ cube) in Unity.

**(2) Reference object embedding:** We inserted standardized reference geometries (1cm$^3$ cube, 2mm polyp, 1mm$^3$ cube) into the Unity colon model (Figure 5-Right). Visual comparison confirmed their proportions were consistent with clinical expectations within the simulated view.

**(3) Total length measurement:** We measured the full virtual colon length along its central path using a spline tool (Figure 6). The total virtual length (approx. 852mm) was consistent with expected anatomical lengths, accounting for modeled segments. This verified longitudinal scaling.

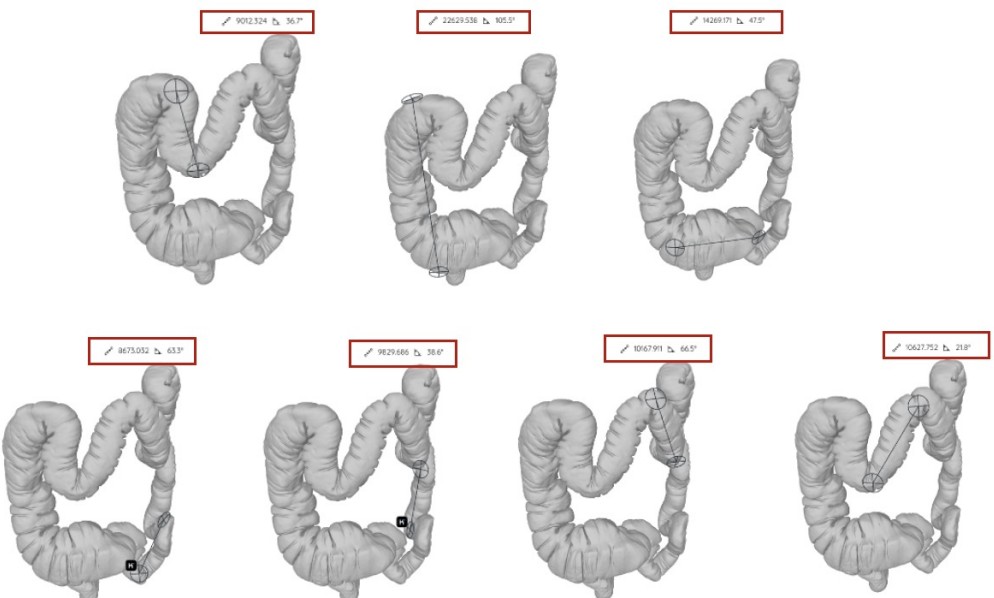

Figure 6: Illustration of total colon length measurement along a curved path using an InVesalius spline tool (or equivalent in Blender/Unity).

### A.2.2 Polyp Modeling and Scaling

We designed 30 polyp models in Blender, covering diverse morphologies and sizes, categorized based on clinical prevalence (<10 mm small, 10-20 mm medium, >20 mm large). Models were textured to match colon surfaces and exported as '.fbx'. Unity's import scaling (0.01 factor) was accounted for, and final in-scene sizes were manually fine-tuned to match target millimeter dimensions (Table 16). Polyps were strategically distributed across all major colon segments in clinically relevant positions

(near folds, flexures) with slight variations in orientation and size (Figure 7) to enhance dataset diversity and model robustness training.

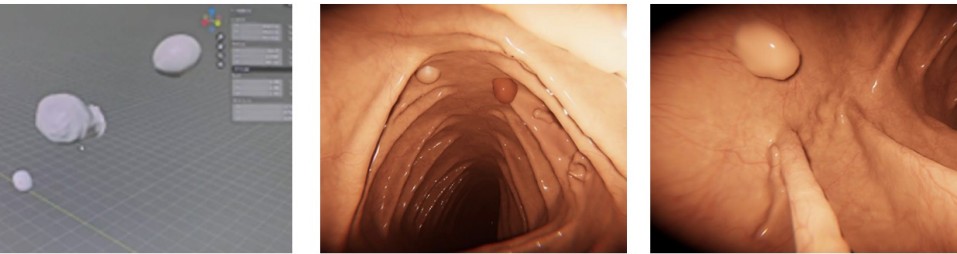

Figure 7: Polyp modeling in Blender (example) and spatial placement within the scaled colon environment in Unity.

This detailed modeling ensures a spatially grounded virtual dataset suitable for benchmarking algorithms aimed at metric measurement.

### A.3 Motion Scale Normalization and Navigation Control

Following model standardization, large-scale data acquisition commenced using a simulated capsule endoscope in Unity, recording synchronized RGB, depth, and pose data aligned to real-world metric scales.

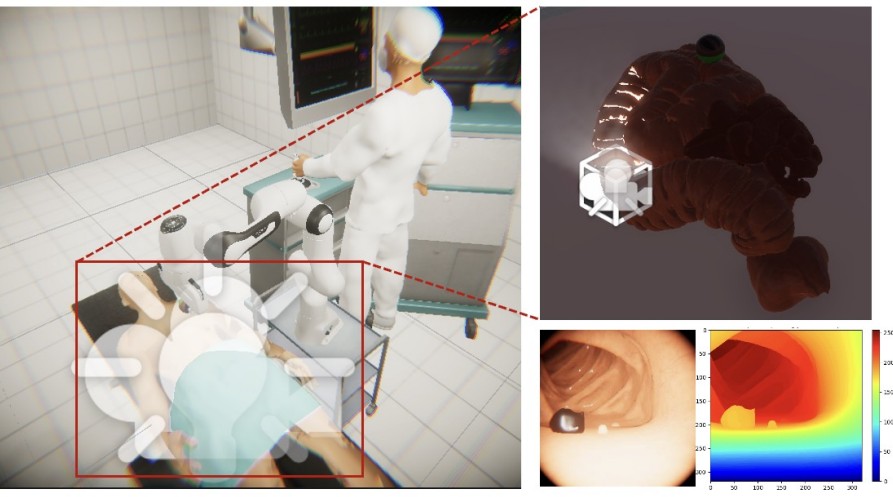

Figure 8: VR-Caps based multimodal data acquisition setup showing synchronized capture pipeline.

Realistic movement was achieved by tuning Unity's physics engine ($FixedTimestep$) and developing custom C# control scripts:

### A.3.1 Trajectory Scale Verification

We performed two quantitative checks: **(1) Reference Cube Tracking:** A $1cm^3$ cube was tracked moving across the FOV (Figure 9). The calculated Euclidean distance between start/end camera poses precisely matched 1cm, confirming the 0.01 scaling factor. **(2) Full Trajectory Length:** Navigating the entire colon model yielded a trajectory length of 1.01m (Figure 10), consistent with anatomical expectations for the modeled segments.

### A.3.2 Motion and Viewpoint Control Scripts

Interactive navigation simulating clinical maneuvering used several C# scripts:

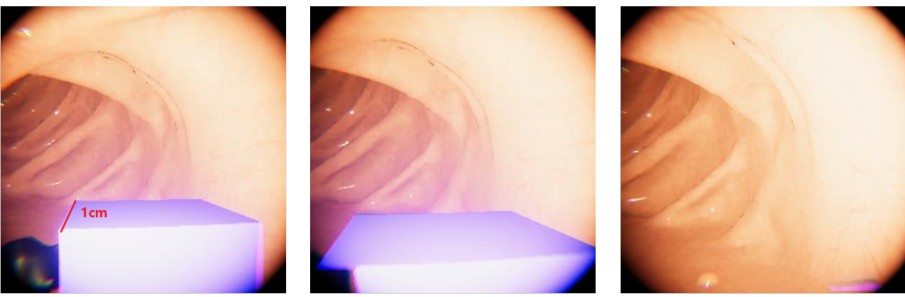

Figure 9: Scale calibration via dynamic reference tracking of a 1cm$^3$ cube.

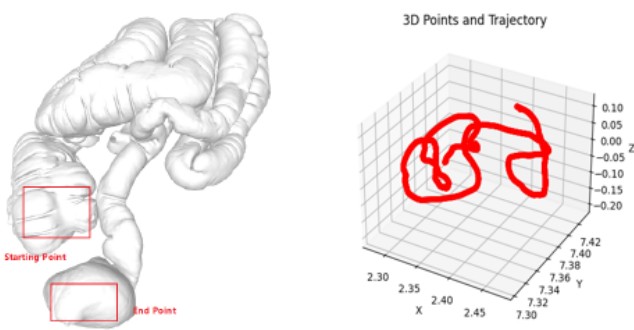

Figure 10: Visualization of the full endoscope trajectory (Right) corresponding to start/end points (Left).

- `PlayerController.cs`: Governed basic forward/backward linear movement (W/S or arrow keys).

- `PlayerControllerForward.cs`: Aligned movement with the camera's forward vector (`capsuleCam.transform.forward`) and added subtle random perturbations for naturalism.

- `KeyboardCameraController.cs`: Allowed keyboard-based pitch/yaw adjustments.

- `MouseCameraController.cs`: Enabled smooth, intuitive viewpoint rotation via mouse input for exploration.

- `FrictionalForce.cs`: Implemented Rigidbody friction to prevent unrealistic sliding on mucosal surfaces.

These scripts facilitated smooth, realistic navigation essential for collecting diverse multi-view data.

### A.4 Data Acquisition and Annotation Details

Synchronized acquisition of RGB and depth data utilized Unity's Recorder features, with careful tuning to ensure data quality.

### A.4.1 RGB Image Capture

We tuned camera (FOV, position, clipping planes) and lighting parameters (Spot Light angles, radius, intensity) to prevent artifacts like overexposure or mesh penetration (Figure 11). RGB images ($1024 \times 1024$, subsequently processed to $320 \times 320$ for benchmark experiments) were captured via Image Sequence Recorder in PNG format, preserving temporal order.

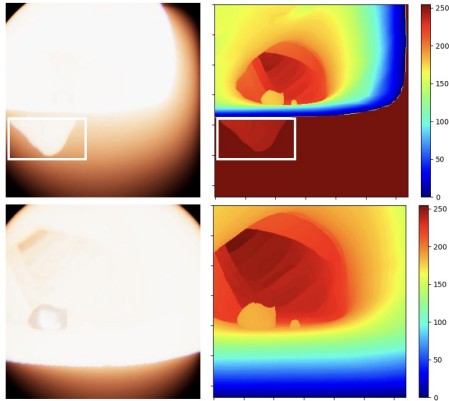

Figure 11: Examples of rendering artifacts avoided through careful parameter tuning: Mesh penetration (Top) and overexposure (Bottom).

### A.4.2 Depth Map Capture

Dense depth maps were extracted using the AOV Image Sequence Recorder. Unity's rendering pipeline calculates per-pixel depth based on geometry relative to the camera. The output is a single-channel grayscale image encoding normalized depth [0,1]. The transformation involves the Model-View-Projection (MVP) matrix to map world space $(x_w, y_w, z_w)$ to clip space, followed by perspective division to get normalized device coordinates $(z_{NDC})$:

$$\text{ClipSpace} = \mathbf{P} \cdot \mathbf{V} \cdot \mathbf{M} \cdot \begin{bmatrix} x_w & y_w & z_w & 1 \end{bmatrix}^T \tag{1}$$

$$z_{NDC} = \frac{(\mathbf{PVM})_{3,1}x_w + (\mathbf{PVM})_{3,2}y_w + (\mathbf{PVM})_{3,3}z_w + (\mathbf{PVM})_{3,4}}{(\mathbf{PVM})_{4,1}x_w + (\mathbf{PVM})_{4,2}y_w + (\mathbf{PVM})_{4,3}z_w + (\mathbf{PVM})_{4,4}} \tag{2}$$

Unity typically uses a reversed z-buffer for better precision near the camera:

$$\text{depth}_{encoded} = 1 - z_{NDC} \tag{3}$$

This means depth=1 is near, depth=0 is far. The relation to metric depth $z_w$ is non-linear. An alternative form often cited is:

$$\text{depth}_{encoded} \approx \frac{n}{z_w} \quad \text{(for large } f\text{)} \quad \text{or more precisely} \quad \frac{(f - z_w)n}{(f - n)z_w} \tag{4}$$

where $n$ and $f$ are the near and far clipping plane distances, respectively. Understanding this mapping is crucial for converting recorded values back to metric distances. Depth maps (16-bit PNG) are pixel-aligned with RGB frames.

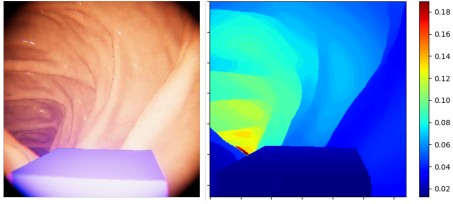

Figure 12: Depth map corresponding to the 1cm$^3$ reference cube, showing consistency.

### A.4.3 Polyp Mask Annotation

Pixel-level binary segmentation masks were generated automatically post-rendering for frames containing polyps (Figure 13). The automated process used image processing:

1. **Grayscale Conversion:** Convert RGB frame to grayscale.



Figure 13: Example: Rendered RGB frame with polyp (left) and automatically generated binary segmentation mask (right).

2. **Threshold Segmentation:** Apply a global intensity threshold, leveraging the distinct rendering of polyp models.

3. **Morphological Refinement:** Use dilation (e.g., with a small kernel) to fill gaps and ensure the mask fully covers the polyp region, compensating for soft edges or minor occlusions.

These automatically generated masks were subsequently reviewed and, if necessary, manually corrected by experienced clinical endoscopists to ensure pixel-perfect accuracy. This pipeline provides accurate, co-registered segmentation masks alongside RGB and dense metric depth, forming a complete dataset for training and evaluating spatial perception and measurement models in a controlled virtual setting.

```
Virtual Dataset
    ├── Virtual Dataset For PolypSense3D.md # This documentation file
    ├── depth_estimation/ # Depth estimation data
    |   ├── camera.txt # Camera parameters file
    |   ├── depths/ # Depth maps directory （PNG）
    |   |   └── *.png # Depth maps（PNG）
    |   ├── images/ # Original images directory （JPG）
    |   |   └── *.jpg # Original image
    |   ├── position_rotation.csv # Camera position and rotation parameters
    ├── image_segmentation/  # Image segmentation data
    |   ├── images/ # Original images directory （PNG）
    |   |   └── *.png # Original image
    |   ├── masks/ # Segmentation masks directory （PNG）
    |   |   └── *.png # Corresponding Segmentation Masks
    └── polyp_estimation/ # Polyp estimation data
        ├── polys_size.xlsx # Polyp size data sheet
        ├── depths/ # Depth maps directory （PNG）
        |   └── *.png # Corresponding Depth maps
        ├──images/ # Images directory （PNG）
        |   └── *.png # Original image
        └──masks/ # Masks directory （PNG）
            └── *.png # Corresponding Segmentation Masks
```

Figure 14: The figure shows the file hierarchy and storage structure of the virtual dataset.

## B  Physical Phantom Dataset Collection and Annotation

This appendix details the construction of the physical phantom dataset (Section 3.2), designed to provide a reproducible and geometry-accurate tested for algorithm evaluation. While the 3D-printed colon model lacks realistic mucosal textures due to material and fabrication limitations, it retains anatomically faithful structure and serves as a practical intermediate between fully synthetic environments and in vivo clinical data. This phantom avoids challenges from intestinal motion and fluid interference but lacks realistic textures, posing difficulties for perception tasks. Such physical phantoms have been widely adopted in medical imaging research [6, 54] as a useful complement for virtual and clinical benchmarking.

## B.1 Colon and Polyp Modeling for 3D Printing

The phantom geometry was derived from the OlympusCAM CT scans within the EndoSLAM dataset [39]. Using custom scripts, DICOM slices were processed into a watertight 3D mesh. We selected the transverse colon segment (modeled length: 50cm) for fabrication due to its clinical relevance and manageable complexity. The model was split into three segments connected via precisely designed interlocking joints (Figure 15) to facilitate high-resolution 3D printing (Material: White Resin; Printer: UnionTech SLA lite600) and accurate reassembly. A key design feature is the semi-open structure, allowing external visualization of the endoscope's navigation and interaction with polyps (Figure 15), greatly aiding experimental observation and validation.

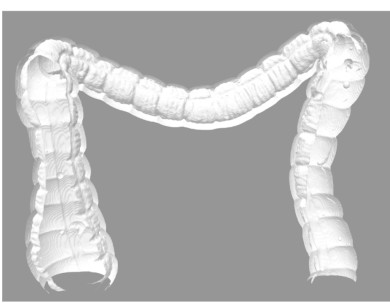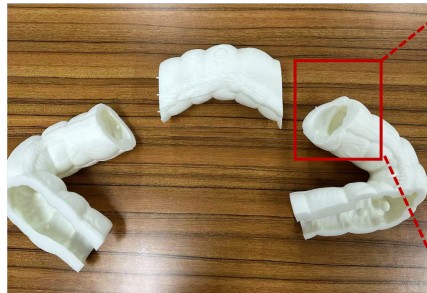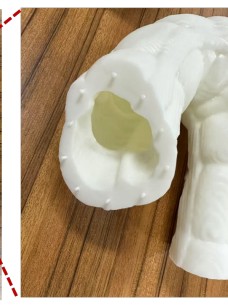

Figure 15: Physical Phantom Design: Digital model (Left), assembled phantom (Center), interlocking joint detail (Right).

13 solid polyps (simulated as spheres/hemispheres, diameters 4.0-24.3 mm, see Table 17) were digitally designed and embedded into the colon mesh prior to printing, placed at clinically relevant locations (e.g., near folds, varying positions along the lumen wall) to introduce realistic challenges (Figure 16).

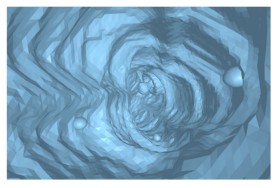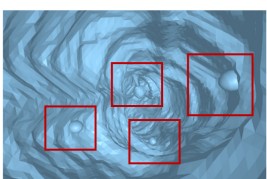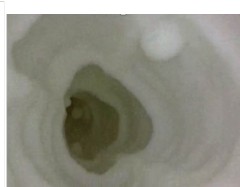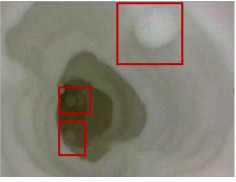

Figure 16: Polyp Implantation: Virtual model showing embedded locations (Left), assembled 3D-printed phantom with visible internal polyps (Right).

## B.2 Camera Calibration and Data Acquisition

We used a compact veterinary endoscope (Model: Y17, Shenda Endoscopy) with manual controls and USB output ($640 \times 480$ resolution) for data acquisition using eCap software.

Meticulous camera calibration preceded data collection: **(1) Optimal Checkerboard Selection:** We evaluated $4 \times 5$ checkerboards with 3mm, 5mm, 8mm, and 10mm squares (Figure 18). The 3mm board (square size of 3mm) provided the best balance of fitting within the FOV at close range and yielding clean undistortion results with minimal residual warping. The final intrinsic parameters used for the physical dataset were derived from this $4 \times 5 \times 3$mm checkerboard calibration (Table 9, first row). **(2) Working Distance Determination:** We tested distances from 5mm to 22mm (Figure 19). The 22mm distance provided stable framing and reliable corner detection across the FOV and was adopted for final calibration captures. **(3) Parameter Estimation:** Using MATLAB's Camera Calibration Toolbox and Zhang's method [70], we recorded videos capturing the $4 \times 5 \times 3$mm board (Figure 20) in all four image quadrants at 22mm distance. Estimated intrinsic parameters (focal lengths, principal point, distortion coefficients) for the chosen configuration are provided in Table 9 (first row). **(4) Field of View (FOV) Calculation:** Using geometric principles at a known distance (14.3mm), the horizontal FOV ($\theta_h$) was calculated as approximately 93.5° and the vertical FOV ($\theta_v$) as approximately 73.6° (Figure 21).

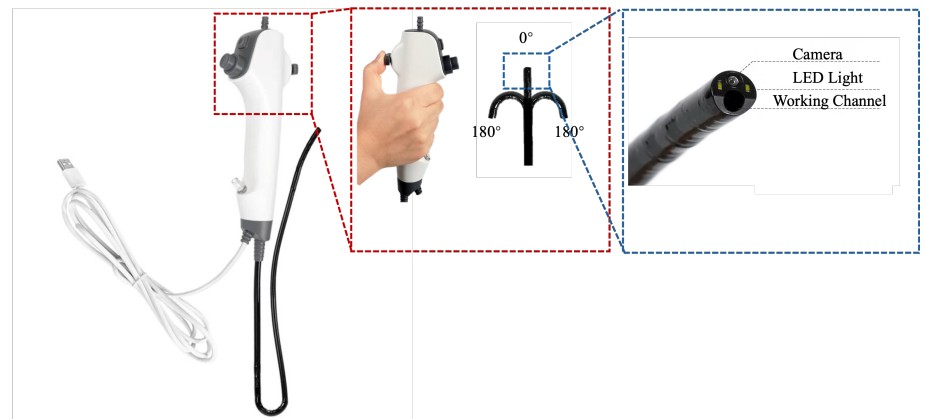

Figure 17: Veterinary endoscope (Model: Y17, Shenda Endoscopy) used for physical phantom data acquisition.

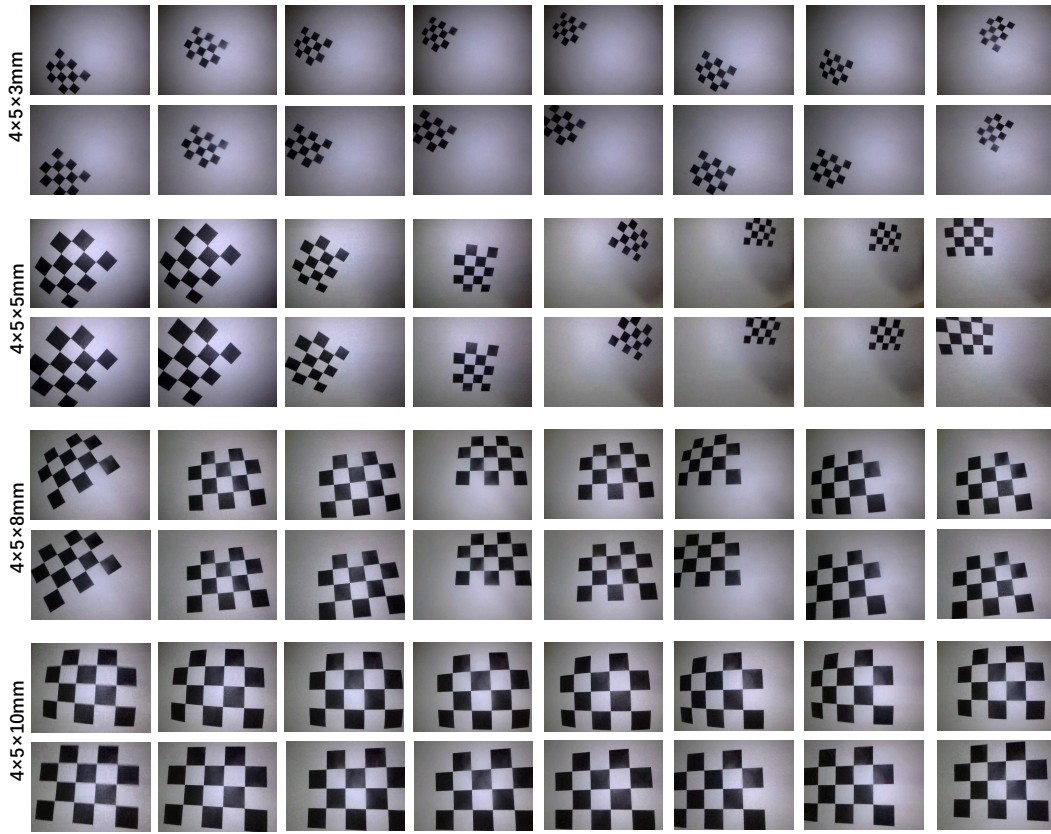

Figure 18: Calibration frames using different checkerboard sizes and their undistorted outputs.

Table 9: Camera Intrinsic Parameters Under Different Calibration Plate Sizes. The parameters from the $4 \times 5 \times 3$mm checkerboard (first row) were used for the physical dataset.

| Checkerboard Size | fx | fy | cx | cy | k1 | k2 | k3 | p1 | p2 |
|---|---|---|---|---|---|---|---|---|---|
| 4×5×3 mm (Used) | 281.7153 | 282.0169 | 351.8296 | 260.3609 | -0.09837 | 0.00298 | 0.00149 | 0.0 | 0.0 |
| 4×5×5 mm | 390.1712 | 389.9811 | 339.1663 | 257.0687 | -0.17276 | 0.00878 | 0.00887 | 0.0 | 0.0 |
| 4×5×8 mm | 258.0550 | 256.1892 | 352.0732 | 263.3593 | -0.06585 | 0.00301 | -0.00089 | 0.0 | 0.0 |
| 4×5×10 mm | 284.7747 | 284.2033 | 350.4888 | 252.4943 | -0.08973 | -0.00929 | 0.00792 | 0.0 | 0.0 |

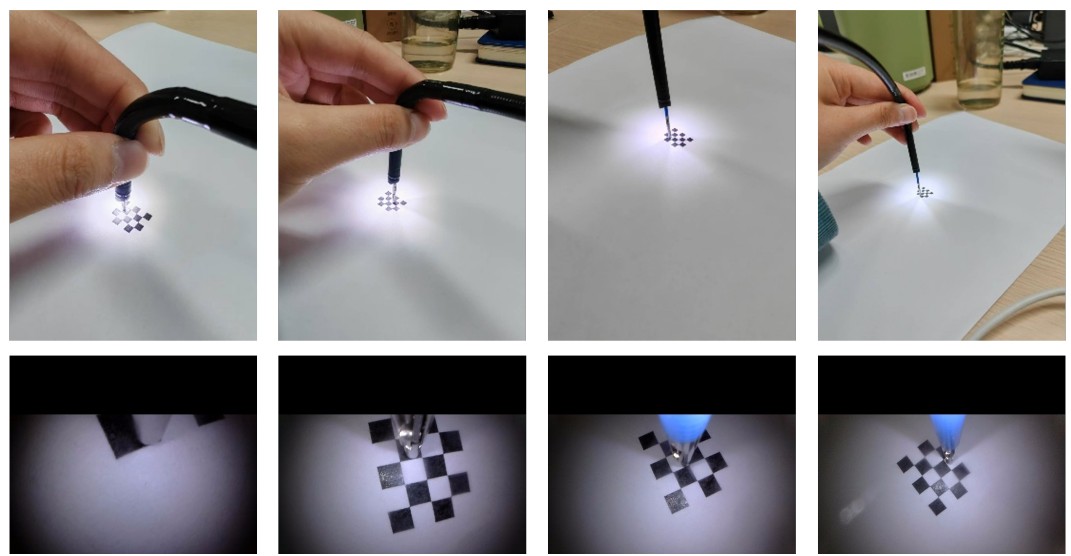

Figure 19: Views captured at different working distances (5mm to 22mm).

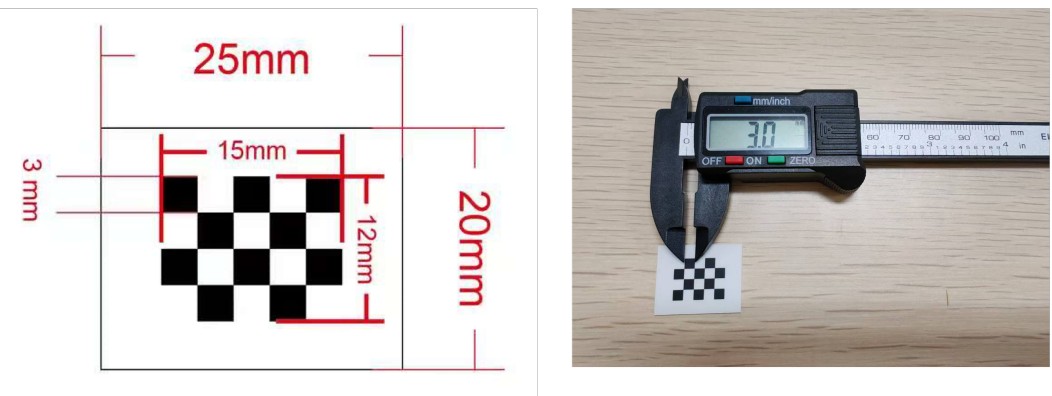

Figure 20: Selected $4 \times 5 \times 3$mm checkerboard used for final calibration.

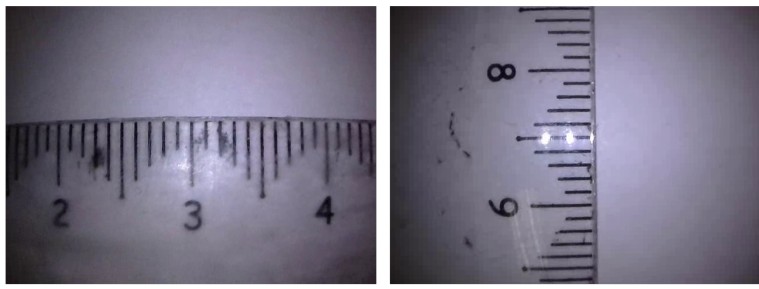

Figure 21: FOV estimation setup showing horizontal (Left) and vertical (Right) extents.

Data acquisition involved manually guiding the endoscope through the phantom while recording video (640x480 resolution) via eCap software. Videos were decomposed into frames. Since polyp locations and sizes are known from the design, and camera intrinsics are calibrated, this dataset enables polyp size estimation using segmentation images. The scale of the reconstruction was aligned using known dimensions within the phantom.

```
Physical Dataset
      ├── Physical Dataset For PolypSense3D.md # This documentation file
      ├── camera.xlsx # Camera parameters (Excel file)
      ├── polys_size.xlsx # Polygon size measurements
      ├── images/ # Extracted video frames (PNG)
      |   └── *.png # Extracted video frame with polyp
      ├── masks/ # Corresponding segmentation masks (PNG)
      |   └── *.png # segmentation mask with polyp
      ├── videos/ # Source video files (WMV)
      |   └── *.wmv # video files with polyp
      └── calibration/ # Camera calibration data (PNG)
          ├── 12_9_3/ # Calibration set 1
          |   └── *.png # the calibration board with 12 × 9 × 3 mm.
          ├── 4_5_3/ # Calibration set 2
          |   └── *.png # the calibration board with 4 × 5 × 3 mm.
          └── FOV/ # Field of view calibration
              └── *.png # the photo for field of view calibration
```

Figure 22: The figure shows the file hierarchy and storage structure of the physical dataset.

## C  Real Clinical Dataset Collection and Annotation

### C.1  Background and Motivation

This appendix extends Section 3.3, detailing the collection and annotation of the real clinical dataset. While synthetic and phantom data are valuable, validating algorithms on authentic clinical data is essential. Existing public datasets often lack either realistic polyp instances or reliable depth/size annotations for those instances, hindering clinical translation. Our clinical subset aims to fill this critical gap by providing in-vivo polyp data annotated with size and sparse depth cues using a novel, practical technique.

### C.2  Camera Calibration and Data Acquisition

Prior to clinical procedures, the Olympus CF-H290I colonoscope was calibrated using the same protocol as the phantom endoscope (Section B.2, Appendix B), employing the $4 \times 5 \times 3$mm checkerboard (see in Figure 23) at a 22mm working distance to estimate intrinsic parameters. The specific intrinsic parameters obtained and used for the clinical dataset are: $f_x = 585.3, f_y = 586.1, c_x = 358.2, c_y = 287.5$, with radial distortion coefficients $k_1 = -0.105, k_2 = 0.012, k_3 = 0.001$ and tangential distortion coefficients $p_1 = 0.0, p_2 = 0.0$. Clinical videos were recorded using the integrated hospital system (Olympus EVIS EXERA III) during routine colonoscopies (resolution 1157x1006). Data collection focused on the withdrawal phase. When polyps were identified, physicians momentarily paused ("freeze-frame") for inspection and to perform the annotation maneuvers described below. These stable segments were manually identified via timestamps and extracted for annotation.

### C.3  3D Polyp Annotation from Clinical Video

Obtaining precise 3D ground truth in-vivo is challenging. We developed a protocol leveraging standard biopsy forceps as a calibrated reference tool. The validation of this protocol's reproducibility and accuracy is detailed in Appendix D.

#### C.3.1  Real Polyp Size Annotation via Proximal Comparison

This technique estimates size from a single frame using forceps as a scale bar.

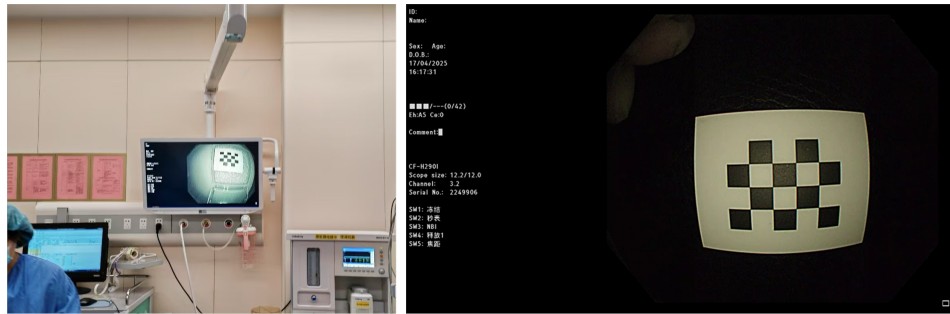

Figure 23: Camera Calibration Process for Clinical Datasets.

1. **Frame Selection:** Identify a clear, stable frame where the physician has positioned fully opened biopsy forceps (known $d_{real}$=5mm width) near the target polyp, ideally at a similar visual depth (Figure 24, Frame 1).

2. **Forceps Annotation:** Trained annotators mark the two metallic endpoints of the forceps tips in the image (Figure 24, Frame 2). Pixel distance $d_{pixel}$ is measured.

3. **Scale Calculation:** Compute local scale: $scale = d_{real}/d_{pixel}$ (mm/pixel).

4. **Polyp Segmentation:** Generate a binary mask of each polyp using prompt-based segmentation methods such as SAM [23] with point or box prompts. This step is completed by trained annotators under the supervision of experienced endoscopists, ensuring accurate delineation of polyp boundaries.

5. **Size Calculation:** Apply the scale factor to mask dimensions (e.g., major axis $L_{pixels}^{major}$) to get metric size: $L_{mm}^{major} = L_{pixels}^{major} \times scale$. This method involves computing the major axis of the minimum area bounding rectangle of the segmentation mask in pixels, then converting it to millimeters using the derived scale factor. Similarly, the minor axis can be computed. The polyp size is reported as the larger of these two metric dimensions. The equations are:

$$L_{pixel}^{major}, L_{pixel}^{minor} = \text{Axes}(\text{MinAreaRect}(\text{Mask})) \tag{5}$$

$$L_{mm}^{major} = L_{pixel}^{major} \times \text{scale} \tag{6}$$

$$L_{mm}^{minor} = L_{pixel}^{minor} \times \text{scale} \tag{7}$$

$$\text{PolypSize}_{mm} = \max(L_{mm}^{major}, L_{mm}^{minor}) \tag{8}$$

This provides an objective, quantifiable estimate grounded in a physical reference, improving upon purely subjective visual assessment.

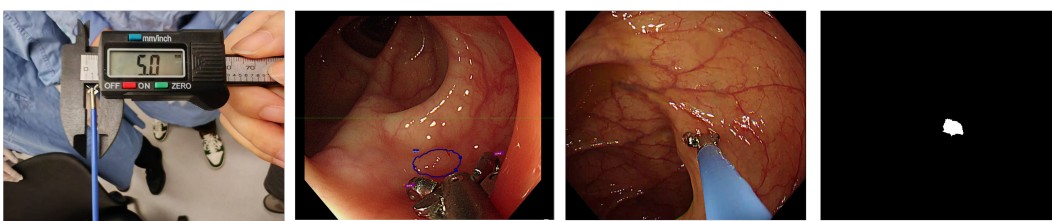

Figure 24: Workflow for polyp size annotation via proximal comparison: (1) Forceps opened (5mm ref); (2) Endpoints annotated; (3) RGB frame; (4) Segmentation mask. Scale factor derived from (2) is applied to (4).

### C.3.2 Depth Annotation via Extended Biopsy Forceps Contact

This technique provides sparse metric depth points using markings on the forceps shaft.

1. **Forceps Preparation:** Biopsy forceps (Olympus FB-240U) were prepared by adding clear black markers at 5mm intervals along the blue shaft, creating 8 extension levels (0mm to 35mm), as shown in Figure 25.

2. **Contact Maneuver:** During the procedure, in a stable frame, the physician extends the forceps until the tip gently contacts the polyp apex or adjacent flat mucosa. The extension level (number of visible 5mm segments) is noted if possible, or the visible pixel length is measured later.

3. **Perspective Rectification:** Endoscopic views often show the forceps shaft at an angle, causing perspective distortion, as shown in Figure 26. To standardize measurements, we use a homography transformation. A universal homography $\mathbf{H_{all}}$ was pre-computed by capturing multiple views of the forceps shaft at various known orientations and distances against a planar surface. Corner points of the forceps shaft segments were manually annotated in these calibration images. The Direct Linear Transform (DLT) algorithm was used to compute the homography mapping these observed points to a canonical frontal view where the shaft is parallel to the image plane. The transformation is defined by:

$$\mathbf{x'} = \mathbf{H_{all}x} \tag{9}$$

where $\mathbf{x}$ are homogeneous coordinates of points in the original image and $\mathbf{x'}$ are coordinates in the rectified view. $\mathbf{H_{all}}$ is a $3 \times 3$ matrix. Robustness of this universal homography was validated across typical endoscopic angulations encountered during the procedure (see Appendix D). Applying $\mathbf{H_{all}}$ to any frame rectifies the forceps shaft to a standard perspective.

4. **Pixel-to-Depth Regression:** Using a reference extension sequence captured with the calibrated endoscope (rectified using $\mathbf{H_{all}}$), we measured the vertical pixel coordinate $x$ of each 5mm marker in the rectified view and paired it with its known depth $Z_{marker}$ (0, 5, 10... 35mm). We fit a 2nd-order polynomial regression to these pairs (Figure 27):

$$Z_{marker}(x) = ax^2 + bx + c \tag{10}$$

For our setup, the fitted coefficients were $a = 0.02308$, $b = 3.27$, and $c = -0.001045$.

5. **Depth Estimation:** For any clinical frame, apply $\mathbf{H_{all}}$ to rectify the forceps view. Measure the vertical pixel coordinate $x_{tip}$ corresponding to the visible end of the extended forceps shaft (at the contact point). Apply the regression function $Z_{marker}(x_{tip})$ to estimate the metric depth of the contact point.

6. **Auxiliary Depth Points:** To enhance spatial supervision, we additionally annotate four key points: the frontmost tip of the biopsy forceps (whose depth is $Z_{marker}(x_{tip})$) and three adjacent points on the intestinal wall or lesion surface in direct contact with, or very close proximity to, the tip (Figure 28). These auxiliary contact points are typically chosen on the surrounding mucosa near the tip region. The depth of these auxiliary points is assumed to be approximately equal to $Z_{marker}(x_{tip})$, allowing sparse yet clinically meaningful depth propagation from the forceps to adjacent areas.

This process yields sparse but valuable metric depth measurements directly from clinical video, enabling evaluation of depth-aware algorithms under real conditions. Quality control included annotator training on identifying stable frames and accurate point marking, with cross-validation checks detailed in Appendix D. Potential errors (operator variability, contact pressure affecting depth, rectification inaccuracies) are acknowledged limitations discussed in the main paper.

## D  Clinical Annotation Protocol Validation

This section details the validation procedures for the biopsy-forceps-assisted annotation protocol used for the clinical dataset, addressing its reproducibility and potential sources of variability.

### D.1  Validation of Universal Homography for Forceps Rectification

To assess the robustness of our universal homography matrix ($\mathbf{H_{all}}$) under viewpoint variation, we conducted an experiment using two 50-frame videos with the endoscope tip angled left and right. To provide a clear comparison of the tilt differences, we fixed the forceps at 46 mm, and depth estimates

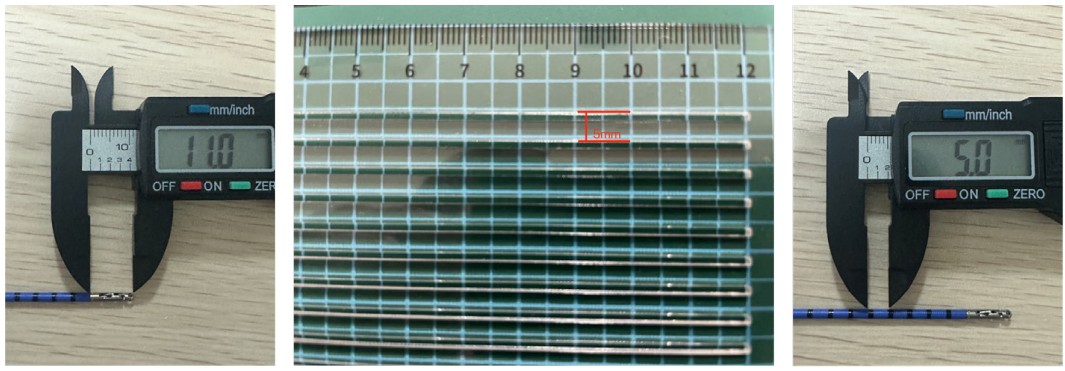

Figure 25: Discrete absolute scale labels were created for the biopsy forceps using a custom-designed calibration ruler.

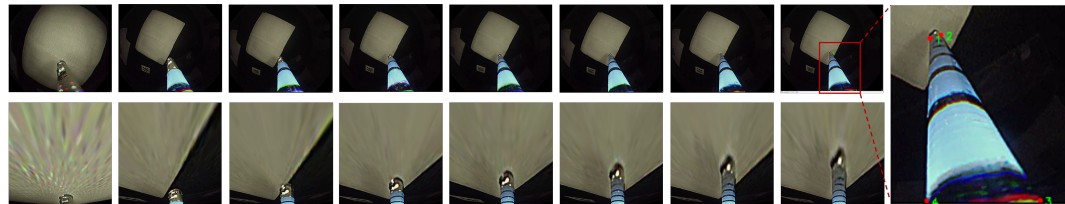

Figure 26: Perspective rectification sequence: Original frames (top row) and rectified frames (bottom row) for an extension sequence used to build the pixel-to-depth mapping.

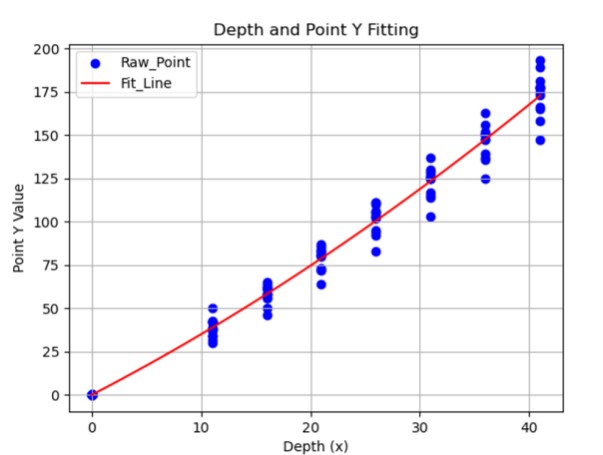

Figure 27: Polynomial regression fitting pixel coordinates (vertical position in rectified view) to known metric depth (mm) based on forceps markers. The fitted curve is $Z(x) = 0.02308x^2 + 3.27x - 0.001045$.

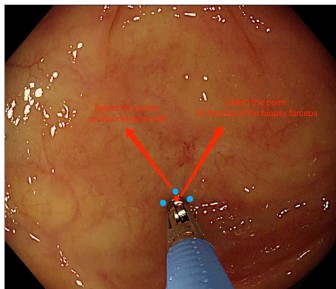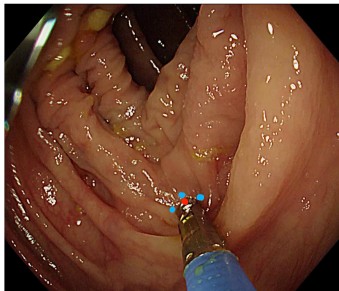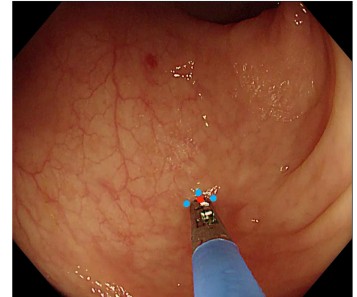

Figure 28: The metallic tip of the forceps (red) is extended until it gently touches the polyp or adjacent mucosa. In addition to the forceps tip, three nearby contact points on the mucosal surface (blue dots) are annotated. These auxiliary points lie in close proximity to the tip and are used to infer local tissue depth, based on the metric depth derived from the forceps shaft scale.

Table 10: Impact of Viewpoint Variation on Depth Estimation

| Video Sequence | Viewpoint Angle | Est. Depth | Abs. Error |
|---|---|---|---|
| Vertical-1 | Vertical (Straight) | 48.52 | 2.52 |
| Left-1 | Left Deviations | 51.03 | 5.03 |
| Left-2 | Left Deviations | 53.15 | 7.15 |
| Left-3 | Left Deviations | 53.85 | 7.85 |
| Left-4 | Left Deviations | 51.39 | 5.39 |
| Left-5 | Left Deviations | 51.74 | 5.74 |
| Right-1 | Right Deviations | 52.27 | 6.27 |
| Right-2 | Right Deviations | 54.89 | 8.89 |
| Right-3 | Right Deviations | 64.03 | 18.03 |
| Right-4 | Right Deviations | 65.94 | 19.94 |
| Right-5 | Right Deviations | 68.61 | 22.61 |

were computed every 10 frames using $\mathbf{H_{all}}$. As shown in Table 10, the estimation error was low under stable, straight-on views (2.52 mm), but increased significantly with angular deviations, up to 22.61 mm. These results validate both the accuracy of $\mathbf{H_{all}}$ in intended conditions and the importance of selecting stable frames for reliable ground truth.

## D.2 Training and Quality Control for Annotators

All annotators involved in the clinical data processing underwent a standardized training program. This included:

- Review of endoscopic anatomy and polyp morphology.
- Detailed instruction on the annotation protocol, including criteria for stable frame selection, precise marking of forceps tips, and meticulous polyp boundary delineation with SAM and manual refinement.
- Practice sessions on a pilot set of images with feedback from experienced endoscopists.

During dataset creation, a random 10% sample of annotations was cross-checked by a senior annotator or physician. Any discrepancies exceeding a predefined tolerance (e.g., >10% in size, >2mm in depth point location) triggered a re-annotation and consensus discussion. This iterative quality control process aimed to minimize systematic errors and ensure annotation consistency.

While these validation steps demonstrate the general reliability and reproducibility of our clinical annotation protocol, inherent challenges in in-vivo settings, such as tissue deformation, specular reflections, and subtle movements, mean that clinical annotations will always have a degree of uncertainty compared to ground truth from controlled virtual or physical environments. This is a realistic reflection of the difficulties faced in clinical practice.

```
Clinical Dataset
    ├── Clinical Dataset For PolypSense3D.md # This documentation file
    ├── calibrations # Calibration and lens tracking data
    │   ├── XX(Random Number)_XX(Lens ID)/ # Calibration images for a specific lens
    │   │   └── *.png # Checkerboard calibration images
    │   ├── camera.xlsx # Camera intrinsic and distortion parameters
    │   └── video_info.xlsx # Table mapping videos to Lens IDs
    ├── labeled_data # Labeled data
    │   ├── depth # Coarse depth annotations
    │   │   ├── json/ # Biopsy forceps point annotations (JSON format)
    │   │   └── rgb/ # Clear video frames of biopsy forceps closing action (JPG format)
    │   ├── segmentation # Precise polyp segmentation masks
    │   │   ├── mask/ # Manual polyp segmentation masks (PNG format)
    │   │   └── rgb/ # Corresponding clear polyp video frames (JPG format)
    │   ├── size # Polyp size annotations
    │   │   ├── json/ # Biopsy forceps open for comparison point annotations (JSON format)
    │   │   └── rgb/ # Clear video frames of biopsy forceps open for polyp comparison (JPG format)
    ├── polyp_frame # Polyp video frame image data
    │   └── sequenceXXX/
    │       └── polypY/
    │           └── frame_XXXX.jpg
    └── polyp_video # Polyp video data
        └── sequenceXXX/
            └── polypY/
                └── polypY.mp4
```

Figure 29: The figure shows the file hierarchy and storage structure of the clinical dataset.

# E    Detailed Dataset Statistics

This appendix provides comprehensive statistical analysis supplementing Section 4 and Table 2. It includes detailed breakdowns of frame counts and polyp instances per source (virtual, physical, clinical), distributions of polyp sizes, locations within colon segments, annotated morphologies (where available), image resolutions, depth value ranges (virtual and estimated physical), camera parameter variations, and an analysis of data diversity and subset-specific limitations. We have conducted a detailed analysis of the polyp information in all datasets, including the virtual dataset (30 in total) and the clinical dataset (53 in total), as shown in Table 1617 18.

Table 11: Comparative Analysis of Polyp Size Statistics Across Datasets (Unit: mm)

| Metric | Virtual Dataset (ref to Tab. 16) | Physical Dataset (ref to Tab. 17) | Clinical Dataset (ref to Tab. 18) |
|---|---|---|---|
| Mean | 9.57 | 9.28 | 3.28 |
| Median | 9.75 | 6.45 | 2.85 |
| Maximum | 20.52 (largest dimension) | 14.89 | 9.98 |
| Minimum | 1.79 (smallest dimension) | 4.00 | 1.39 |
| Distribution (<10mm) | 15/30 (50.0%) | 9/13 (69.2%) | 53/53 (100%) |
| Distribution (10-20mm) | 13/30 (43.3%) | 4/13 (30.8%) | 0/53 (0%) |
| Distribution (>20mm) | 2/30 (6.7%) | 0/13 (0.0%) | 0/53 (0%) |

Our multi-source dataset reveals distinct and complementary polyp size distributions across the virtual, physical, and clinical domains (Table 11). The virtual dataset covers the widest dynamic range (1.79–20.52mm), with a relatively even size distribution: 50.0% of polyps are under 10mm, 43.3% between 10–20mm, and 6.7% above 20mm, making it well-suited for scale-aware model development and controlled benchmarking. The physical dataset narrows this spectrum slightly, with 69.2% of polyps below 10mm. Despite simplified texture and lighting, it provides real-world imaging characteristics and depth sensor responses, serving as a practical intermediate domain. In stark contrast, the clinical dataset is heavily biased toward small polyps: 100% are under 10mm. The median polyp size is just 2.85mm, less than half that of the virtual and physical domains. It is worth noting that large polyps are relatively rare in clinical practice, and our dataset will be further updated as more such cases are collected. This reflects real-world clinical practice, where early-stage lesions are both common and hard to detect or quantify due to their subtle appearance, motion artifacts, and visual occlusion.

Image resolutions vary: virtual data is captured at $1024 \times 1024$ (PNG) and processed to $320 \times 320$ for experiments, physical phantom data is $640 \times 480$ (frames from video), and clinical data is typically $1157 \times 1006$ (frames from video). Depth value ranges for virtual data are [0,1] normalized, corresponding to metric distances typically from a few millimeters to several centimeters depending

on camera clipping planes. For estimated physical depth and sparse clinical depth, values are in millimeters. Camera parameter variations primarily concern the calibrated intrinsics for the physical and clinical endoscopes, as detailed in Appendix B.2.

This structure creates a progressive validation framework: virtual data offers perfect ground truth, physical phantoms introduce real-world textures, and clinical data provides authentic challenges. While each component has limitations (e.g., clinical data currently lacks large polyps), their complementary strengths enable comprehensive evaluation from controlled to real-world conditions. The tiered size distributions particularly support progressive algorithm validation across difficulty levels.

# F Benchmark Implementation Details

This appendix provides exhaustive details regarding the implementation of the benchmark experiments described in Section 5, ensuring reproducibility.

## F.1 Baseline Model Implementation

Polyp size estimation relies heavily on two core tasks: accurate segmentation of polyps in 2D images and reliable estimation of depth in endoscopic scenes. To benchmark these capabilities on our dataset, we evaluate a suite of representative models covering both segmentation and depth estimation.

**Segmentation Models.** For polyp segmentation, we select several widely used and state-of-the-art models to demonstrate the dataset's value as a benchmark platform. It is important to note that these models were trained on existing public datasets, such as Kvasir-SEG[20] and CVC-ClinicDB[4]. MS-Net [28], an encoder-decoder network, uses domain-specific normalization and knowledge transfer for robust segmentation of heterogeneous data. First, as a classic and domain-specific baseline, we adopt PraNet [14], a leading architecture for polyp segmentation prior to the foundation model era. PraNet integrates a parallel partial decoder and reverse attention modules, enabling precise boundary recovery and achieving high performance on multiple public polyp datasets.

Considering the computational limitations and real-time requirements of clinical endoscopy, we purposefully select lightweight large-scale segmentation models. Thus, we include SAM [23] (specifically, the ViT-H variant for its strong performance) and MobileSAM [65], derived from the Segment Anything Model (SAM), a prompt-driven segmentation framework proposed by Meta AI. SAM is trained on a massive corpus of general images and can segment arbitrary objects, including unseen categories, via flexible prompt mechanisms. Its generalization ability has shown promise even in medical domains. MobileSAM, a distilled version of SAM specifically optimized for lightweight deployment, is over 60 times smaller and 5 times faster than SAM yet retains competitive accuracy, making it suitable for real-time or mobile applications.

We also evaluate VM-UNet [45], a novel pure state space model (SSM)-based architecture designed for medical image segmentation. Unlike conventional CNN or Transformer models, VM-UNet leverages Mamba blocks for long-range context modeling with linear complexity. It represents a new generation of lightweight and scalable architectures tailored to structured visual understanding.Finally, we also employed two state-of-the-art SOTA methods: ASPS[25] and Sam2-Unet[57]. ASPS is an enhanced arbitrary segmentation model designed for polyp segmentation tasks, while Sam2-Unet provides a powerful encoder for natural and medical image segmentation.

**Depth Estimation Models.** Robust monocular depth estimation is critical for accurate polyp size measurement but challenging due to scale ambiguity and complex textures in endoscopic imagery. To benchmark depth estimation capabilities, we trained models using the densely annotated data from the virtual dataset we proposed. Subsequently, we select representative state-of-the-art models: Depth Anything V1 (DAM V1) [60] and V2 (DAM V2) [61], ZoeDepth [5], and Depth Anything V2-mini [61], each offering unique advantages suited to medical contexts. Depth Anything V1 and V2 leverage massive unlabeled datasets and advanced knowledge distillation techniques, demonstrating exceptional zero-shot generalization capabilities vital for adapting effectively to novel endoscopic scenes. ZoeDepth uniquely integrates relative and metric depth estimation through extensive multi-dataset pre-training and targeted metric fine-tuning, providing reliable absolute depth predictions essential for clinical measurement tasks. DAM V2-mini produces high-resolution metric depth maps with precise boundary delineation using a multi-scale Vision Transformer architecture. Its built-in focal length estimation module enables accurate scale inference without explicit camera calibration, directly addressing scale ambiguities inherent to monocular endoscopic images. These advanced models comprehensively validate our dataset's utility as a challenging benchmark and contribute significantly to advancing depth estimation research in endoscopic medical imaging.

## F.2 Training and Fine-tuning Details

To ensure fair evaluation, all models were trained and tested under the PyTorch framework (version 1.13) using a consistent protocol. The number of training epochs was fixed to 10 across all segmentation models, and varied for depth models as specified in Table 13. Batch size was adjusted based on GPU memory consumption: PraNet, MobileSAM, and VM-UNet were trained with a batch size

of 16. Sam2-unet was trained with a batch size of 12. SAM and ASPS, due to their higher memory demand, were trained with a batch size of 4. Depth models also used varying batch sizes as detailed in Table 13. All experiments were conducted on a single NVIDIA RTX 4090 GPU with 24GB VRAM.

The PolypSense3D dataset was split into training and test sets using a 7:3 ratio. As shown in Table 2. The virtual (Unity) subset has a total of 32,000 RGB-D frames with aligned absolute-scale dense depth maps, including over 16,000 frames with polyp annotations. The physical phantom subset includes 13 video sequences with 74 frames with polyps. The clinical dataset comprises over 11,000 frames, including 438 frames with polyps. Patient-level separation was ensured between the train and test sets for the clinical data to prevent data leakage. The high-quality pixel-wise polyp segmentation masks all three subsets, manually annotated and reviewed by experienced endoscopists and trained annotators. These masks enable the dataset to be directly used for supervised training and fine-tuning of polyp segmentation models. The Unity dataset, with its larger scale and balanced composition, serves as the primary training source, while the physical and clinical subsets are intended for fine-tuning and evaluating cross-domain generalization. For depth estimation tasks, the Unity subset provides dense, per-pixel ground truth depth with absolute metric scale across all 32K frames, making it suitable for supervised depth model training. The clinical dataset includes sparse, forceps-calibrated depth at contact points, which are used for evaluation only. All input images were resized to 320×320 pixels unless otherwise specified by the original model implementation.

Hyperparameters used for training/fine-tuning models on the PolypSense3D training split are shown in Table 12 for segmentation and Table 13 for depth estimation. Each model was trained with an optimizer and learning rate schedule adapted to its architectural characteristics. Data augmentation strategies included normalization for all models. For segmentation, VM-UNet additionally used random horizontal flipping, vertical flipping, and rotation. For depth estimation, Depth Anything models used random horizontal flipping, unified normalization, cropping, and image size adjustment.

Table 12: Training hyperparameters for polyp segmentation models.

| Model | Learning Rate | Batch Size | Optimizer | Weight Decay | Epochs | LR Schedule | Loss Function | Data Augmentation |
|---|---|---|---|---|---|---|---|---|
| MSNet [28] | $5 \times 10^{-2}$ | 16 | SGD | 0.0005 | 10 | Step Decay (step=5, gamma=0.1) | CrossEntropyLoss | Unified Normalization + Random Horizontal Flip + Vertical Flip + Rotation |
| PraNet [14] | $1 \times 10^{-4}$ | 16 | Adam | 0 | 10 | Step Decay (step=3, gamma=0.1) | Custom Structure Loss (BCE + IoU) | Unified Normalization |
| SAM [23] | $8 \times 10^{-4}$ | 4 | Adam | 0.0001 | 10 | LambdaLR (linear warmup then decay) | Focal Loss + Dice Loss + MSE Loss | Unified Normalization |
| MobileSAM [65] | $1 \times 10^{-5}$ | 16 | AdamW | 0 | 10 | CosineAnnealingLR | Focal Loss + Dice Loss + MSE Loss | Unified Normalization |
| VM-UNet [45] | $1 \times 10^{-3}$ | 16 | AdamW | 0.01 | 10 | CosineAnnealingLR | BCE-Dice Loss | Unified Normalization + Random Horizontal Flip + Vertical Flip + Rotation |
| ASPS [25] | $1 \times 10^{-5}$ | 4 | AdamW | 0.0001 | 10 | Fixed Learning Rate | Seg Loss (BCE Loss + 0.5 * Dice Loss + MSE Loss)+ lmbda * Confidence Loss | Unified Normalization + Random Horizontal Flip + Vertical Flip + Rotation + Random Color Jitter + Random Blur and Noise |
| Sam2-unet [57] | $1 \times 10^{-3}$ | 12 | AdamW | 0.0005 | 10 | CosineAnnealingLR | Custom Structure Loss (3x) | Resize + Random Horizontal Flip + Vertical Flip + Normalize |

Table 13: Training hyperparameters for depth estimation models on the virtual training split.

| Model | Learning Rate | Batch Size | Optimizer | Weight Decay | Epochs | LR Schedule | Loss Function | Data Augmentation |
|---|---|---|---|---|---|---|---|---|
| DAM V1 [60] | $8 \times 10^{-5}$ | 16 | AdamW | 0.01 | 5 | LambdaLR (linear warmup then decay) | SILog Loss | Random Horizontal Flip |
| DAM V2 [61] | $5 \times 10^{-6}$ | 8 | AdamW | 0.01 | 30 | Polynomial Decay (power=0.9) | SILog Loss | Unified Normalization, Cropping (random), Image Size Adjustment |
| ZoeDepth [5] | $1.61 \times 10^{-6}$ | 4 | AdamW | 0.01 | 5 | LambdaLR (linear warmup then decay) | SILog Loss | None (as per original paper for fine-tuning) |
| DAM V2-mini [61] | $5 \times 10^{-6}$ | 4 | AdamW | 0.01 | 30 | Polynomial Decay (power=0.9) | SILog Loss | Unified Normalization, Cropping (random), Image Size Adjustment |

## F.3 Evaluation Protocol

To ensure comprehensive and consistent performance evaluation, we adopt widely used metrics for both segmentation and depth estimation tasks.

### F.3.1 Segmentation Evaluation

For polyp segmentation, we use five standard quantitative metrics:

- **Mean Dice Coefficient (mDice)**: Measures overlap between predicted and ground truth masks. The Dice coefficient for a class is calculated as $Dice = \frac{2 \times |P \cap G|}{|P| + |G|}$, where $P$ is the

predicted mask, $G$ is the ground truth mask, and $|\cdot|$ denotes the number of pixels. mDice is the average Dice over all polyp instances.

- **Mean Intersection-over-Union (mIoU)**: Evaluates the area of intersection divided by the area of union between prediction and ground truth. The IoU for a class is $IoU = \frac{|P \cap G|}{|P \cup G|}$. mIoU is the average IoU over all polyp instances.

- **Recall (Sensitivity)**: The ratio of correctly predicted positive pixels to all actual positives, $Recall = \frac{TP}{TP+FN}$, averaged over instances.

- **Precision (Positive Predictive Value)**: The ratio of correctly predicted positive pixels to all predicted positives, $Precision = \frac{TP}{TP+FP}$, averaged over instances.

- **F1 Score**: The harmonic mean of precision and recall, $F1 = \frac{2 \times Precision \times Recall}{Precision + Recall}$, averaged over instances.

### F.3.2 Depth Estimation Evaluation

For monocular depth prediction, we use the following standard metrics, evaluated on pixels corresponding to segmented polyp regions or the entire image where appropriate:

- **Root Mean Squared Error (RMSE)**: $\sqrt{\frac{1}{N} \sum_{i=1}^{N} (d_i - d_i^*)^2}$, where $d_i$ is the predicted depth, $d_i^*$ is the ground truth depth, and $N$ is the total number of valid pixels.

- **Mean Absolute Relative Error (AbsRel)**: $\frac{1}{N} \sum_{i=1}^{N} \frac{|d_i - d_i^*|}{d_i^*}$.

- **Log$_{10}$ Error (Log10)**: $\frac{1}{N} \sum_{i=1}^{N} |\log_{10}(d_i) - \log_{10}(d_i^*)|$.

- $\boldsymbol{\delta < thr}$: The percentage of predicted depth values $d_i$ such that $\max(\frac{d_i}{d_i^*}, \frac{d_i^*}{d_i}) < thr$. We use $thr = 1.25, 1.25^2$.

### F.3.3 Polyp Size Estimation

For the final polyp size estimation task, we report:

- **Mean Absolute Error (MAE)**: Measures the average absolute difference between the predicted and true polyp size in millimeters, MAE = $\frac{1}{M} \sum_{j=1}^{M} |S_j^{pred} - S_j^{GT}|$, where $M$ is the number of polyps.

These metrics together provide a balanced evaluation of both pixel-wise accuracy and spatial structure consistency in endoscopic scenes.

### F.4 Computational Resources

Computational efficiency is crucial for clinical applicability. Table 14 and Table 15 summarize the inference speed and model size for the benchmarked models. Segmentation models were benchmarked on 30 frames of $320 \times 320$ resolution. Depth estimation models were benchmarked on a per-frame basis with their respective input resolutions. PraNet demonstrates high efficiency suitable for real-time clinical deployment. Foundation models like SAM are larger and slower but offer strong generalization. MobileSAM provides a good trade-off. VM-UNet shows intermediate efficiency. For depth models, DAM V2-mini is notably efficient among the high-performing methods. The optimal choice often involves balancing accuracy, speed, and resource constraints.

Table 14: Comparison of computational resources for polyp segmentation models (Inference on NVIDIA RTX 4090).

| Model | Inference Time (ms for 30 frames) | FPS (frames/sec) | Model Size (MB) | #Parameters (M) | #GPUs Used |
|---|---|---|---|---|---|
| MSNet [28] | 3980 | 7.54 | 113.69 | 29.74 | 1 |
| PraNet [14] | 830 | 36.14 | 124.41 | 32.55 | 1 |
| SAM (ViT-H) [23] | 7540 | 3.98 | 2445.57 | 641.09 | 1 |
| MobileSAM [65] | 1340 | 22.39 | 45.17 | 10.13 | 1 |
| VM-UNet [45] | 3750 | 8.00 | 104.63 | 27.43 | 1 |
| ASPS [25] | 9355 | 3.21 | 199.56 | 52.27 | 1 |
| Sam2-unet [57] | 3784 | 7.93 | 826.03 | 216.53 | 1 |

Table 15: Comparison of computational resources for monocular depth estimation models (Inference on NVIDIA RTX 4090).

| Model | Inference Time (ms/frame) | FPS (frames/sec) | Model Size (MB) | #Parameters (M) | #GPUs Used |
|---|---|---|---|---|---|
| DAM V1 (ViT-L) [60] | 130 | 7.69 | 1331.2 | 335.79 | 1 |
| DAM V2 (ViT-L) [61] | 135 | 7.41 | 1330.0 | 335.00 | 1 |
| ZoeDepth (NK) [5] | 90 | 11.11 | 1433.6 | 344.82 | 1 |
| DAM V2-mini [61] | 66 | 15.15 | 410.0 | 100.00 | 1 |

# G    Extended Benchmark Results and Error Analysis

This appendix provides a comprehensive extension of the experimental results presented in the main paper (Section 5). In addition to evaluating the segmentation and depth estimation models, we further benchmark their integration in the downstream task of polyp size estimation. The evaluation is conducted across three distinct subsets: the virtual simulation dataset, the physical platform dataset, and the real clinical dataset. Detailed quantitative results are presented in Section G.1. We further conduct a detailed error analysis. This includes identifying sources of estimation error, analyzing outliers, and interpreting the clinical relevance of misestimations, particularly around size thresholds critical for cancer risk stratification.

To complement the numerical analysis, we also provide extensive qualitative visualizations in Section G.2, including examples of segmentation masks, depth predictions, and resulting 3D-aware measurements.

## G.1    Quantitative Results: Per-Polyp Size Estimation

This section presents a comprehensive evaluation of per-polyp size estimation across three datasets. When calculating polyp size, we typically select the best polyp size estimation result as a reference. However, the model that performs best may not always be used, which could be due to the varying effectiveness of different models in handling local details. The estimation pipeline relies on segmentation and depth prediction outputs, combined with known or calibrated camera intrinsics. Once the segmentation mask is obtained, the polyp region is enclosed by its minimum area bounding rectangle to determine its pixel-level width ($W_{px}$) and height ($H_{px}$). We then extract the corresponding depth map—either from synthetic ground truth or depth estimation models—and use the depth value $Z_{center}$ at the center of this bounding box as the representative depth for size computation.

For the virtual dataset, the camera intrinsics (focal length $f_{virtual}$, principal point $c_{virtual}$) are directly available from the Unity rendering engine. For the physical and real clinical datasets, intrinsics ($f_{calib}$, $c_{calib}$) are obtained via the calibration procedure detailed in Appendices B.2 and C.2. The size estimation follows the pinhole camera projection model. Specifically, the real-world width $W_{mm}$ and height $H_{mm}$ are computed as:

$$W_{mm} = \frac{W_{px} \times Z_{center}}{f_x} \tag{11}$$

$$H_{mm} = \frac{H_{px} \times Z_{center}}{f_y} \tag{12}$$

where $f_x$ and $f_y$ are the focal lengths in pixels (from $f_{virtual}$ or $f_{calib}$). The estimated polyp size for that frame is then taken as $\max(W_{mm}, H_{mm})$.

**Virtual Dataset:** As shown in Table 16, the virtual dataset enables per-polyp size estimation with complete 3D ground truth. Each polyp's size label ("Label" column) is computed from the maximum

of its X and Y dimensions in millimeters, representing the largest visible extent. We evaluate two configurations:

- Seg(Pred)+Depth(GT): Predicted segmentation (MobileSAM) with ground truth depth.

- Seg(GT)+Depth(Pred): Ground truth segmentation with predicted depth (DAM V2-mini).

Table 16: Benchmark on Unity Dataset: Per-Polyp Size Estimation (mm). "Label" is max(X,Y) of 3D model.

| Polyp ID | Label | Seg(Pred)+Depth(GT) | | Seg(GT)+Depth(Pred) | | Doctor Visual Est. | |
|---|---|---|---|---|---|---|---|
| | | Predicted | Abs. Error | Predicted | Abs. Error | Predicted | Abs. Error |
| P1 | 11.55 | 13.46 | 1.92 | 19.06 | 7.51 | 9.26 | 2.29 |
| P2 | 2.11 | 1.95 | 0.16 | 2.98 | 0.87 | 0.87 | 1.24 |
| P3 | 10.04 | 10.02 | 0.02 | 14.53 | 3.49 | 0.72 | 9.32 |
| P4 | 20.52 | 17.42 | 3.10 | 25.82 | 5.30 | 8.14 | 12.39 |
| P5 | 4.51 | 1.25 | 3.26 | 2.28 | 2.23 | 3.27 | 1.24 |
| P6 | 11.00 | 3.84 | 7.15 | 6.93 | 4.07 | 15.34 | 4.34 |
| P7 | 2.47 | 13.19 | 10.72 | 18.41 | 15.94 | 5.33 | 2.86 |
| P8 | 14.74 | 13.34 | 1.40 | 19.84 | 5.10 | 6.97 | 7.77 |
| P9 | 20.04 | 3.10 | 16.94 | 4.96 | 15.08 | 30.21 | 10.17 |
| P10 | 19.97 | 18.81 | 1.16 | 33.39 | 13.42 | 12.63 | 7.35 |
| P11 | 3.51 | 15.80 | 12.29 | 22.62 | 19.11 | 9.88 | 6.37 |
| P12 | 6.39 | 1.17 | 5.22 | 2.83 | 3.56 | 4.43 | 1.97 |
| P13 | 8.66 | 13.09 | 4.43 | 20.44 | 11.78 | 8.00 | 0.66 |
| P14 | 1.79 | 5.47 | 3.68 | 6.62 | 4.83 | 4.55 | 2.76 |
| P15 | 11.54 | 9.45 | 2.09 | 15.46 | 3.92 | 4.55 | 6.99 |
| P16 | 8.30 | 7.89 | 0.41 | 12.32 | 4.02 | 7.50 | 0.80 |
| P17 | 16.90 | 16.97 | 0.07 | 24.75 | 7.85 | 12.00 | 4.90 |
| P18 | 2.44 | 16.32 | 13.89 | 24.41 | 21.98 | 9.45 | 7.01 |
| P19 | 7.10 | 2.00 | 5.10 | 3.48 | 3.62 | 1.73 | 5.38 |
| P20 | 4.88 | 7.50 | 2.62 | 11.14 | 6.26 | 3.30 | 1.58 |
| P21 | 12.98 | 4.06 | 8.92 | 6.63 | 6.34 | 2.15 | 10.83 |
| P22 | 11.98 | 11.13 | 0.85 | 20.62 | 8.64 | 8.95 | 3.03 |
| P23 | 10.90 | 12.06 | 1.15 | 19.77 | 8.87 | 6.95 | 3.95 |
| P24 | 5.30 | 10.32 | 5.02 | 15.50 | 10.20 | 9.10 | 3.80 |
| P25 | 10.18 | 5.34 | 4.83 | 8.55 | 1.62 | 2.45 | 7.73 |
| P26 | 19.52 | 17.02 | 2.50 | 26.09 | 6.57 | 11.00 | 8.52 |
| P27 | 9.53 | 3.59 | 5.94 | 5.94 | 3.59 | 3.15 | 6.38 |
| P28 | 3.79 | 8.36 | 4.56 | 13.35 | 9.56 | 5.00 | 1.21 |
| P29 | 4.51 | 4.59 | 0.08 | 7.30 | 2.79 | 3.45 | 1.06 |
| P30 | 10.04 | 9.48 | 0.56 | 16.48 | 6.44 | 7.55 | 2.49 |
| **Average** | **9.57** | **9.27** | **4.34** | **14.42** | **7.52** | **7.26** | **4.88** |

Discussion for Virtual Dataset: In Table 16, the former configuration achieves an average MAE of 4.34mm, indicating that even when depth is accurate, segmentation inaccuracies (e.g., partial boundaries or over-segmentation) can significantly affect final measurements. In the latter configuration, the MAE increases to 7.52mm, confirming that depth estimation remains a key bottleneck, introducing geometric distortions even under ideal segmentation conditions. Overall, this analysis highlights the compounded nature of perception errors in polyp size estimation. While the virtual environment offers dense supervision and noise-free geometry, it also reveals that inaccuracies in either segmentation or depth can propagate downstream, emphasizing the importance of optimizing both components in real-world deployments.

**Physical Dataset**: Polyp size ground truth labels are derived from the known 3D geometry of printed polyp models. We evaluate two configurations:

- Seg(GT) + Depth(Pred): Uses manually annotated segmentation masks combined with depth predictions from DAM V2.

- Seg(Pred) + Depth(Pred): Uses predicted segmentation (SAM (ViT-H)) and predicted depth (DAM V2) to compute size.

Table 17: Benchmark on 3D-Printed Dataset: Per-Polyp Size Estimation (mm)

| Polyp ID | Label | Seg(GT)+Depth(Pred) | | Seg(Pred)+Depth(Pred) | |
|---|---|---|---|---|---|
| | | Predicted | Abs. Error | Predicted | Abs. Error |
| P1 | 6.20 | 6.22 | 0.02 | 8.46 | 2.26 |
| P2 | 8.01 | 8.27 | 0.26 | 11.28 | 3.26 |
| P3 | 4.06 | 4.64 | 0.58 | 6.29 | 2.23 |
| P4 | 13.26 | 7.59 | 5.67 | 10.07 | 3.19 |
| P5 | 5.65 | 4.09 | 1.56 | 6.98 | 1.33 |
| P6 | 10.50 | 6.96 | 3.54 | 9.19 | 1.31 |
| P7 | 4.12 | 3.23 | 0.89 | - | - |
| P8 | 12.99 | 11.75 | 1.24 | 12.86 | 0.13 |
| P9 | 4.00 | 4.58 | 0.58 | 7.11 | 3.11 |
| P10 | 6.45 | 6.17 | 0.28 | 8.30 | 1.85 |
| P11 | 14.89 | 12.61 | 2.28 | 14.33 | 0.56 |
| P12 | 9.24 | 9.74 | 0.50 | 9.88 | 0.64 |
| P13 | 4.94 | 6.74 | 1.80 | 8.89 | 3.95 |
| Average | 8.03 | 8.38 | 1.48 | 9.48 | 1.99 |

Discussion for Physical Dataset: The physical phantom evaluation (Table 17) shows an MAE of 1.48mm when using ground truth segmentation and predicted depth. The fully automated pipeline (predicted segmentation and depth) resulted in an MAE of 1.99mm. This indicates that the sim-to-real gap affects both segmentation and depth estimation. The errors are larger than in the virtual setting but demonstrate the feasibility of applying these methods to real-world textures and lighting, albeit with reduced accuracy. The P7 segmentation failure in automated mode highlights challenges with specific material properties or polyp appearances in the phantom.

**Real Clinical Dataset:** Polyp size ground truth (Label (Forceps)) was generated via the biopsy forceps calibration strategy (Section C.3). We evaluated three configurations:

- Seg(GT) + Depth(Pred): Manually refined segmentation masks with depth predictions from DAM V1.
- Seg(Pred) + Depth(Pred): Predicted segmentation (PraNet) and predicted depth (DAM V1).
- Doctor Visual Estimation: Average of four clinicians' visual estimates.

Discussion for Clinical Dataset: The clinical data (Table 18), shows an MAE of 1.19mm for GT segmentation with predicted depth, and an MAE of 0.95mm for the fully automated pipeline. This relatively strong performance in the fully automated setting for small polyps suggests that errors in segmentation and depth estimation might partially compensate for each other in some cases, or that the chosen models generalize reasonably well to these challenging, small lesions. Physician visual estimates had an MAE of 1.84mm, indicating that the automated methods are competitive for this specific task of small polyp sizing. The results underscore the dataset's utility for developing algorithms robust to real-world clinical noise, specularities, and subtle polyp features.

To investigate the potential systematic bias of using the bounding box's center depth ($Z_{center}$), we implemented and evaluated the more complex 5-point average method, inspired by recent clinical work[53]. Our new experiments revealed that, the original $Z_{center}$ method yielded better results, achieving a Mean Absolute Error (MAE) of 1.08 mm compared to 1.44 mm for the 5-point method Table 19.

While the 5-point method appears more comprehensive, it is more susceptible to noise from the upstream perception modules. The four extremal points (top, bottom, left, right) often lie on the curved boundary of the polyp where: the segmentation mask can be imprecise, causing these points to fall on the background tissue, and the predicted depth map is often least accurate at object edges and sharp discontinuities. Averaging in these potentially erroneous depth values from the unstable boundary can corrupt the final estimate. In contrast, $Z_{center}$ typically lies on the flat, central region of the polyp, offering more stable and reliable estimates.

### G.2 Qualitative Results and Failure Case Analysis

Qualitative examples of segmentation performance are shown in Figure 30. While models like PraNet and SAM generally perform well on clear, well-defined polyps, challenges arise with:

Table 18: Benchmark on Clinical Dataset: Per-Case Size Estimation (mm)

| Polyp ID | Label (Forceps) | Seg(GT)+Depth(Pred) | | Seg(Pred)+Depth(Pred) | | Doctor Visual Est.(Avg) | |
|---|---|---|---|---|---|---|---|
| | | Predicted | Abs. Error | Predicted | Abs. Error | Predicted | Abs. Error |
| P1 | 2.31 | 2.19 | 0.12 | 2.38 | 0.07 | 3.43 | 1.12 |
| P2 | 2.34 | 3.31 | 0.97 | 3.35 | 1.01 | 3.95 | 1.61 |
| P3 | 3.88 | 3.50 | 0.38 | 3.39 | 0.49 | 4.75 | 0.87 |
| P4 | 2.60 | 2.86 | 0.26 | 2.67 | 0.07 | 2.08 | 0.52 |
| P5 | 3.80 | 2.66 | 1.14 | 2.50 | 1.30 | 1.83 | 1.97 |
| P6 | 2.30 | 3.58 | 1.28 | 2.40 | 0.10 | 4.65 | 2.35 |
| P7 | 1.86 | 2.36 | 0.50 | 1.66 | 0.20 | 2.30 | 0.44 |
| P8 | 3.15 | 2.11 | 1.04 | 1.63 | 1.52 | 2.20 | 0.95 |
| P9 | 2.05 | 1.77 | 0.28 | 1.69 | 0.36 | 2.48 | 0.43 |
| P10 | 3.46 | 3.03 | 0.43 | 3.47 | 0.01 | 6.05 | 2.59 |
| P11 | 1.96 | 2.27 | 0.31 | 2.00 | 0.04 | 4.00 | 2.04 |
| P12 | 2.24 | 3.27 | 1.03 | 2.23 | 0.01 | 3.28 | 1.04 |
| P13 | 1.39 | 1.62 | 0.23 | 1.50 | 0.11 | 1.80 | 0.41 |
| P14 | 3.28 | 1.73 | 1.55 | 2.17 | 1.11 | 4.20 | 0.92 |
| P15 | 3.04 | 1.77 | 1.27 | 2.03 | 1.01 | 3.73 | 0.69 |
| P16 | 2.82 | 2.03 | 0.79 | 2.00 | 0.82 | 3.13 | 0.31 |
| P17 | 7.87 | 3.18 | 4.69 | 3.31 | 4.56 | 5.00 | 2.87 |
| P18 | 4.52 | 5.12 | 0.60 | 5.02 | 0.50 | 7.88 | 3.36 |
| P19 | 2.00 | 1.67 | 0.33 | 1.25 | 0.75 | 3.58 | 1.58 |
| P20 | 2.14 | 3.53 | 1.39 | 2.18 | 0.04 | 3.50 | 1.36 |
| P21 | 5.38 | 2.99 | 2.39 | 3.11 | 2.27 | 3.30 | 2.08 |
| P22 | 3.23 | 1.95 | 1.28 | 2.18 | 1.05 | 4.00 | 0.77 |
| P23 | 1.39 | 2.52 | 1.13 | 2.34 | 0.95 | 6.33 | 4.94 |
| P24 | 3.45 | 1.77 | 1.68 | 2.19 | 1.26 | 5.00 | 1.55 |
| P25 | 1.75 | 2.23 | 0.48 | 2.15 | 0.40 | 3.33 | 1.58 |
| P26 | 2.46 | 2.10 | 0.36 | 1.93 | 0.53 | 4.38 | 1.92 |
| P27 | 3.18 | 2.41 | 0.77 | 2.20 | 0.98 | 3.60 | 0.42 |
| P28 | 5.01 | 2.55 | 2.46 | 3.06 | 1.95 | 5.13 | 0.12 |
| P29 | 2.07 | 3.00 | 0.93 | 2.56 | 0.49 | 5.25 | 3.18 |
| P30 | 1.57 | 1.57 | 0.00 | 1.29 | 0.28 | 2.33 | 0.76 |
| P31 | 2.04 | 1.78 | 0.26 | 2.01 | 0.03 | 4.88 | 2.84 |
| P32 | 1.61 | 6.47 | 4.86 | 4.82 | 3.21 | 5.83 | 4.22 |
| P33 | 1.88 | 2.47 | 0.59 | 1.90 | 0.02 | 5.13 | 3.25 |
| P34 | 2.88 | 5.26 | 2.38 | 1.61 | 1.27 | 4.43 | 1.55 |
| P35 | 2.46 | 4.10 | 1.64 | 2.48 | 0.02 | 4.40 | 1.94 |
| P36 | 2.41 | 2.83 | 0.42 | 2.28 | 0.13 | 2.48 | 0.07 |
| P37 | 3.03 | 2.21 | 0.82 | 3.00 | 0.03 | 4.48 | 1.45 |
| P38 | 3.09 | 3.12 | 0.03 | 3.35 | 0.26 | 7.30 | 4.21 |
| P39 | 4.79 | 1.75 | 3.04 | 1.54 | 3.25 | 6.08 | 1.29 |
| P40 | 4.47 | 2.32 | 2.15 | 1.72 | 2.75 | 2.48 | 1.99 |
| P41 | 4.28 | 7.07 | 2.79 | 4.20 | 0.08 | 10.50 | 6.22 |
| P42 | 3.45 | 1.49 | 1.96 | 1.47 | 1.98 | 5.30 | 1.85 |
| P43 | 6.07 | 4.24 | 1.83 | 3.76 | 2.31 | 4.38 | 1.69 |
| P44 | 3.84 | 0.87 | 2.97 | 0.95 | 2.89 | 1.85 | 1.99 |
| P45 | 2.21 | 2.86 | 0.65 | 2.62 | 0.41 | 3.13 | 0.92 |
| P46 | 1.83 | 1.72 | 0.11 | 0.68 | 1.15 | 2.73 | 0.90 |
| P47 | 3.65 | 1.56 | 2.09 | 1.73 | 1.92 | 2.20 | 1.45 |
| P48 | 2.00 | 3.49 | 1.49 | 2.73 | 0.73 | 4.25 | 2.25 |
| P49 | 2.93 | 3.81 | 0.88 | 3.28 | 0.35 | 4.03 | 1.10 |
| P50 | 2.68 | 4.06 | 1.38 | 2.64 | 0.04 | 2.33 | 0.35 |
| P51 | 7.74 | 5.68 | 2.06 | 6.21 | 1.53 | 10.83 | 3.09 |
| P52 | 9.98 | 8.91 | 1.07 | 10.26 | 0.28 | 19.33 | 9.35 |
| P53 | 5.95 | 5.99 | 0.04 | 4.22 | 1.73 | 8.67 | 2.72 |
| **Average** | **3.28** | **2.80** | **1.19** | **2.66** | **0.95** | **4.73** | **1.84** |

Table 19: Polyp Size Estimation Using the 5-Point Average Method (mm)

| Polyp ID | Label (Forceps) | Seg(GT)+Depth(Pred) | | Seg(Pred)+Depth(Pred) | |
|---|---|---|---|---|---|
| | | Predicted | Abs. Error | Predicted | Abs. Error |
| P1 | 2.31 | 2.60 | 0.29 | 2.45 | 0.15 |
| P2 | 2.34 | 3.35 | 1.01 | 3.39 | 1.06 |
| P3 | 3.88 | 4.40 | 0.52 | 4.26 | 0.39 |
| P4 | 2.60 | 2.82 | 0.22 | 3.14 | 0.54 |
| P5 | 3.80 | 3.30 | 0.50 | 3.08 | 0.72 |
| P6 | 2.30 | 3.95 | 1.65 | 2.48 | 0.18 |
| P7 | 1.86 | 2.91 | 1.05 | 1.64 | 0.22 |
| P8 | 3.15 | 2.72 | 0.43 | 1.92 | 1.23 |
| P9 | 2.05 | 1.77 | 0.28 | 2.07 | 0.02 |
| P10 | 3.46 | 3.94 | 0.48 | 3.52 | 0.06 |
| P11 | 1.96 | 2.13 | 0.17 | 2.01 | 0.05 |
| P12 | 2.24 | 3.03 | 0.79 | 2.23 | 0.01 |
| P13 | 1.39 | 1.46 | 0.07 | 1.50 | 0.11 |
| P14 | 3.28 | 1.84 | 1.44 | 2.27 | 1.01 |
| P15 | 3.04 | 1.93 | 1.12 | 2.11 | 0.93 |
| P16 | 2.82 | 1.99 | 0.83 | 1.99 | 0.83 |
| P17 | 7.87 | 3.90 | 3.97 | 4.02 | 3.85 |
| P18 | 4.52 | 6.56 | 2.03 | 5.76 | 1.24 |
| P19 | 2.00 | 2.06 | 0.06 | 1.53 | 0.47 |
| P20 | 2.14 | 4.21 | 2.07 | 2.54 | 0.40 |
| P21 | 5.38 | 2.93 | 2.45 | 3.07 | 2.31 |
| P22 | 3.23 | 2.35 | 0.88 | 2.50 | 0.73 |
| P23 | 1.39 | 2.47 | 1.09 | 2.36 | 0.98 |
| P24 | 3.45 | 2.02 | 1.44 | 2.81 | 0.65 |
| P25 | 1.75 | 2.22 | 0.47 | 2.19 | 0.44 |
| P26 | 2.46 | 2.12 | 0.34 | 1.94 | 0.52 |
| P27 | 3.18 | 2.44 | 0.74 | 2.20 | 0.98 |
| P28 | 5.01 | 2.58 | 2.43 | 3.05 | 1.96 |
| P29 | 2.06 | 2.86 | 0.79 | 2.55 | 0.49 |
| P30 | 1.58 | 1.57 | 0.01 | 1.38 | 0.20 |
| P31 | 2.04 | 2.38 | 0.33 | 2.16 | 0.11 |
| P32 | 1.62 | 6.02 | 4.40 | 5.49 | 3.87 |
| P33 | 1.88 | 2.54 | 0.66 | 1.90 | 0.02 |
| P34 | 2.88 | 5.10 | 2.22 | 1.61 | 1.27 |
| P35 | 2.46 | 4.13 | 1.67 | 2.52 | 0.06 |
| P36 | 2.41 | 2.73 | 0.33 | 2.31 | 0.10 |
| P37 | 3.03 | 2.83 | 0.20 | 2.98 | 0.05 |
| P38 | 3.09 | 3.84 | 0.76 | 3.60 | 0.51 |
| P39 | 4.79 | 2.28 | 2.50 | 1.63 | 3.15 |
| P40 | 4.47 | 2.33 | 2.14 | 2.14 | 2.33 |
| P41 | 4.28 | 7.59 | 3.31 | 5.19 | 0.92 |
| P42 | 3.45 | 2.22 | 1.23 | 1.89 | 1.56 |
| P43 | 6.07 | 5.10 | 0.97 | 3.70 | 2.37 |
| P44 | 3.84 | 1.26 | 2.58 | 1.07 | 2.76 |
| P45 | 2.21 | 2.88 | 0.67 | 2.66 | 0.45 |
| P46 | 1.83 | 1.72 | 0.11 | 0.78 | 1.05 |
| P47 | 3.65 | 2.04 | 1.61 | 2.07 | 1.57 |
| P48 | 2.00 | 4.02 | 2.02 | 2.96 | 0.96 |
| P49 | 2.93 | 3.61 | 0.68 | 3.20 | 0.27 |
| P50 | 2.67 | 3.89 | 1.22 | 3.25 | 0.59 |
| P51 | 7.74 | 5.74 | 2.00 | 6.33 | 1.41 |
| P52 | 9.98 | 8.82 | 1.16 | 8.72 | 1.26 |
| P53 | 5.95 | 6.11 | 0.16 | 4.54 | 1.41 |
| **Average** | **3.28** | **3.27** | **1.18** | **2.83** | **0.95** |

- **Small or Flat Polyps** (column 11 in Fig.30): These can be missed or poorly delineated, especially if their texture is very similar to the surrounding mucosa (e.g., VM-UNet on some clinical cases).

- **Occlusion and Specular Highlights** (column 4 Fig.30): Partial occlusion by folds or instruments, and strong specular reflections, can lead to incomplete or fragmented masks. MobileSAM, being lightweight, can be more susceptible.

- **Motion/Fluid Blur**(column 12 Fig.30): Clinical videos often retain residual camera motion or rapid scope withdrawal, and intraluminal fluid further smears edges. These factors soften boundaries and cause over-smoothed predictions across models.

- **Domain Shift:** As evident in Table 3, performance drops from virtual to physical, and further to clinical, indicating that visual features learned in simulation do not perfectly transfer.

For depth estimation, common failure modes include:

- **Textureless Regions** (column 7 Fig.32): Monocular depth cues are weak in uniformly textured areas, leading to smoothed or inaccurate depth.

- **Scale Ambiguity:** Training and fine-tuning on datasets with metric ground truth (Tables 10, 5) constrain our models to absolute, millimetre-level accuracy. Nevertheless, small scale errors still emerge when the target polyp lies far from the calibration surrogate—typically the opened forceps—causing slight under/over-estimation of true size. As shown in [**?** ]

- **Non-Lambertian Surfaces** (row 3 Fig.31): The shiny, mucous surface of the colon violates Lambertian assumptions, affecting photometric consistency if used by self-supervised methods, and can confuse supervised methods trained primarily on matte surfaces.

Error propagation to size estimation is direct:

- An overestimated segmentation mask combined with overestimated depth will lead to a significantly overestimated size.

- An underestimated mask with underestimated depth might coincidentally yield an accurate size, but for the wrong reasons.

- The choice of which dimension of the bounding box (major, minor, or average) to use as "size" also impacts results, especially for elongated or irregularly shaped polyps. Our use of the maximum dimension is a common convention but can be sensitive to segmentation orientation.

The clinical dataset, with its small polyps, is particularly sensitive to small absolute errors in segmentation or depth, as these can translate to large relative errors in size. For instance, a 1mm error on a 3mm polyp is a 33% relative error. Future work should focus on robust uncertainty quantification alongside point estimates for size to better inform clinical decision-making.

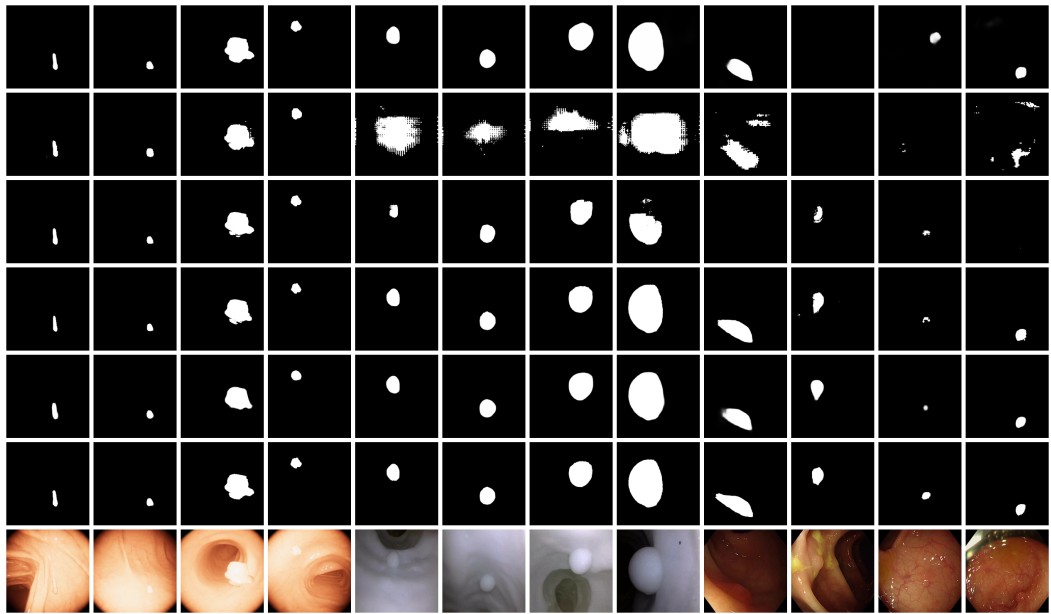

Figure 30: Visual comparison of polyp segmentation results on the virtual dataset. From bottom to top: RGB image, ground truth mask, and predictions from MSNet, PraNet, SAM, MobileSAM, and VM-Unet.

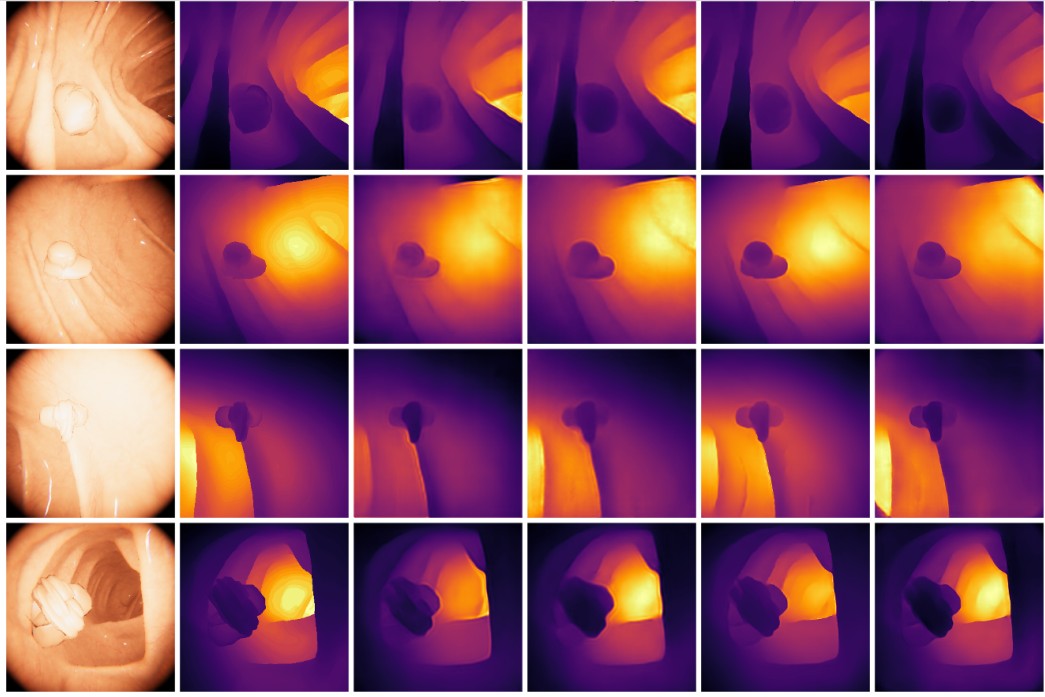

Figure 31: Visual comparison of depth estimation results on the virtual dataset. From left to right: RGB image, ground truth depth, and predictions from DAM V1, DAM V2, ZoeDepth, and DAM V2-mini.

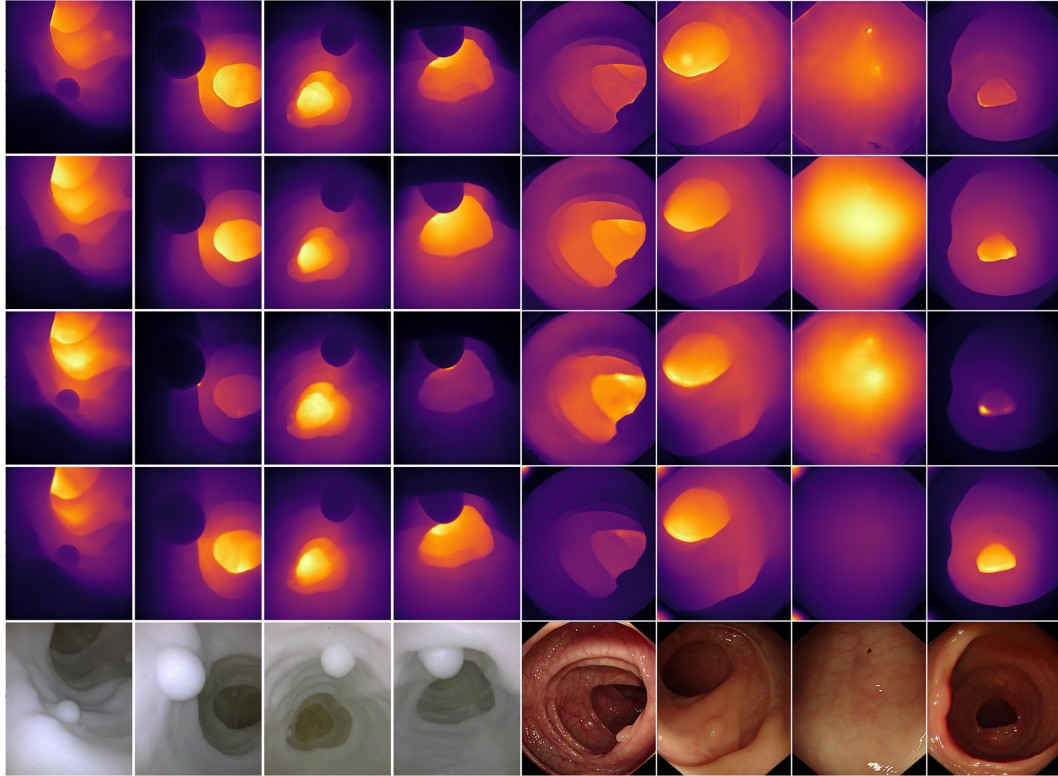

Figure 32: Visual comparison of depth estimation results on the physical and clinical dataset. From bottom to top: RGB image, and predictions from DAM V1, DAM V2, ZoeDepth, and DAM V2-mini.

## NeurIPS Paper Checklist

1. **Claims**

   Question: Do the main claims made in the abstract and introduction accurately reflect the paper's contributions and scope?

   Answer: [Yes]

   Justification: The abstract and introduction clearly state the main claims, including the introduction of the PolypSense3D dataset, the novel in-vivo annotation protocol, the comprehensive benchmarking, and the release of open resources. These claims are supported by the content of the paper, which details the dataset construction, the annotation method, the experiments, and the data/code release plans.

   Guidelines:

   - The answer NA means that the abstract and introduction do not include the claims made in the paper.
   - The abstract and/or introduction should clearly state the claims made, including the contributions made in the paper and important assumptions and limitations. A No or NA answer to this question will not be perceived well by the reviewers.
   - The claims made should match theoretical and experimental results, and reflect how much the results can be expected to generalize to other settings.
   - It is fine to include aspirational goals as motivation as long as it is clear that these goals are not attained by the paper.

2. **Limitations**

   Question: Does the paper discuss the limitations of the work performed by the authors?

   Answer: [Yes]

Justification: The paper discusses several limitations, including the realism gap between virtual and real data, the simplification of tissue in the physical phantom, the sparse depth information and potential variability in the clinical annotation method, and potential biases due to the specific clinical context. These limitations are detailed in the Discussion section of the paper.

Guidelines:

- The answer NA means that the paper has no limitation while the answer No means that the paper has limitations, but those are not discussed in the paper.
- The authors are encouraged to create a separate "Limitations" section in their paper.
- The paper should point out any strong assumptions and how robust the results are to violations of these assumptions (e.g., independence assumptions, noiseless settings, model well-specification, asymptotic approximations only holding locally). The authors should reflect on how these assumptions might be violated in practice and what the implications would be.
- The authors should reflect on the scope of the claims made, e.g., if the approach was only tested on a few datasets or with a few runs. In general, empirical results often depend on implicit assumptions, which should be articulated.
- The authors should reflect on the factors that influence the performance of the approach. For example, a facial recognition algorithm may perform poorly when image resolution is low or images are taken in low lighting. Or a speech-to-text system might not be used reliably to provide closed captions for online lectures because it fails to handle technical jargon.
- The authors should discuss the computational efficiency of the proposed algorithms and how they scale with dataset size.
- If applicable, the authors should discuss possible limitations of their approach to address problems of privacy and fairness.
- While the authors might fear that complete honesty about limitations might be used by reviewers as grounds for rejection, a worse outcome might be that reviewers discover limitations that aren't acknowledged in the paper. The authors should use their best judgment and recognize that individual actions in favor of transparency play an important role in developing norms that preserve the integrity of the community. Reviewers will be specifically instructed to not penalize honesty concerning limitations.

3. **Theory assumptions and proofs**

Question: For each theoretical result, does the paper provide the full set of assumptions and a complete (and correct) proof?

Answer: [No]

Justification: The paper focuses on dataset creation and empirical evaluation. While it includes equations related to depth map captureand size estimation, these are descriptions of established geometric principles rather than novel theoretical results requiring formal proofs. The paper does not present new theorems or lemmas that necessitate proofs.

Guidelines:

- The answer NA means that the paper does not include theoretical results.
- All the theorems, formulas, and proofs in the paper should be numbered and cross-referenced.
- All assumptions should be clearly stated or referenced in the statement of any theorems.
- The proofs can either appear in the main paper or the supplemental material, but if they appear in the supplemental material, the authors are encouraged to provide a short proof sketch to provide intuition.
- Inversely, any informal proof provided in the core of the paper should be complemented by formal proofs provided in appendix or supplemental material.
- Theorems and Lemmas that the proof relies upon should be properly referenced.

4. **Experimental result reproducibility**

Question: Does the paper fully disclose all the information needed to reproduce the main experimental results of the paper to the extent that it affects the main claims and/or conclusions of the paper (regardless of whether the code and data are provided or not)?

Answer: [Yes]

Justification: The paper prioritizes reproducibility by providing comprehensive details regarding the experimental methodology. This includes clear definitions of the tasks and evaluation metrics employed, thorough descriptions of the baseline models with appropriate references and implementation specifics, and detailed accounts of the training and fine-tuning procedures, covering hyperparameters, optimizers, loss functions, and data augmentation techniques. Furthermore, the hardware specifications used for the experiments are outlined, and extensive information is given on the dataset itself, encompassing its statistics and the data processing and size estimation pipeline. The combination of this detailed methodological information and the planned release of code and data ensures a solid foundation for reproducing the reported results.

Guidelines:

- The answer NA means that the paper does not include experiments.
- If the paper includes experiments, a No answer to this question will not be perceived well by the reviewers: Making the paper reproducible is important, regardless of whether the code and data are provided or not.
- If the contribution is a dataset and/or model, the authors should describe the steps taken to make their results reproducible or verifiable.
- Depending on the contribution, reproducibility can be accomplished in various ways. For example, if the contribution is a novel architecture, describing the architecture fully might suffice, or if the contribution is a specific model and empirical evaluation, it may be necessary to either make it possible for others to replicate the model with the same dataset, or provide access to the model. In general. releasing code and data is often one good way to accomplish this, but reproducibility can also be provided via detailed instructions for how to replicate the results, access to a hosted model (e.g., in the case of a large language model), releasing of a model checkpoint, or other means that are appropriate to the research performed.
- While NeurIPS does not require releasing code, the conference does require all submissions to provide some reasonable avenue for reproducibility, which may depend on the nature of the contribution. For example
  (a) If the contribution is primarily a new algorithm, the paper should make it clear how to reproduce that algorithm.
  (b) If the contribution is primarily a new model architecture, the paper should describe the architecture clearly and fully.
  (c) If the contribution is a new model (e.g., a large language model), then there should either be a way to access this model for reproducing the results or a way to reproduce the model (e.g., with an open-source dataset or instructions for how to construct the dataset).
  (d) We recognize that reproducibility may be tricky in some cases, in which case authors are welcome to describe the particular way they provide for reproducibility. In the case of closed-source models, it may be that access to the model is limited in some way (e.g., to registered users), but it should be possible for other researchers to have some path to reproducing or verifying the results.

5. **Open access to data and code**

Question: Does the paper provide open access to the data and code, with sufficient instructions to faithfully reproduce the main experimental results, as described in supplemental material?

Answer: [Yes]

Justification: The paper provides open access to both the dataset and source code. The dataset is publicly hosted at Harvard Dataverse(`https://doi.org/10.7910/DVN/LKDIEK`), and the codebase is accompanied by detailed documentation and scripts in the supplemental material. Reproduction instructions include environment setup, dependencies, data access steps, and command-line examples to run polyp size measurement experiments are available at:`https://github.com/HNUicda/PolypSense3D`. The release covers raw and processed data, along with all necessary annotations. These resources enable reviewers and readers to faithfully reproduce the main results reported in the paper.

Guidelines:

- The answer NA means that paper does not include experiments requiring code.
- Please see the NeurIPS code and data submission guidelines (`https://nips.cc/public/guides/CodeSubmissionPolicy`) for more details.
- While we encourage the release of code and data, we understand that this might not be possible, so "No" is an acceptable answer. Papers cannot be rejected simply for not including code, unless this is central to the contribution (e.g., for a new open-source benchmark).
- The instructions should contain the exact command and environment needed to run to reproduce the results. See the NeurIPS code and data submission guidelines (`https://nips.cc/public/guides/CodeSubmissionPolicy`) for more details.
- The authors should provide instructions on data access and preparation, including how to access the raw data, preprocessed data, intermediate data, and generated data, etc.
- The authors should provide scripts to reproduce all experimental results for the new proposed method and baselines. If only a subset of experiments are reproducible, they should state which ones are omitted from the script and why.
- At submission time, to preserve anonymity, the authors should release anonymized versions (if applicable).
- Providing as much information as possible in supplemental material (appended to the paper) is recommended, but including URLs to data and code is permitted.

6. **Experimental setting/details**

   Question: Does the paper specify all the training and test details (e.g., data splits, hyperparameters, how they were chosen, type of optimizer, etc.) necessary to understand the results?

   Answer: [Yes]

   Justification: The paper details the experimental setup in Section 5.1 and Appendix E, specifying tasks, metrics, baseline models, and training procedures. Key details include data splits, hyperparameters, optimizer types, and their selection.

   Guidelines:

   - The answer NA means that the paper does not include experiments.
   - The experimental setting should be presented in the core of the paper to a level of detail that is necessary to appreciate the results and make sense of them.
   - The full details can be provided either with the code, in appendix, or as supplemental material.

7. **Experiment statistical significance**

   Question: Does the paper report error bars suitably and correctly defined or other appropriate information about the statistical significance of the experiments?

   Answer: [Yes]

   Justification: While the paper does not consistently report error bars or confidence intervals, it does provide relevant quantitative metrics for the segmentation, depth estimation, and polyp size measurement tasks. These different metrics (e.g., mDice, mIoU, RMSE, MAE) offer various perspectives on the algorithms' performance and error characteristics, allowing for a detailed analysis of result variability and significance, even without explicit statistical significance tests.

   Guidelines:

   - The answer NA means that the paper does not include experiments.
   - The authors should answer "Yes" if the results are accompanied by error bars, confidence intervals, or statistical significance tests, at least for the experiments that support the main claims of the paper.
   - The factors of variability that the error bars are capturing should be clearly stated (for example, train/test split, initialization, random drawing of some parameter, or overall run with given experimental conditions).

- The method for calculating the error bars should be explained (closed form formula, call to a library function, bootstrap, etc.)
- The assumptions made should be given (e.g., Normally distributed errors).
- It should be clear whether the error bar is the standard deviation or the standard error of the mean.
- It is OK to report 1-sigma error bars, but one should state it. The authors should preferably report a 2-sigma error bar than state that they have a 96% CI, if the hypothesis of Normality of errors is not verified.
- For asymmetric distributions, the authors should be careful not to show in tables or figures symmetric error bars that would yield results that are out of range (e.g. negative error rates).
- If error bars are reported in tables or plots, The authors should explain in the text how they were calculated and reference the corresponding figures or tables in the text.

8. **Experiments compute resources**

Question: For each experiment, does the paper provide sufficient information on the computer resources (type of compute workers, memory, time of execution) needed to reproduce the experiments?

Answer: [Yes]

Justification: The paper provides information on the GPU type used (NVIDIA RTX 4090) in Appendix that specifies the number of GPUs used. While it doesn't give precise execution times for every experiment, it does provide inference times for the models, which gives some indication of computational cost.

Guidelines:

- The answer NA means that the paper does not include experiments.
- The paper should indicate the type of compute workers CPU or GPU, internal cluster, or cloud provider, including relevant memory and storage.
- The paper should provide the amount of compute required for each of the individual experimental runs as well as estimate the total compute.
- The paper should disclose whether the full research project required more compute than the experiments reported in the paper (e.g., preliminary or failed experiments that didn't make it into the paper).

9. **Code of ethics**

Question: Does the research conducted in the paper conform, in every respect, with the NeurIPS Code of Ethics `https://neurips.cc/public/EthicsGuidelines`?

Answer: [Yes]

Justification: The research conducted in this paper adheres to the NeurIPS Code of Ethics. This commitment is reflected in several key aspects of the work, including the rigorous ethical review and approval obtained for the collection of clinical data, the stringent anonymization procedures implemented to protect patient privacy, the explicit acknowledgment of the dataset's limitations and its intended use solely for research, precluding direct clinical application, and the transparent discussion of potential biases that could influence its interpretation and use.

Guidelines:

- The answer NA means that the authors have not reviewed the NeurIPS Code of Ethics.
- If the authors answer No, they should explain the special circumstances that require a deviation from the Code of Ethics.
- The authors should make sure to preserve anonymity (e.g., if there is a special consideration due to laws or regulations in their jurisdiction).

10. **Broader impacts**

Question: Does the paper discuss both potential positive societal impacts and negative societal impacts of the work performed?

Answer: [Yes]

Justification: The paper discusses the potential positive societal impact of enabling more accurate and objective polyp size measurement, which can improve clinical decision-making and patient care in colorectal cancer screening (Introduction Section). It also acknowledges the potential negative impact of the dataset being misused for direct clinical diagnosis or decision-making without proper validation, which could lead to harm (Ethics Statement Section 9).

Guidelines:

- The answer NA means that there is no societal impact of the work performed.
- If the authors answer NA or No, they should explain why their work has no societal impact or why the paper does not address societal impact.
- Examples of negative societal impacts include potential malicious or unintended uses (e.g., disinformation, generating fake profiles, surveillance), fairness considerations (e.g., deployment of technologies that could make decisions that unfairly impact specific groups), privacy considerations, and security considerations.
- The conference expects that many papers will be foundational research and not tied to particular applications, let alone deployments. However, if there is a direct path to any negative applications, the authors should point it out. For example, it is legitimate to point out that an improvement in the quality of generative models could be used to generate deepfakes for disinformation. On the other hand, it is not needed to point out that a generic algorithm for optimizing neural networks could enable people to train models that generate Deepfakes faster.
- The authors should consider possible harms that could arise when the technology is being used as intended and functioning correctly, harms that could arise when the technology is being used as intended but gives incorrect results, and harms following from (intentional or unintentional) misuse of the technology.
- If there are negative societal impacts, the authors could also discuss possible mitigation strategies (e.g., gated release of models, providing defenses in addition to attacks, mechanisms for monitoring misuse, mechanisms to monitor how a system learns from feedback over time, improving the efficiency and accessibility of ML).

11. **Safeguards**

Question: Does the paper describe safeguards that have been put in place for responsible release of data or models that have a high risk for misuse (e.g., pretrained language models, image generators, or scraped datasets)?

Answer: [NA]

Justification: This paper does not involve models or data with high risk for misuse.

Guidelines:

- The answer NA means that the paper poses no such risks.
- Released models that have a high risk for misuse or dual-use should be released with necessary safeguards to allow for controlled use of the model, for example by requiring that users adhere to usage guidelines or restrictions to access the model or implementing safety filters.
- Datasets that have been scraped from the Internet could pose safety risks. The authors should describe how they avoided releasing unsafe images.
- We recognize that providing effective safeguards is challenging, and many papers do not require this, but we encourage authors to take this into account and make a best faith effort.

12. **Licenses for existing assets**

Question: Are the creators or original owners of assets (e.g., code, data, models), used in the paper, properly credited and are the license and terms of use explicitly mentioned and properly respected?

Answer: [Yes]

Justification: The paper properly credits and cites the open-source data, algorithms, and tools utilized.

Guidelines:

- The answer NA means that the paper does not use existing assets.
- The authors should cite the original paper that produced the code package or dataset.
- The authors should state which version of the asset is used and, if possible, include a URL.
- The name of the license (e.g., CC-BY 4.0) should be included for each asset.
- For scraped data from a particular source (e.g., website), the copyright and terms of service of that source should be provided.
- If assets are released, the license, copyright information, and terms of use in the package should be provided. For popular datasets, `paperswithcode.com/datasets` has curated licenses for some datasets. Their licensing guide can help determine the license of a dataset.
- For existing datasets that are re-packaged, both the original license and the license of the derived asset (if it has changed) should be provided.
- If this information is not available online, the authors are encouraged to reach out to the asset's creators.

13. **New assets**

    Question: Are new assets introduced in the paper well documented and is the documentation provided alongside the assets?

    Answer: [Yes]

    Justification: The paper introduces a new dataset (PolypSense3D) as its primary asset. The code and data associated with this paper will be released concurrently with the paper's formal submission, providing further documentation and accessibility of these new assets.

    Guidelines:

    - The answer NA means that the paper does not release new assets.
    - Researchers should communicate the details of the dataset/code/model as part of their submissions via structured templates. This includes details about training, license, limitations, etc.
    - The paper should discuss whether and how consent was obtained from people whose asset is used.
    - At submission time, remember to anonymize your assets (if applicable). You can either create an anonymized URL or include an anonymized zip file.

14. **Crowdsourcing and research with human subjects**

    Question: For crowdsourcing experiments and research with human subjects, does the paper include the full text of instructions given to participants and screenshots, if applicable, as well as details about compensation (if any)?

    Answer: [NA]

    Justification: The paper does not involve crowdsourcing or direct interaction with human subjects in the data collection process. Clinical data was collected from routine procedures, and the paper details the ethical review process and anonymization procedures used to protect patient privacy.

    Guidelines:

    - The answer NA means that the paper does not involve crowdsourcing nor research with human subjects.
    - Including this information in the supplemental material is fine, but if the main contribution of the paper involves human subjects, then as much detail as possible should be included in the main paper.
    - According to the NeurIPS Code of Ethics, workers involved in data collection, curation, or other labor should be paid at least the minimum wage in the country of the data collector.

15. **Institutional review board (IRB) approvals or equivalent for research with human subjects**

Question: Does the paper describe potential risks incurred by study participants, whether such risks were disclosed to the subjects, and whether Institutional Review Board (IRB) approvals (or an equivalent approval/review based on the requirements of your country or institution) were obtained?

Answer: [Yes]

Justification: This study has obtained ethical approval from the Institutional Review Board (IRB) of Sir Run Run Shaw Hospital, Zhejiang University. The research strictly adheres to the NeurIPS Code of Ethics and complies with institutional and national regulations governing human subjects research.

Guidelines:

- The answer NA means that the paper does not involve crowdsourcing nor research with human subjects.
- Depending on the country in which research is conducted, IRB approval (or equivalent) may be required for any human subjects research. If you obtained IRB approval, you should clearly state this in the paper.
- We recognize that the procedures for this may vary significantly between institutions and locations, and we expect authors to adhere to the NeurIPS Code of Ethics and the guidelines for their institution.
- For initial submissions, do not include any information that would break anonymity (if applicable), such as the institution conducting the review.

16. **Declaration of LLM usage**

Question: Does the paper describe the usage of LLMs if it is an important, original, or non-standard component of the core methods in this research? Note that if the LLM is used only for writing, editing, or formatting purposes and does not impact the core methodology, scientific rigorousness, or originality of the research, declaration is not required.

Answer: [NA]

Justification: LLMs are not part of this work's core methods, experiments, or scientific contributions.

Guidelines:

- The answer NA means that the core method development in this research does not involve LLMs as any important, original, or non-standard components.
- Please refer to our LLM policy (`https://neurips.cc/Conferences/2025/LLM`) for what should or should not be described.

