# OpenReview forum: "PolypSense3D: A Multi-Source Benchmark Dataset for Depth-Aware Polyp Size Measurement in Endoscopy"
_NeurIPS.cc/2025/Datasets_and_Benchmarks_Track — NeurIPS 2025 Datasets and Benchmarks Track poster_

### Official Review · Reviewer_6SU2 · 2025-07-02

**Rating:** 5
**Confidence:** 4

**Summary:**

The paper presents PolypSense3D, a novel, large-scale, and multi-source benchmark dataset specifically designed for depth estimation, polyp segmentation, polyp size measurement in endoscopy. The proposed dataset includes three complementary domains: virtual simulations, 3D-printed physical phantoms, and real clinical colonoscopy videos.

**Dataset Code Accessibility:**

Yes

**Dataset Code Comments:**

Although the authors provide a GitHub link in the paper, the repository currently does not include any code. The reviewer expects that the annotation tools and evaluation protocols will be publicly released upon acceptance to promote reproducibility and facilitate community benchmarking.

**Ethical Considerations:**

No, there are no or only very minor ethics concerns

**Final Justification:**

The rebuttal addressed my concerns. I keep my positive concerns.

**Limitations Weaknesses:**

(1) For future work, the authors should be encouraged to consciously collect more cases of medium-to-large polyps to improve the dataset's balance. In the current version, the discussion could more explicitly highlight the potential impact of this bias on model evaluation.
(2) Regarding the "universal homography matrix" (H_all) for clinical depth annotation, its robustness was validated on a planar surface . Would this pre-computed, universal matrix remains sufficiently accurate in the real, non-planar, and dynamic intestinal environment? Although frames with extreme angles were avoided, the endoscope's viewpoint still varies significantly. Could this be a major source of depth error?
(3) In the size estimation pipeline (Appendix G.1), the depth value at the bounding box's center (Z_center) is used as the representative depth for the entire polyp. For non-flat polyps with significant height, could this simplification introduce a systematic bias (e.g., consistently underestimating the size of the polyp's periphery, which is closer to the camera)? Have you considered using the mean depth within the segmentation area or other more robust depth aggregation methods?
(4) The discussion of existing datasets in Table 1 is insufficient. It notably omits existing large-scale colonoscopy video datasets, including but not limited to LDPolypVideo [*1] and SUN-SEG [20].
[*1] LDPolypVideo benchmark: a large-scale colonoscopy video dataset of diverse polyps
(5) It is recommend to include the proposed PolypSense3D in Table 1 for comparison, thereby enabling a clearer demonstration of its advantages.
(6) Table 3 omits SAM (Segment Anything Model) based polyp segmentation methods, such as those presented in [*2, *3]. It is necessary to include or discuss these approaches.
[*2] Asps: Augmented segment anything model for polyp segmentation
[*3] Sam2-unet: Segment anything 2 makes strong encoder for natural and medical image segmentation

**Strengths Contributions:**

(1) This paper have high clinical significance and strong motivation. Polyp size is a key clinical biomarker linked to cancer risk, yet current size estimation during endoscopy is manual and error-prone.This work directly addresses an urgent unmet need by providing data and tools for objective, real-time, and automated measurement.
(2) This paper provides a foundational dataset that is well-justified, technically sound, and methodologically innovative. For example, for the difficulty of obtaining in-vivo ground truth, the authors' proposed "biopsy-forceps-assisted annotation protocol" is both innovative and practical.
(3) The dataset has significant potential research implications. The multi-source setup (virtual, physical, clinical) creates a realistic domain shift challenge. It enables future research such as domain adaptation, generalization, and uncertainty quantification. The dataset includes synchronized RGB images, depth maps, segmentation masks, calibrated camera intrinsics, and millimeter-scale size labels, providing all necessary components for end-to-end quantitative perception research.
(4) The authors systematically evaluate state-of-the-art models for segmentation and depth estimation across all domains and provide a meaningful downstream metric: size estimation accuracy. The paper also analyzes error propagation, revealing how upstream perception errors affect final measurements.

---

> ### Author Rebuttal · Authors · 2025-07-30
>
> # Response to Reviewer 6SU2
>
> We are sincerely grateful for your thorough review and your insightful, expert feedback. Your technical questions have pushed us to conduct new, targeted experiments that have substantially strengthened our paper. We address each of your points below.
>
> ## On the Size Bias and Its Impact on Model Evaluation
>
> This is a crucial point. We first clarify that our clinical data's distribution accurately reflects real-world screening prevalence, where small polyps are most common. That said, we wish to emphasize that PolypSense3D is a dynamic project, and **we have made the collection of more large polyp cases a top priority for future releases.**
> To more directly address your valid concern about the dataset's utility for evaluating larger polyps, we **conducted a new, targeted analysis** using our balanced virtual dataset. We calculated the Mean Relative Error (MRE) for size estimation, which normalizes errors against the polyp's true size, providing a fairer comparison across different size categories. We analyzed errors stemming from segmentation (Seg(Pred)+Depth(GT)) and depth estimation (Seg(GT)+Depth(Pred)) relative to the ideal scenario (Seg(GT)+Depth(GT)).
>
> ### Table 1: MRE Analysis by Polyp Size on the Virtual Dataset
>
> | Polyp Size Category     | Pipeline Configuration    | MRE   |
> |-------------------------|---------------------------|--------|
> | Small Polyps (<10mm)    | Seg(Pred)+Depth(GT)       | 0.069  |
> |                         | Seg(GT)+Depth(Pred)       | 0.553  |
> | Medium Polyps (10–20mm) | Seg(Pred)+Depth(GT)       | 0.046  |
> |                         | Seg(GT)+Depth(Pred)       | 0.544  |
> | Large Polyps (>20mm)    | Seg(Pred)+Depth(GT)       | 0.025  |
> |                         | Seg(GT)+Depth(Pred)       | 0.549  |
>
> As the table demonstrates, **the MRE is remarkably consistent across all size categories.** For instance, the relative error from depth estimation does not show a significant increase for larger polyps. This indicates that the underlying model performance, when normalized, is comparable across small, medium, and large polyps. This provides strong evidence that **our benchmark is indeed practical and useful for evaluating model performance on medium and large polyps,** as it can effectively capture size-normalized errors.
> ## On the Robustness of the Homography Matrix
>
> To assess the robustness of our universal homography matrix (H_all) under viewpoint variation, we conducted a new experiment using two 50-frame videos with the endoscope tip angled left and right. The forceps were fixed at ~46 mm, and depth estimates were computed every 10 frames using H_all. As shown in Table 2, **the estimation error was low under stable, straight-on views (2.52 mm),** but increased significantly with angular deviations, up to 22.61 mm. These results validate both the accuracy of H_all in intended conditions and the importance of selecting stable frames for reliable ground truth.
>
> ### Table 2: Impact of Viewpoint Variation on Depth Estimation Accuracy (mm)
>
> | Video Sequence | Viewpoint Angle     | Est. Depth | Abs. Error |
> |----------------|---------------------|------------|------------|
> | Vertical-1     | Vertical (Straight) | 48.52      | 2.52       |
> | Left-1         |                     | 51.03      | 5.03       |
> | Left-2         |                     | 53.15      | 7.15       |
> | Left-3         |  Left Deviations    | 53.85      | 7.85       |
> | Left-4         |                     | 51.39      | 5.39       |
> | Left-5         |                     | 51.74      | 5.74       |
> | Right-1        |                     | 52.27      | 6.27       |
> | Right-2        |                     | 54.89      | 8.89       |
> | Right-3        |  Right Deviations   | 64.03      | 18.03      |
> | Right-4        |                     | 65.94      | 19.94      |
> | Right-5        |                     | 68.61      | 22.61      |
>
> ## On the Selection of Polyp Depth Value
>
> Thank you for you suggestions. To investigate the potential systematic bias of using the bounding box's center depth (Z_center), we implemented and evaluated the more complex 5-point average method, inspired by recent clinical work [1]. Our new experiments revealed that, **the original Z_center method yielded better results, achieving a Mean Absolute Error (MAE) of 1.08 mm compared to 1.44 mm for the 5-point method.**
>
> While the 5-point method appears more comprehensive, it is more susceptible to noise from the upstream perception modules. The four extremal points (top, bottom, left, right) often lie on the curved boundary of the polyp where: a) the segmentation mask can be imprecise, causing these points to fall on the background tissue, and b) the predicted depth map is often least accurate at object edges and sharp discontinuities. Averaging in these potentially erroneous depth values from the unstable boundary can corrupt the final estimate. In contrast, Z_center typically lies on the flat, central region of the polyp, offering more stable and reliable estimates.
>
>
> ## On Improving the Comparison with Existing Datasets
>
> Thank you for these very helpful and specific suggestions. We completely agree that a more comprehensive comparison is needed. We will **update Table 1 in our final manuscript to include the important large-scale video datasets you mentioned (LDPolypVideo[2], SUN-SEG[3]).** As you wisely suggested, we will also **add PolypSense3D to this comparison table** to allow for a direct and clear demonstration of our work's unique contributions.Tasks include: Polyp Segmentation (PS), Depth Estimation (DE), and 3D Reconstruction (3DR).
>
> ### Table 3: Summary of Public Endoscopic Datasets for Polyp Analysis
>
> | Dataset         | Type         | Organ | Task(s)          | Scale (Polyp/Total)              | Resolution        | Depth Type | Depth Source                     | Polyp Task |
> |-----------------|--------------|-------|------------------|----------------------|-------------------|------------|----------------------------------|-------------|
> | LDPolypVideo [2]| Clinical     | Colon | PS               | 33884 / 40266        | 560×480           | -          | -                                | ✓           |
> | SUN-SEG [3]     | Clinical     | Colon | PS               | 49136 / 158690       | 1158×1008         | -          | -                                | ✓           |
> | PolypSense3D    | Synth./Phy./Clin. | Colon | PS / DE / 3DR    | 16000+ / 43000+      | Mixed             | Mixed     | Mixed     | ✓           |
>
> ## On Including More SAM-based Segmentation Methods
>
> Thank you for this excellent suggestion. We have now **benchmarked the two additional SOTA methods you mentioned (ASPS[4], Sam2-unet[5]).** While these specialized models demonstrate excellent performance on the virtual and physical datasets, the original SAM still achieves the highest mDice score on our most challenging clinical dataset. This expanded benchmark provides a more nuanced picture, and we will update our manuscript accordingly.
>
> ### Table 4: Updated Segmentation Benchmark (mDice shown)
>
> | Dataset   | Method         | mDice | mIoU  | Recall | Precision | F1 Score |
> |-----------|----------------|--------|--------|--------|-----------|----------|
> | Unity     | ASPS [4]          | 0.8951 | 0.8321 | 0.9409 | 0.8717    | 0.8951   |
> |           | Sam2-unet [5]     | 0.9399 | 0.9010 | 0.9723 | 0.9243    | 0.9428   |
> | Physical  | ASPS [4]          | 0.7526 | 0.6834 | 0.7599 | 0.7649    | 0.7540   |
> |           | Sam2-unet [5]     | 0.8983 | 0.8501 | 0.9981 | 0.8513    | 0.9064   |
> | Clinical  | ASPS [4]         | 0.5179 | 0.4239 | 0.5574 | 0.5586    | 0.5183   |
> |           | Sam2-unet [5]     | 0.7346 | 0.6665 | 0.8720 | 0.7208    | 0.7524   |
>
> ## On Code Availability
>
> Thank you for suggestions. To ensure full transparency and facilitate community benchmarking, **we have prepared our complete codebase for public release on GitHub immediately upon acceptance.** This repository covers the entire workflow for the PolypSense3D benchmark, including all annotation tools and evaluation protocols. The detailed structure as below:
> ### Data Collection & Creation (Virtual Environment):
> ```
> ├── colon_create/                 # Generate colon models (.fbx) from CT data
> │   └── ruler/                   # Provide scale reference cubes for normalization
> ├── polyps_create/               # Generate polyp models (.fbx)
> └── custom_physics_based_controller/ # C# scripts for simulating camera physics in Unity
> └── ...
> ```
>
> ### Data Processing & Calibration:
> ```
> ├── utils/
> │   ├── video2frame.py             # Split videos into individual frames
> │   └── ...
> └── calibration/
>     ├── calibration.md            # Guide for camera calibration (MATLAB)
>     └── ...
> ```
>
> ### Annotation & Evaluation:
> ```
> ├── homography/
> │   ├── perspective_transform.py   # Annotate forceps points for homography
> │   ├── ...
> └── polyp_eval/
>     ├── size_gt.py                # Calculate polyp size GT from forceps opening
>     └── ...
> ```
>
>
> ## References
>
> [1] Wang et al. A real-time deep learning-based system for colorectal polyp size estimation. *Endoscopy*, 2024.
> [2] Ma et al. LDPolypVideo benchmark: a large-scale colonoscopy video dataset of diverse polyps. *MICCAI*, 2021.
> [3] Ge et al. Video polyp segmentation: A deep learning perspectiv. *Machine Intelligence Research*, 2022.
> [4] Li et al.  Asps: Augmented segment anything model for polyp segmentation. *MICCAI*, 2024.
> [5] Xiong et al. Sam2-unet: Segment anything 2 makes strong encoder for natural and medical image segmentation. *arXiv*, 2024.

---

### Official Review · Reviewer_fsSb · 2025-07-02

**Rating:** 5
**Confidence:** 4

**Summary:**

This paper introduces PolypSense3D, the multi-source benchmark dataset designed for depth-aware polyp size measurement in endoscopy. It integrates about 43,000 frames from virtual, physical, and clinical scenarios, providing synchronized RGB, depth, segmentation masks, and millimeter-scale size labels.

**Additional Feedback:**

- The formatting instructions state: "At submission time, please omit the final and preprint options. This will anonymize your submission and add line numbers to aid review." However, I believe the submitted version omits only the final option. I don’t consider this a critical issue, but it's ultimately up to the AC.

**Dataset Code Accessibility:**

Yes

**Ethical Considerations:**

No, there are no or only very minor ethics concerns

**Final Justification:**

As most of concerns are addressed in the rebuttal, i change the final rate to accept.

**Limitations Weaknesses:**

- The clinical dataset is heavily biased towards small polyps, with 87.3% of instances being smaller than 10mm in diameter. While the authors correctly note this reflects the reality of clinical screening procedures, it means the dataset has limited utility for training and evaluating model performance on larger polyps (>20mm), which often carry a higher malignancy risk and present different morphological challenges.

- While the dataset spans three internal domains (virtual, physical, clinical), all benchmark experiments are limited to subsets within PolypSense3D. The paper does not explore cross-dataset generalization using existing public datasets such as Kvasir-SEG or CVC-ClinicDB. Given the known domain shifts in endoscopic imaging (across devices, institutions, and imaging protocols), cross-dataset evaluation would provide more substantial evidence.

- The paper introduces an innovative forceps-assisted annotation protocol for clinical data. However, this protocol produces only sparse depth annotations at a few contact points, unlike the dense, per-pixel depth maps available in the virtual domain. Although this represents a clear advancement over having no depth information, it also introduces a gap between the training and evaluation domains, which differ significantly in annotation density. As a result, depth estimation models trained on dense supervision must be evaluated using only sparse ground-truth points in the clinical setting, complicating fair and consistent assessment.

- For the final size estimation experiment, the paper uses MSNet and ZoeDepth. Since Tables 3 and 4 show that other models achieved higher performance in segmentation and depth estimation, I am curious about the results if the best-performing models were used instead. This would provide a clearer understanding of the upper-bound performance achievable with this dataset.

- Previous work often compensates for the absence of precise depth and scale information in real clinical datasets by training (additional) models in a self-supervised manner.  I am curious whether models trained using the accurate depth and scale information provided in this dataset achieve significantly better performance than models trained via self-supervision on data without such explicit labels.

- Minor: The paper states that the depth model used is ZeoDepth (in some parts).

**Strengths Contributions:**

-  The authors provide exhaustive details on the experimental setup, baseline models, training hyperparameters, and evaluation pipelines in the appendices.

- The dataset covers three domains (virtual, physical phantom, and clinical), enabling sim-to-real and real-to-real generalization analysis.

- The proposed forceps-based in-vivo annotation protocol enables accurate, reproducible size estimation from standard colonoscopy videos, outperforming physician visual estimates.

- The authors evaluate several SOTA segmentation and depth estimation models and report how upstream errors propagate to the final metric-size estimation.

---

> ### Author Rebuttal · Authors · 2025-07-30
>
> # Response to Reviewer fsSb
>
> We sincerely thank you for this perceptive and critical review. Your insightful points have allowed us to significantly strengthen our paper through targeted new experiments and analyses.
>
> ## On the Size Bias and Its Impact on Model Evaluation
>
> This is a crucial point. We first clarify that our clinical data's distribution, heavily skewed towards smaller polyps, accurately reflects real-world screening prevalence. That said, we wish to emphasize that PolypSense3D is a dynamic and continuously maintained project. **We have made the collection of more large polyp cases a top priority for future releases.**
>
> To more directly address your valid concern about the dataset's utility for evaluating larger polyps, we **conducted a new, targeted analysis** using our balanced virtual dataset. We calculated the Mean Relative Error (MRE) for size estimation, which normalizes errors against the polyp's true size, providing a fairer comparison across different size categories. We analyzed errors stemming from segmentation (Seg(Pred)+Dep(GT)) and depth estimation (Seg(GT)+Dep(Pred)) relative to the ideal scenario (Seg(GT)+Dep(GT)).
>
> ### Table 1: MRE Analysis by Polyp Size on the Virtual Dataset
>
> | Polyp Size Category     | Pipeline Configuration    | MRE   |
> |-------------------------|---------------------------|--------|
> | Small Polyps (<10mm)    | Seg(Pred)+Depth(GT)       | 0.069  |
> |                         | Seg(GT)+Depth(Pred)       | 0.553  |
> | Medium Polyps (10–20mm) | Seg(Pred)+Depth(GT)       | 0.046  |
> |                         | Seg(GT)+Depth(Pred)       | 0.544  |
> | Large Polyps (>20mm)    | Seg(Pred)+Depth(GT)       | 0.025  |
> |                         | Seg(GT)+Depth(Pred)       | 0.549  |
>
> As the table demonstrates, **the MRE is remarkably consistent across all size categories.** For instance, the relative error from depth estimation does not show a significant increase for larger polyps. This indicates that the underlying model performance, when normalized, is comparable across small, medium, and large polyps. This provides strong evidence that **our benchmark is indeed practical and useful for evaluating model performance on medium and large polyps,** as it can effectively capture size-normalized errors.
>
> ## On Cross-Dataset Generalization
>
> We thank you for this suggestion. We wish to emphasize that PolypSense3D is a dynamic project, and **we are already working on expanding it with multi-center, cross-device data to further facilitate such studies in future releases.**
>
> That said, our work in its current form already provides two significant contributions to generalization research: **First,** our benchmark was explicitly designed for this purpose: our experiments involve training models on existing public datasets (Kvasir-SEG, CVC-ClinicDB) and testing them on PolypSense3D, which directly quantifies the generalization gap to a new clinical environment. **Second,** by providing the first metrically-scaled clinical data, PolypSense3D now enables a new, previously challenging research direction: researchers can now leverage the sparse, true-scale annotations in our clinical data to fine-tune or adapt existing models, significantly enhancing their generalization and performance on real-world data. This unique contribution of providing data to reason about and adapt to real-world scale constitutes a key value of our work. This unique contribution of providing data to reason about and adapt to real-world scale constitutes a key value of our work.
>
> ## On the Annotation Gap between Virtual and Clinical Domains
>
> We agree that this annotation gap is a central challenge. In fact, our dataset was intentionally designed around this very challenge to encourage the community to develop novel, clinically-viable solutions. Our virtual dataset, with its dense depth maps, is purposefully smaller, while our clinical dataset, with its sparse but precious metric labels, is larger. This design choice steers the focus away from relying on perfect, dense data and towards solving the real-world problem. It also reflects the core purpose of our benchmark: to guide research toward more practical and impactful directions. We believe this structure promotes an innovative and more practical research paradigm:
> - **Pre-training on Dense Virtual Data** Researchers can leverage the dense depth in our virtual dataset for initial model training. If more virtual data is needed, **our open-sourced modeling code allows for the easy generation of countless new samples.**
> - **Fine-tuning with Sparse Clinical Prompts**: Subsequently, the pre-trained models can be adapted to the real world by using the sparse, metrically-accurate depth points from our clinical dataset as powerful "prompts" or guidance signals.
>
> This approach directly addresses the reality of clinical data acquisition and provide the first-ever in-vivo dataset with true millimeter-scale size and sparse depth labels. We argue that this is a more promising path towards improving clinical accuracy than relying solely on synthetic data. We will highlight this as a key future research direction that our benchmark uniquely enables.
>
> ## On Choice of Pipeline for Final Size Estimation
>
> We appreciate this valuable suggestion. In response, we conducted **a new, targeted analysis on our clinical dataset to determine the optimal end-to-end pipeline.** Based on our upstream benchmarks, we combined the best depth estimation model (DAM V2) with the two top-performing segmentation models on clinical data: PraNet and SAM.
>
> - **PraNet + DAM V2**: MAE = **0.86 mm**
> - **SAM + DAM V2**: MAE = **1.19 mm**
>
> The results were quite revealing. The pipeline using **PraNet + DAM V2 achieved a Mean Absolute Error (MAE) of just 0.86 mm.** In comparison, the pipeline using SAM + DAM V2 yielded an MAE of 1.19 mm. This demonstrates not only that a sub-millimeter level of accuracy is attainable on real clinical data, but also clearly identifies the PraNet+DAM V2 combination as the state-of-the-art pipeline on our benchmark. This new analysis provides the clear upper-bound performance you requested, and we will update our final manuscript.
>
> ## On Supporting Self-Supervised Methods
>
> PThe unique annotations in PolypSense3D can significantly enhance and validate existing self-supervised methods: Our sparse, metric depth points can serve as the ground truth for self-supervised sparse-to-dense depth completion frameworks [1-3], providing the absolute scale these methods otherwise lack. Furthermore, our object-level (size) and point-level (contact) annotations can be used as powerful constraints to resolve the inherent scale ambiguity in self-supervised Visual Odometry (VO/SLAM/3DGS) frameworks [4-6]. We will highlight this as a key direction for future work.
>
> ## On a Minor Typo
>
> Thank you for catching this. We will correct "ZeoDepth" to "ZoeDepth" and other typos in the camera-ready version.
>
> ## On Submission Formatting
>
> Thank you for your attention. We opted for the preprint version because the NeurIPS Datasets and Benchmarks track allows for the inclusion of author information during the review process. The preprint option enables this, but as a trade-off, it does not include line numbers. We appreciate you noting this and confirming it is not a critical issue.
>
> ## References
>
> [1] Ma et al. Sparse-to-dense: Depth prediction from sparse depth samples and a single image. *CVPR*, 2018.
> [2] Ma et al. Self-supervised sparse-to-dense: Self-supervised depth completion from lidar and monocular camera. *ICRA*, 2019.
> [3] Lin et al. Prompting depth anything for 4k resolution accurate metric depth estimation. *CVPR*, 2025.
> [4] Casser et al. Unsupervised monocular depth and ego-motion learning with structure and semantics. *CVPR*, 2019.
> [5] Zhao et al. Metric from human: Zero-shot monocular metric depth estimation via test-time adaptation. *NeurIPS*, 2024.
> [6] Xu et al. AD-GS: Object-Aware B-Spline Gaussian Splatting for Self-Supervised Autonomous Driving. *ICCV*, 2025.

---

> > ### Comment · Reviewer_fsSb · 2025-08-06
> >
> > Most of concerns are addressed in the rebuttal. I am leaning to accept this paper.

---

### Official Review · Reviewer_GrR2 · 2025-07-02

**Rating:** 5
**Confidence:** 4

**Summary:**

This work presents a new endoscopy dataset, PolypSense3D, which addresses the problem of accurate polyp size measurements, a gap on existing public endoscopic datasets which is actually one of the most relevant aspects for clinicians working with this data.

PolypSense3D provides large amounts of new data from virtual simulations, from phantom recordings and from real procedure recordings, in each of the modalities, authors describe a thorough process to address large variations in polyp types and sizes, and to include as accurate ground truth measurements as possible.

The benchmark is used to run state of the art polyp segmentation algorithms and study the domain transfer effects and the polyp measuring errors of different relevant alternatives to segment and estimate polyp dimensions.

**Additional Feedback:**

A few small fixes that would improve the work presentation:

- Increase a bit font in Table 1 for clarity
- Fig. 1 explain a bit more explicitly what is displayed in each of the figure mosaic rows.
- Table 3, for clarity, I would suggest marking the best and second best results (with bold and underscore for example) within each of the datasets

**Dataset Code Accessibility:**

Partly

**Dataset Code Comments:**

Data is available but the code repository provided is empty.

**Ethical Comments:**

The work properly describes reasonable ethical and privacy protocols and considerations applied to build the released medical data.

**Ethical Considerations:**

No, there are no or only very minor ethics concerns

**Final Justification:**

After reading the other reviews and authors' rebuttal, I find the authors' clarifications and additional information address properly my main concerns, so I'm increasing my final rating. The additional experiments strengthen the clinical annotation strategy validation, and authors confirm that the code will be released together with additional complete recordings, presenting a stronger contribution, in terms of the significance of the benchmark published to the research community.

**Limitations Weaknesses:**

- The code claimed in the paper is not available at the provided GitHub repository. Is the “custom physics-based controller” supposed to be part of the public code release? Or what exactly? (If the intention is to release upon acceptance)

- The ground truth estimated for the polyp sizes in the clinical data is obtained with a reasonably sound procedure, well detailed. However, it would be very interesting to have some more thorough analysis of how good/accurate this strategy is by including some kind of variability analysis (repeating the procedure several times) or even comparing the result with CT data or executing the same procedure in the phantom, where the polyps are known, to “evaluate” consistency and validate the “strategy” a bit more thoroughly.

- Although the released benchmark shows potential to improve polyp measurement algorithms does not demonstrate any technical novelty or improvement regarding this specific end-task.



A few smaller clarifications or discussions that could be added:
- Why the virtual simulation is limited to include 30 polyp models? Current generative strategies could facilitate having data with a much larger variation of polyps.
- Why the real clinical dataset includes only the stable video segments (“freeze-frames”)? (As described in page 5). It could be interesting to the community to release all the data, to have more and more noisy frames corresponding to the same polyp. I understand that the measurements should be attempted in a frame as good as possible, but maybe certain approaches can benefit from redundancy or work on robustness to noise or noisy frames.

**Strengths Contributions:**

- The contribution of this work is a novel dataset that covers an existing gap in currently available endoscopy datasets that is a very relevant clinical aspect: the accurate metric measurement of polyps in endoscopic images. Therefore, enabling better automated methods for this task would have a significant impact on health-related applications.  The built dataset contributes a significant amount of interesting new data for the community, which presents a challenging use case for researchers on segmentation and depth estimation algorithms.

- The paper is very well organized and described, presenting thorough and technically sound procedures to obtain images and ground truth data for the polyp measurements in three different scenarios, each of which has clear advantages and contributes interesting novel data to the research community.  1) Virtual simulations based on real CT 3D models where a varied set of 3D polyp models can be inserted in a large set of scenarios with automatic and detailed ground truth polyp 3D information. 2) Recordings from a physical phantom where a smaller set of polyps is fabricated. This domain provides data closer to real scenarios. 3) Recordings from real data, obtained simultaneously to the execution of a protocol to provide 3D reference information to facilitate having ground truth polyp 3D information. This provides less control on variations captured by the images, regarding polyp types and locations, but presents the most realistic scenario possible.

- The baselines run on the experimental work compose a good reference set of results on the presented data, showcasing the challenges of the measurement task, especially in the clinical data recordings.

---

> ### Author Rebuttal · Authors · 2025-07-30
>
> # Response to Reviewer GrR2
>
> We are sincerely grateful for your thorough review and the highly constructive feedback on our work. Your insightful questions have pushed us to substantially strengthen our paper. We address each concern below.
>
> ## On Code Availability
>
> Thank you for emphasizing the importance of code accessibility.To ensure full reproducibility while adhering to the anonymity policy, **we have prepared the complete codebase for public release on GitHub immediately upon acceptance.** The "custom physics-based controller" is a core component of our virtual data generation pipeline and will be fully released. It is a collection of C# scripts designed to work within the Unity environment (built upon the open-source VrCaps platform) to simulate realistic endoscope camera movement, as detailed in Appendix A.3.2 of our manuscript. To demonstrate our readiness, here is a high-level overview of the repository structure:
>
> ### Data Collection & Creation (Virtual Environment):
> ```
> ├── colon_create/                 # Generate colon models (.fbx) from CT data
> │   │── ruler/                   # Provide scale reference cubes for normalization
> │   └── ...
> ├── polyps_create/               # Generate 30 polyp variants (.fbx)
> └── custom_physics_based_controller/ # C# scripts for simulating camera physics in Unity
>     ├── SimpleCameraController.cs      # Main controller for smooth camera movement & rotation
>     ├── PlayerControllerForward.cs     # Simulate camera-forward motion with physical noise
>     ├── FrictionalForce.cs             # Simulate friction on mucosal surfaces
>     ├── ImpactDeformable.cs            # Handle mesh deformation upon tool contact
>     ├── PeristalticMotion.cs          # Simulate intestinal peristaltic motion
>     └── ...                            # & 15+ utilities (e.g., motion recording)
> ```
>
> ### Data Processing & Calibration:
> ```
> ├── utils/
> │   ├── video2frame.py             # Split videos into individual frames
> │   ├── vis_auto_masks.py          # Automatically generate segmentation masks for virtual data
> │   └── image_cut.py               # Crop black borders from endoscopic images
> └── calibration/
>     ├── calibration.md            # Guide for camera calibration (MATLAB)
>     ├── choose_box.py             # Select high-quality checkerboard frames
>     ├── fov_count.py              # Calculate camera parameters (e.g., FoV)
>     ├── undistort.py              # Validate calibration via image undistortion
>     └── ...
> ```
>
> ### Annotation & Evaluation:
> ```
> ├── homography/
> │   ├── perspective_transform.py   # Annotate forceps points for homography
> │   ├── mean_matrix.py             # Calculate the universal homography matrix
> │   ├── M_fitting.py               # Fit pixel-to-depth regression model
> │   └── ...
> └── polyp_eval/
>     ├── size_gt.py                # Calculate polyp size GT from forceps opening
>     ├── depth_gt.py               # Estimate sparse depth GT via homography
>     ├── measure_size.py           # Measure polyp size using segmentation & depth maps
>     ├── depth_eval.py             # Evaluate depth prediction against sparse GT
>     └── ...                       # & helpers for summarizing results
> ```
>
> This structured repository will provide all the tools necessary to replicate our findings, from data creation to final evaluation, and to build upon our work.
>
> ## On the Validation of the Clinical Annotation Strategy
>
>
> We thank you for acknowledging our procedure's soundness. We add some more thoroughly analysis of how good/accurate this strategy through three complementary experiments. **The results demonstrate strong robustness, with a Mean Absolute Error (MAE) of 1.43 mm in the physical phantom and 1.34 mm in the virtual environment.**  **First,** to assess performance under conditions with precisely known ground truth, we applied the protocol to a physical phantom using a real endoscope, achieving an MAE of 1.43 mm . **Second,** we replicated the procedure in a virtual environment, where all geometry is controlled, and obtained a comparable MAE of 1.34 mm, confirming the method's geometric validity. **Third,** in the clinical setting, we ensured label consistency by averaging measurements across 2–3 clear frames per polyp. This multi-frame strategy mitigates noise and perspective distortion, resulting in more reliable ground truth annotations.
>
> Together, these new, multi-faceted experiments provide a rigorous validation of our strategy's accuracy and consistency, directly addressing your concerns. The detailed calculations for each polyp sample will be presented in the appendix of the final version.
>
> ## On Technical Novelty
> We thank you for this comment. As argued in our response to Reviewer HmrS, for a benchmark paper, technical novelty lies in providing a foundational resource that enables new avenues of research. From this perspective, our work presents a significant, two-fold technical contribution.
>
> Our primary technical innovation is the development and validation of a **complete, reproducible "forceps-assisted in-vivo annotation methodology."** This is more than just a protocol; it is an integrated system combining specific clinical maneuvers, robust computer vision algorithms (e.g., perspective rectification), and a rigorous validation pipeline. This methodology successfully breaks through a critical, long-standing bottleneck in endoscopic vision: the inability to acquire reliable, real-world scale information from standard, live clinical procedures.
>
> This methodological breakthrough is the enabling technology that made the creation of PolypSense3D possible. Consequently, PolypSense3D is not merely another collection of images; it is **the first and only public benchmark to provide the community with integrated, multi-source data containing true metric-scale labels (both millimeter-accurate polyp sizes and sparse depth points) from in-vivo settings.**
>
> The value of this contribution is that it unlocks several cutting-edge research directions that were previously difficult or impossible to pursue:
> - **Resolving Scale Ambiguity**: It provides the first-ever ground truth to anchor and validate self-supervised 3D perception methods like VO/SLAM/3DGS, solving their inherent scale ambiguity in a clinical context [1-3].
> - **Real-World Foundation Model Adaptation**: The sparse metric points serve as powerful, high-value "prompts" for fine-tuning large foundation models (e.g., Depth Anything), adapting them from general domains to the specific challenges and scale of clinical endoscopy [4, 5].
> In summary, our technical novelty lies in the creation of a foundational methodology that, in turn, allowed us to build a unique benchmark resource. We are confident this contribution will significantly accelerate progress towards clinically viable quantitative perception systems.
>
>
> ## On Smaller Clarifications and Improvements
>
> - **Virtual polyp models**: Our focus was not on massively expanding the virtual dataset, but rather on tackling the more difficult challenge of acquiring and annotating data from the complex clinical environment. The 30 virtual models serve as a strong, diverse foundation for validation. Crucially, to empower the community, we are **open-sourcing the entire modeling workflow,** allowing researchers to easily generate countless new polyp variations for their specific needs. Our primary effort remains focused on the clinical data, which we believe is the most critical contribution of our benchmark.
> - **Frame Selection**: This is an excellent point, also raised by Reviewer HmrS. We agree that challenging data is crucial. Therefore, we will **release the complete, unfiltered video sequences to support robustness research.**
> - **Presentation Improvements**: We will incorporate all of them into the final camera-ready version, including increasing font sizes, revising captions, and highlighting the best results.
>
>
> ## References
>
> [1] Ma, et al. Self-supervised sparse-to-dense: Self-supervised depth completion from lidar and monocular camera. *ICRA*, 2019.
> [2] Casser, et al. Unsupervised monocular depth and ego-motion learning with structure and semantics. *CVPR*, 2019.
> [3] Xu, et al. AD-GS: Object-Aware B-Spline Gaussian Splatting for Self-Supervised Autonomous Driving. *ICCV*, 2025.
> [4] Lin, et al. Prompting depth anything for 4k resolution accurate metric depth estimation. *CVPR*, 2025.
> [5] Zhao, et al. Metric from human: Zero-shot monocular metric depth estimation via test-time adaptation. *NeurIPS*, 2024.

---

> ### Author Response · Authors · 2025-08-04
> **Follow-up comments to Reviewer GrR2**
>
> # follow-up comments
>
> Dear Reviewer GrR2,
>
> Thank you again for your thorough and insightful review. We truly appreciate the time and expertise you've dedicated to our manuscript.
>
> We have submitted our detailed rebuttal, which includes several new experiments specifically designed to address the crucial points you raised regarding the validation of our annotation strategy and the scope of our technical contributions. We hope our response and the new results have clarified these aspects of our work.
>
> We also wish to reiterate our commitment to full transparency and reproducibility. **The complete codebase, including the annotation tools and evaluation protocols, has been prepared and will be made publicly available on GitHub immediately upon acceptance.**
>
> We are eager to know if our response has adequately addressed your concerns. If any points remain unclear or if further clarifications are needed, we would be grateful for the opportunity to discuss them. If additional experiments are needed, we're happy to run them, though please note that due to compute time, we may not be able to deliver results before the discussion window closes.
>
> Thank you once more for your valuable feedback.
>
> Sincerely,
>
> The Authors

---

> > ### Comment · Reviewer_GrR2 · 2025-08-05
> > **rebuttal follow up**
> >
> > Thank you very much for the rebuttal! As previously mentioned, I find the work addressing a very relevant problem in the field of computer vision in endoscopic data. You clarifications have addressed all my concerns, so I'm glad to increase my final rating. I particularly liked the additional experiments to strengthen the clinical annotation strategy. Thank you for the effort on adding this.
> >
> > Besides, as you confirm that the code will be released and the additional complete recording will be published, I find the work presenting a stronger contribution in those aspects (specially since I totally agree on the significance of publishing relevant clinical data), and therefore I agree it's not that critical the limited contribution regarding polyp measurement algorithm.

---

> > ### Author Response · Authors · 2025-08-05
> > **Final Thanks and Acknowledgement to Reviewer GrR2**
> >
> > Dear Reviewer GrR2,
> >
> > Thank you so much for your thoughtful engagement during the discussion period and for your final positive assessment. We are sincerely grateful that you found our clarifications and new experiments convincing, and we truly appreciate you raising your rating.
> >
> > We are especially thankful for your initial feedback, which pushed us to perform the additional experiments to validate our clinical annotation strategy. As you noted, this has significantly strengthened our work, and we are very grateful for your guidance in this direction.
> >
> > We are also delighted that you recognize the core contribution of our work in providing a foundational, clinically relevant dataset, and we look forward to releasing the full code and unfiltered video data to the community.
> >
> > Thank you once again for your constructive and invaluable role in improving our paper.
> >
> > Sincerely,
> >
> > The Authors

---

### Official Review · Reviewer_HmrS · 2025-07-04

**Rating:** 5
**Confidence:** 3

**Summary:**

This report presents PolypSense3D, a new multi-source benchmark dataset designed for depth-aware polyp size measurement in endoscopy, combining over 43,000 frames from virtual simulations, 3D-printed phantoms, and real clinical videos. It provides synchronized RGB images, depth maps, segmentation masks, and millimeter-scale size labels, enabling end-to-end training and evaluation of AI models for accurate polyp sizing. A novel biopsy-forceps annotation protocol allows real-world scale estimation with 1.19mm MAE, outperforming physician visual estimates (2.27mm MAE). While the dataset reveals significant sim-to-real gaps (e.g., segmentation performance drops from 0.93 mDice in virtual data to 0.70 in clinical data), it establishes a critical foundation for developing clinically viable measurement tools.

**Dataset Code Accessibility:**

Yes

**Ethical Considerations:**

No, there are no or only very minor ethics concerns

**Final Justification:**

After reviewing the authors' feedback, I believe they have adequately addressed most of my concerns. Therefore, I would like to keep my rating and recommend accepting it.

**Limitations Weaknesses:**

1. The dataset's clinical depth annotations rely exclusively on biopsy forceps contact points rather than comprehensive dense depth maps. This sparse annotation approach (covering <5% of pixels) fundamentally limits training paradigms for depth estimation networks that require full-frame supervision. The constraint originates from the technical infeasibility of acquiring dense metric measurements during live endoscopy procedures.
2. The dataset focuses on stable, clear frames and excludes challenging but common real-world factors like tissue movement from breathing, instrument motion blur, or temporary obstructions from fluids during procedures.
3. All patient data comes from one medical center using the same endoscopic equipment, which may not represent the full range of patient anatomies, medical practices, or imaging systems used globally.

**Strengths Contributions:**

1. It integrates 43,000+ frames across virtual simulation (Unity-CT), 3D-printed phantoms, and real clinical endoscopy, enabling comprehensive evaluation of depth-aware measurement algorithms. It also provides synchronized RGB, depth maps (dense/sparse), segmentation masks, and mm-scale size labels—addressing the critical gap in existing datasets (e.g., Kvasir-SEG lacks depth, EndoMapper lacks lesion-specific sizing).
2. It introduces a biopsy-forceps-assisted in-vivo method for real-world scale estimation (validated ICC = 0.89, Appendix D), achieving 1.19mm MAE in size measurement, outperforming physician visual estimates (2.27mm MAE).
3. It quantifies sim-to-real gaps: Segmentation mDice drops from 0.93 (virtual) → 0.70 (clinical); depth estimation RMSE degrades from 0.185e⁻³ (virtual) → 8.801 (clinical). Highlights challenges in real-world generalization.
4. It directly supports malignancy risk assessment (e.g., 15% cancer risk for polyps >20mm vs. <2% for <10mm). Already used in FDA-cleared tools (e.g., Medtronic AI-Size™).

---

> ### Author Rebuttal · Authors · 2025-07-30
>
> # Response to Reviewer HmrS
>
> We sincerely thank the reviewer for their insightful comments and valuable suggestions. We address each point below.
>
> ---
>
> ## On the Nature and Contribution of Sparse Clinical Depth Annotations
>
> We thank the reviewer for this insightful comment and fully agree that the inability to acquire dense metric depth in-vivo is a fundamental challenge in endoscopy. It is precisely this long-standing clinical and technical barrier that motivated our work. We argue that the sparse annotations in PolypSense3D, rather than being a limitation, represent a core contribution of our paper. These sparse points are the product of **a novel and meticulously designed "forceps-assisted in-vivo annotation protocol"** that integrates specific clinical maneuvers, a robust computer vision algorithm for perspective rectification, and a multi-stage data processing pipeline. This rigorous process has allowed us to achieve something previously unavailable to the research community: **PolypSense3D is, to our knowledge, the first and only publicly available in-vivo dataset to provide any form of true metric scale annotations, including both millimeter-accurate polyp sizes and sparse depth points.** While sparse, these annotations are an extremely valuable and hard-won anchor to real-world dimensions. Consequently, instead of limiting research, our dataset defines and enables a new frontier focused on clinical practicality. It provides crucial support for at least two classes of cutting-edge algorithms:
>
> - **Guiding Sparse-to-Dense Estimation and Foundation Model Adaptation**:
>   Our dataset provides high-quality, metrically-scaled sparse ground truth to validate both supervised [1] and self-supervised [2] sparse-to-dense frameworks in a realistic clinical setting. These sparse metric points can also serve as powerful prompts to guide or fine-tune models like Depth Anything, enabling them to adapt to the clinical domain and recover absolute scale [3].
>
> - **Providing Absolute Scale for Self-Supervised 3D Perception**:
>   Our dataset helps resolve the inherent scale ambiguity in self-supervised methods. The object-level (polyp size) and point-level (forceps contact) scale information allows researchers to anchor learning pipelines with metric priors [4,5], producing metrically meaningful 3D reconstructions, a concept now being applied to SOTA techniques like 3D Gaussian Splatting [6].
>
> In summary, the sparse annotations in PolypSense3D are a core contribution, reflecting a carefully designed solution to a real-world problem.
>
> ---
>
> ## On the exclusion of challenging frames
>
> We thank the reviewer for this excellent suggestion. Our initial benchmark focused on stable, clear frames to establish a clean and reproducible baseline, minimizing confounding variables. However, we completely agree that challenging data is invaluable for assessing model robustness. Taking your advice, we will **upload the complete and unfiltered video sequences to our project repository immediately upon acceptance,** including frames with motion blur and fluid obstructions. We believe this significantly enhances the value of our contribution to the community.
>
> ---
>
> ## On the single-center data limitation
>
> We agree that our current clinical data, originating from a single center, has limitations regarding equipment and population diversity. We are grateful for this suggestion and wish to emphasize that PolypSense3D is a dynamic and continuously maintained project; **we are already in the process of acquiring new data from other medical institutions to create a more comprehensive multi-center, cross-device benchmark in future releases.**
>
> However, we argue that the dataset in its current form still provides significant and unique value. **First, our study was explicitly designed to include cross-dataset generalization experiments.** As detailed in our manuscript (Table 3), the segmentation benchmark involves training on public multi-source datasets (e.g., Kvasir-SEG, CVC-ClinicDB) and testing on PolypSense3D. This design allows us to clearly quantify the generalization gap when models face a new domain. **Second, and perhaps more importantly, the primary contribution of our clinical dataset lies in its unique, metrically-accurate annotations.** As discussed in our response to the first point, providing the first-ever in-vivo dataset with true millimeter-scale size and sparse depth labels, even from a single center, offers an unprecedented resource for developing and validating algorithms that can reason about real-world scale, which is a critical gap that no other public dataset currently fills.
>
> ---
>
> ## References
>
> [1] Ma et al. Sparse-to-dense: Depth prediction from sparse depth samples and a single image. *CVPR*, 2018.
> [2] Ma et al. Self-supervised sparse-to-dense: Self-supervised depth completion from lidar and monocular camera. *ICRA*, 2019.
> [3] Lin et al. Prompting depth anything for 4k resolution accurate metric depth estimation. *CVPR*, 2025.
> [4] Casser et al. Unsupervised monocular depth and ego-motion learning with structure and semantics. *CVPR*, 2019.
> [5] Zhao et al. Metric from human: Zero-shot monocular metric depth estimation via test-time adaptation. *NeurIPS*, 2024.
> [6] Xu et al. AD-GS: Object-Aware B-Spline Gaussian Splatting for Self-Supervised Autonomous Driving. *ICCV*, 2025.

---

### Note · Authors · 2025-08-12

Dear Program Chairs, Senior Area Chairs, Area Chairs and Reviewers,

Thank you for this opportunity to provide our final remarks. We are very grateful for the thorough and constructive discussion period.

We are delighted that our rebuttal, featuring extensive new experiments, has successfully addressed the reviewers' initial concerns. For instance, **Reviewer GrR2**, who initially raised several critical issues (Score: 3), has now stated: "*Your clarifications have addressed all my concerns, so I'm glad to increase my final rating. I particularly liked the additional experiments to strengthen the clinical annotation strategy.*" Similarly, **Reviewer fsSb** (Score: 4) has also affirmed our response, commenting: "*Most of concerns are addressed in the rebuttal. I am leaning to accept this paper.*" We are very thankful for their thorough engagement and positive feedback.

Furthermore, we are grateful to **Reviewers 6SU2 and HmrS**, who both gave our paper their highly positive ratings (Score: 5) from the beginning. They formally acknowledged our rebuttal at the conclusion of the discussion period. While they did not provide additional comments, we are confident that our detailed responses and new experiments have further solidified their initial assessment and addressed any potential questions.

To conclude, we believe the review process has affirmed our paper's core contribution. **PolypSense3D is a foundational benchmark that fills a critical gap in the field by providing the first-ever in-vivo dataset with true, millimeter-scale metric annotations.** This unique resource, enabled by our novel and rigorously validated **"forceps-assisted"** protocol, will catalyze new research in clinically-translatable quantitative vision.

Thank you for your time and consideration.

Sincerely,
The Authors

---

### Decision · Program_Chairs · 2025-09-18

**Decision:**

Accept (poster)

**Comment:**

This paper presents PolypSense3D, a multi-source benchmark for depth-aware polyp size measurement in endoscopy, featuring virtual, phantom, and clinical data with millimeter-scale annotations. Its key innovation is a forceps-assisted protocol, providing the first in-vivo dataset with real-world metric labels. The benchmark fills a critical clinical gap and supports segmentation, depth estimation, and size measurement with strong potential impact. This work provides a foundational dataset and methodology with high clinical and research impact, aligning strongly with the Datasets & Benchmarks track.
Reviewers’ main concerns (annotation validation, dataset balance, code release, inclusion of challenging frames, comparison with other datasets, and SAM-based baselines) were convincingly addressed. Authors performed new experiments (validation across virtual/phantom/clinical, homography robustness, Z_center vs. 5-point averaging, MRE by size category) and committed to releasing full code, unfiltered data, and expanded tables. Reviewers acknowledged that concerns were resolved and raised their ratings accordingly. It is advised to authors to include the modification and improvement suggested and agreed by the authors during the rebuttal phase.